# Unifying Heterogeneous Degradations: Uncertainty-Aware Diffusion Bridge Model for All-in-One Image Restoration

Luwei Tu [1]   Jiawei Wu [1]   Xing Luo [2]   Zhi Jin [1 3 4]

## Abstract

All-in-One Image Restoration (AiOIR) faces the fundamental challenge in reconciling conflicting optimization objectives across heterogeneous degradations. Existing methods are often constrained by coarse-grained control mechanisms or fixed mapping schedules, yielding suboptimal adaptation. To address this, we propose an Uncertainty-Aware Diffusion Bridge Model (UDBM), which innovatively reformulates AiOIR as a stochastic transport problem steered by pixel-wise uncertainty. By introducing a relaxed diffusion bridge formulation, which replaces the strict terminal constraint with a relaxed constraint, we model the uncertainty of degradations while theoretically resolving the drift singularity inherent in standard diffusion bridges. Furthermore, we devise a dual modulation strategy: the noise schedule aligns diverse degradations into a shared high-entropy latent space, while the path schedule adaptively regulates the transport trajectory motivated by the viscous dynamics of entropy regularization. By effectively rectifying the transport geometry and dynamics, UDBM achieves state-of-the-art performance across diverse restoration tasks within a single inference step. Code is available at https://github.com/Jabruson/UDBM.

[1] School of Intelligent Systems Engineering, Shenzhen Campus of Sun Yat-sen University, Shenzhen, Guangdong 518107, China [2] Department of Strategic and Advanced Interdisciplinary Research, Pengcheng Laboratory, Shenzhen, Guangdong, 518055, P. R. China [3] Guangdong Provincial Key Laboratory of Fire Science and Technology, Guangzhou 510006, China [4] Guangdong Provincial Key Laboratory of Robotics and Digital Intelligent Manufacturing Technology, Guangzhou, 510535, China. Correspondence to: Zhi Jin <jinzh26@mail.sysu.edu.cn>.

*Proceedings of the 43rd International Conference on Machine Learning*, Seoul, South Korea. PMLR 306, 2026. Copyright 2026 by the author(s).

## 1. Introduction

Different from taks-specific image restoration (Li et al., 2022b; Wu et al., 2024; Jin et al., 2024; 2025; Tu et al., 2025), AiOIR aims to restore high-quality images from diverse and potentially unknown degradations within a unified model. A challenge arises from the conflicting optimization objectives inherent to heterogeneous degradations. For example, deblurring requires amplifying high-frequency details, whereas deraining involves suppressing structured streaks. To reconcile this, existing methods generally fall into two paradigms. The first paradigm focuses on modeling degradation heterogeneity through control modulation. Methods such as prompt-learning (Potlapalli et al., 2023; Jiang et al., 2024; Hu et al., 2025; Chen et al., 2025; Rajagopalan et al., 2025) and Mixture-of-Experts (MoEs) (Zhang et al., 2024; Zamfir et al., 2025) selectively activate task-specific weights based on control signals. However, these approaches overlook cross-task commonalities and discretize the continuous degradation space, limiting their representational capacity. Specifically, prompt-based methods often compress spatially varying degradations into compact embeddings, struggling to capture fine-grained corruption patterns. Similarly, MoE architectures rely on discrete routing mechanisms, which are insufficient for modeling continuously varying degradations. Furthermore, these methods are sensitive to imprecise control signals, where erroneous predictions induce incorrect weight activation, resulting in erroneous restoration flows (Fig. 1(a)).

Conversely, the second paradigm emphasizes learning commonality of degradations by reducing the distributional divergence of the inputs. These methods (Zheng et al., 2024; Li et al., 2025) typically rely on fixed schedules to simplify distribution mapping (Fig. 1(a) of fixed schedule). However, heterogeneous degradations exhibit distinct statistical properties. A fixed schedule inevitably compromises adaptability, which is suboptimal for complex corruptions.

To reconcile degradation heterogeneity with restoration commonality, we leverage pixel-wise uncertainty to act as a unified representation for the severity of heterogeneous degradations (Fig. 1(c)) to adaptively steer the restoration process. Specifically, high uncertainty regions require detail restoration, whereas low uncertainty regions need structural

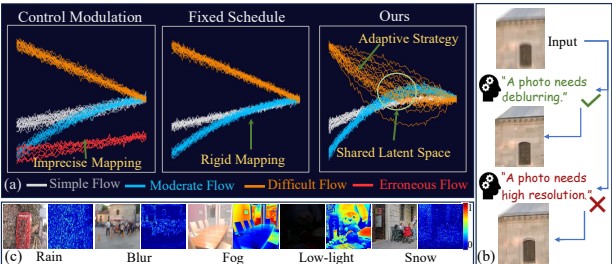

*Figure 1.* Motivation and Analysis. (a) Schematic comparison of different paradigms. Fixed schedule methods employ rigid mappings regardless of degradation severity, while control modulation methods suffer from erroneous flows caused by imprecise conditional. In contrast, our method simultaneously accounts for degradation heterogeneity and shared information. (b) Imposing incorrect prompts in AutoDIR (Jiang et al., 2024) induces artifacts. (c) Uncertainty Visualization. Uncertainty maps (Zhang et al., 2025) naturally capture the severity of diverse degradations.

preservation. We propose an **Uncertainty-Aware Diffusion Bridge Model (UDBM)**, which unifies AiOIR as a stochastic transport problem steered by uncertainty. To model the transport dynamics, UDBM introduces a *relaxed diffusion bridge* that replaces the strict terminal constraint with a relaxed constraint. This relaxation accommodates degradation stochasticity while ensuring Lipschitz continuity of the drift to resolve the drift singularity in standard diffusion bridges. Building upon this, we explicitly decouple the transport dynamics into a deterministic path term and a stochastic noise term. The *noise schedule* constructs a shared high-entropy latent space to align heterogeneous degradation manifolds and facilitates commonality learning. The *path schedule* dynamically regulates the transport trajectory motivated by the viscous dynamics of entropy regularization, allocating dense refinement to high-uncertainty regions while facilitating efficient transition for confident regions. The optimized transport dynamics enables robust AiOIR within a single inference step.

In summary, our main contributions are as follows:

- We propose a relaxed diffusion bridge formulation to explicitly model the inherent uncertainty of heterogeneous degradations in AiOIR while theoretically resolves the drift singularity in the standard diffusion bridge.

- We devise a dual uncertainty-modulation strategy that reconciles the conflict between restoration heterogeneity and commonality. By interpreting the transport process through the lens of entropy-regularized optimal transport, we derive a path schedule that mimics viscous dynamics for adaptive refinement, and a noise schedule that aligns diverse degradation manifolds into a unified high-entropy latent space.

- We enable high-fidelity single-step inference for AiOIR through the joint rectification of transport geometry and dynamics. The proposed method achieves state-of-the-art performance across diverse restoration tasks, outperforming existing diffusion-based methods in both computational efficiency and generalization.

## 2. Related Work

### 2.1. Diffusion Bridge for Image Restoration

Let $\{\mathbf{x}_t\}_{t \in [0,T]}$ be a continuous-time stochastic process, where typically $T = 1$, governed by the Stochastic Differential Equation (SDE). Diffusion bridge is obtained by conditioning the diffusion process on boundary distributions at both endpoints for initial distribution $p_0$ and terminal distribution $p_1$. A classical construction of such a conditioned process is through *Doob's h-transform* (Doob & Doob, 1984; Rogers & Williams, 2000), which modifies the drift of the original SDE to satisfy the boundary marginals:

$$d\mathbf{x}_t = \underbrace{\left[\mathbf{f}(\mathbf{x}_t, t) + g(t)^2 \nabla_x \log h(\mathbf{x}_t, t)\right] dt}_{\text{Drift term}} + \underbrace{g(t)\, d\mathbf{w}_t}_{\text{Diffusion term}},$$
(1)

where $\mathbf{f}$ is the drift function, $g(t)$ is diffusion coefficient, $\mathbf{w}_t$ is Brownian motion, and $h(\mathbf{x}_t, t)$ represents the harmonic function which steers the diffusion towards the target distribution. Building on this, pioneering works like I$^2$SB (Liu et al., 2023), DDBM (Zhou et al., 2023), and IR-SDE (Luo et al., 2023b) construct various diffusion bridges to model the transition probability between degraded and clean image pairs. Meanwhile, IRBridge (Wang et al., 2025) integrates pre-trained generative priors to enhance the perceptual fidelity of the restored images.

However, it is difficult for these standard diffusion bridges to handle the heterogeneity of degradations. Standard diffusion bridges typically employ rigid dynamics that struggle to reconcile diverse degradations. Moreover, their *strict terminal constraints* overlook the inherent uncertainty of degradations, exacerbating drift singularities that destabilize the transport trajectory.

### 2.2. All-in-One Image Restoration

Early AiOIR methods explored task-specific encoders (Li et al., 2022a), contrastive learning (Zhang et al., 2023a), or mask image modeling (Qin et al., 2024) to handle diverse degradation types. To address heterogeneity in degradation, recent approaches predominantly employ control mechanisms. Specifically, prompt-learning frameworks (Potlapalli et al., 2023; Jiang et al., 2024; Ma et al., 2023; Luo et al., 2023a; Cui et al., 2025; Wu et al., 2026; Cui et al., 2026), including those integrated into diffusion models (Luo et al., 2025; Rajagopalan et al., 2025; Hu et al., 2026), rely on explicit descriptors, generative priors, or degradation discrim-

ination to selectively activate task-specific weights, while MoE (Zamfir et al., 2025) utilize discrete routing to modulate the network. However, these mechanisms typically lack the spatial granularity required for scenarios, where degradations are continuous and spatially non-uniform.

Alternatively, recent methods (Zheng et al., 2024; Li et al., 2025) depart from explicit control mechanisms by reducing the distinguishability of heterogeneous inputs to simplify distribution mapping. However, such methods typically utilize a fixed diffusion schedule, which is insufficient for the heterogeneity of degradations. In contrast, the proposed UDBM adaptively regulates transport dynamics to reconcile heterogeneity with commonality. To guide this adaptive regulation, we leverage residual-derived signals. While recent works (Tang et al., 2025; 2026) also exploit residuals to condition optimal transport via a two-pass design or to model degradation-specific residual subspaces, respectively, UDBM interprets the residual as a pixel-wise uncertainty proxy, which explicitly modulates the diffusion process.

## 3. Methodology

In this section, we first formulate AiOIR as a stochastic transport problem. Then we identify two theoretical problems. Finally, we instantiate our theories into the UDBM.

### 3.1. Problem Formulation

Let $\mathbf{x}_0 \equiv \mathbf{x}_{hq} \sim p_{hq}$ denote a clean image sampled from the high-quality image distribution $p_{hq}$. In All-in-One (AiO) scenarios, $\mathbf{x}_0$ is mapped to a degraded observation $\mathbf{x}_{lq} \sim p_{lq}$ via a stochastic transition kernel. Different from task-specific image restoration, the mapping is inherently *stochastic*: a single $\mathbf{x}_0$ can yield diverse degraded realizations $\mathbf{x}_{lq}$ due to heterogeneous and potentially mixed degradation types. To enable AiOIR, we introduce a diffusion bridge $\{\mathbf{x}_t\}_{t \in [0,1]}$, which constructs a transport path between $p_{hq}$ and $p_{lq}$. Given the intrinsic stochasticity in AiO, we argue that the transport dynamics should be steered by the pixel-wise uncertainty $\mathbf{u}$ (detailed in Appendix G).

### 3.2. Relaxed Diffusion Bridge

Due to the stochastic degradation processes addressed by AiOIR, the observed $\mathbf{x}_{lq}$ is subject to uncertainty, which reflects the severity of degradation. However, standard diffusion bridges typically impose a Dirac constraint $\delta(\cdot)$ on the terminal state, i.e., $p(\mathbf{x}_1) = \delta(\mathbf{x}_1 - \mathbf{x}_{lq})$. Such a strict terminal constraint ignores this stochasticity and mathematically induces a *drift singularity* as $t \to 1$. To analyze this behavior rigorously, we first state the conditions for the underlying diffusion process.

**Assumption 3.1.** We assume the drift function $\mathbf{f}(\mathbf{x}, t)$ in Eq. (1) is Lipschitz continuous with respect to $\mathbf{x}$ and

bounded for all $t \in [0, 1]$, and the diffusion coefficient $g(t)$ is strictly positive and bounded.

Under Assumption 3.1, the singularity arises from the terminal constraint (details refer to Appendix A):

**Proposition 3.2.** *For a strict terminal constraint $p(\mathbf{x}_1) = \delta(\mathbf{x}_1 - \mathbf{x}_{lq})$ in standard diffusion bridge, the harmonic function $h(\mathbf{x}_t, t)$ in Eq. (1) is instantiated as $h_{strict}(\mathbf{x}_t, t) \triangleq p(\mathbf{x}_1 = \mathbf{x}_{lq} \mid \mathbf{x}_t)$. Under this setting, the expected magnitude of the score function $\|\mathbf{x}_{lq} - \mathbf{x}_t\|$ scales as $\mathcal{O}(\sqrt{1-t})$, while the time denominator vanishes as $\mathcal{O}(1 - t)$. This convergence mismatch causes the drift term to diverge as $t \to 1$:*

$$\left\| g(t)^2 \nabla_{\mathbf{x}_t} \log h_{strict}(\mathbf{x}_t, t) \right\| \sim \left\| \frac{\mathbf{x}_{lq} - \mathbf{x}_t}{1 - t} \right\| = O\left( \frac{1}{\sqrt{1-t}} \right). \tag{2}$$

Approximating this unbounded drift forces the network to fit divergent targets, resulting in over-smoothed results and limited generalization. While prior works (Liu et al., 2023; Zhou et al., 2023) employ heuristics to mitigate instability, they struggle to resolve the inherent singularity. Crucially, this singularity exacerbates optimization conflicts in AiOIR: the stochastic deviation $\|\mathbf{x}_{lq} - \mathbf{x}_t\|$ varies significantly across heterogeneous degradations. This inter-task discrepancy is further amplified by the singularity, making the AiO mapping difficult for a single network to learn.

To explicitly model the stochasticity in AiO while addressing the singularity, we propose a *relaxed diffusion bridge* with a relaxed terminal constraint. Specifically, we formulate $\mathbf{x}_1$ as a Gaussian distribution centered at $\mathbf{x}_{lq}$:

$$p(\mathbf{x}_1) = \mathcal{N}(\mathbf{x}_1; \mathbf{x}_{lq}, \boldsymbol{\sigma}_u^2), \tag{3}$$

where $\boldsymbol{\sigma}_u > \mathbf{0}$ is set proportional to $\mathbf{u}$. Eq. (3) allows the diffusion bridge to terminate within a plausible neighborhood defined by uncertainty, naturally preserving the inherently stochastic in AiOIR. The formulation further addresses the singularity as illustrated in Theorem 3.3.

**Theorem 3.3.** *Under relaxed terminal constraint in Eq. (3), the introduction of $\boldsymbol{\sigma}_u^2$ regularizes the singularity as $t \to 1$:*

$$\left\| g(t)^2 \nabla_{\mathbf{x}_t} \log h_{relaxed}(\mathbf{x}_t, t) \right\| \sim \left\| \frac{\mathbf{x}_{lq} - \mathbf{x}_t}{(1 - t) + \sigma_{\min}^2} \right\| = \mathcal{O}(1), \tag{4}$$

*where $\sigma_{\min}^2 \triangleq \min(\boldsymbol{\sigma}_u^2)$ denotes the minimum scalar value in the variance $\boldsymbol{\sigma}_u^2$. The relaxed terminal constraint ensures Lipschitz continuity of the drift term at the boundary $t = 1$, providing a stable transport dynamics (details refer to Appendix A).*

### 3.3. Uncertainty-Aware Entropic Regularization

Since $p_{lq}$ in AiOIR represents a mixture of heterogeneous manifolds, the probability distributions of distinct degra-

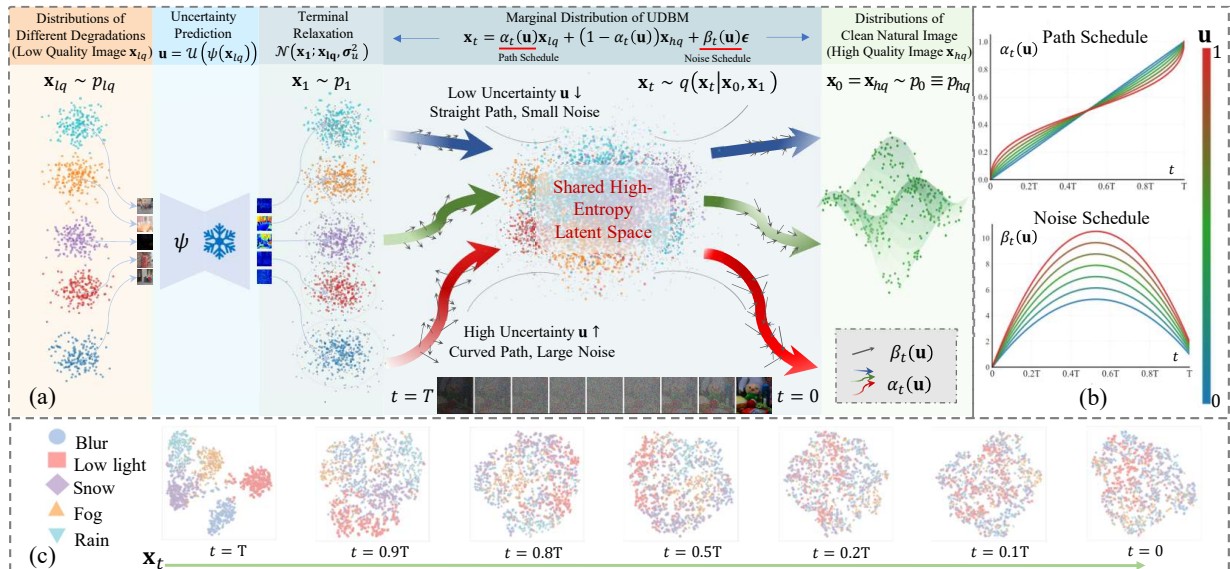

*Figure 2.* Overview of the UDBM. (a) The schematic diagram of marginal distribution of UDBM. (b) The dual uncertainty-guided path and noise schedules. (c) t-SNE visualization of $\mathbf{x}_t$ of different degradations, showing the gradually alignment of diverse degradations in the shared high-entropy latent space.

dations overlap significantly during transport to the shared clean manifold. This overlap induces a state of high uncertainty, forming an *entropic barrier*. This barrier peaks at $t \approx 0.5$, where the state $\mathbf{x}_t$ exhibits a maximum confounding of degradation artifacts and clean semantics, which hinder the effective learning of transport dynamics.

To mitigate this barrier, we adopt *Entropy-Regularized Optimal Transport* (EOT). We model the transport dynamics as a Schrödinger Bridge problem constrained by an uncertainty-adaptive viscosity coefficient $\varepsilon(\mathbf{u})$:

$$\min_{\rho,\mathbf{v}} \int_0^1 \int_{\mathcal{X}} \frac{1}{2}\rho_t(\mathbf{x})\|\mathbf{v}_t(\mathbf{x})\|^2 \, d\mathbf{x} \, dt, \tag{5}$$
$$\text{s.t.} \quad \partial_t\rho_t + \nabla \cdot (\rho_t \mathbf{v}_t) = \varepsilon(\mathbf{u})\Delta\rho_t,$$

where $\rho_t$ is the probability density, $\mathbf{v}_t$ the velocity field, and $\nabla, \Delta$ denote the gradient and Laplacian operators. The optimality conditions for this problem are given by the Proposition 3.4 (details provided in Appendix B).

**Proposition 3.4.** *The optimal velocity field $\mathbf{v}_t^*$ for Eq. (5) is determined by the gradient of potential $\phi_t$, i.e., $\mathbf{v}_t^* = -\nabla\phi_t$. The evolution of $\phi_t$ is governed by the backward Viscous Hamilton–Jacobi–Bellman equation(Léonard, 2013):*

$$\partial_t\phi_t + \varepsilon(\mathbf{u})\Delta\phi_t = \frac{1}{2}\|\nabla\phi_t\|^2. \tag{6}$$

Proposition 3.4 reveals a kinetic mismatch in fixed linear schedules, as they enforce constant or fixed velocity transport even through entropy barriers (analyzed in Remark 3.5).

Such rigid dynamics fundamentally limit the adaptability required to reconcile the heterogeneity inherent in AiOIR. *Remark* 3.5. The Laplacian term $\varepsilon(\mathbf{u})\Delta\phi_t$ in Eq. (6) acts as a geometric smoothing operator, flattening the potential and suppressing drift $\|\mathbf{v}_t^*\|$ under high uncertainty regions. This effect dominates in the entropic barrier, where diffusion prevails. The transport dynamics require deceleration in the intermediate state for exploration, which necessitates compensatory acceleration near the boundaries to meet terminal constraints.

### 3.4. The UDBM Framework

In this subsection, we instantiate the relaxed diffusion bridge and uncertainty-aware entropic regularization into the UDBM. We adopt a residual-based proxy as pixel-wise uncertainty $\mathbf{u}$ for efficiency (Zhang et al., 2025). Specifically, the absolute residual between the degraded input and a preliminary estimation serves as an indicator of restoration difficulty:

$$\mathbf{u} = \mathcal{U}_{\text{res}}(\psi(\mathbf{x}_{lq})) = \frac{1}{2}|\psi(\mathbf{x}_{lq}) - \mathbf{x}_{lq}|, \tag{7}$$

where $\psi(\cdot)$ is as a lightweight restoration network. As illustrated in Fig. 2, the framework operates by first predicting this uncertainty $\mathbf{u}$ from $\mathbf{x}_{lq}$ using a frozen auxiliary network $\psi(\cdot)$ and constructing a relaxed terminal constraint $p(\mathbf{x}_1) = \mathcal{N}(\mathbf{x}_1; \mathbf{x}_{lq}, \boldsymbol{\sigma}_u^2)$. Subsequently, $\mathbf{u}$ adaptively steers the transport dynamics from $\mathbf{x}_1$ to $\mathbf{x}_0$ that the noise schedule constructs a shared latent space to align heterogeneous distributions, while the path schedule regulates the transport

trajectory.

**Marginal Distribution-based Formulation.** While the diffusion bridges are theoretically grounded in Doob's $h$-transform, direct SDE simulation is computationally expensive. Moreover, the relaxed diffusion bridge formulation complicates the explicit derivation of the drift. To address these challenges, we adopt a *marginal distribution* perspective. Leveraging the property that linear SDEs yield Gaussian marginals, the conditional distribution of $\mathbf{x}_t$ can be characterized as $q(\mathbf{x}_t \mid \mathbf{x}_{lq}, \mathbf{x}_{hq}) = \mathcal{N}(\boldsymbol{\mu}_t, \sigma_t^2 \mathbf{I})$. Thus, we can formulate the transition as:

$$\mathbf{x}_t = \underbrace{\alpha_t \mathbf{x}_{lq} + \gamma_t \mathbf{x}_{hq}}_{\text{Path Term}} + \underbrace{\beta_t \boldsymbol{\epsilon}}_{\text{Noise Term}}, \quad \boldsymbol{\epsilon} \sim \mathcal{N}(\mathbf{0}, \mathbf{I}). \quad (8)$$

The *path schedule* $(\alpha_t, \gamma_t)$ governs the deterministic path term, while the *noise schedule* $(\beta_t)$ steers the stochastic noise term. This formulation further unifies prior specific approaches (proven in Appendix C), providing a universal and flexible design for AiOIR.

Following previous works (Li et al., 2023; Liu et al., 2023), by analytically deriving the reverse transition of Eq. (8), we obtain a deterministic update rule analogous to DDIM (comprehensive derivations, including the stochastic DDPM counterpart, are provided in Appendix D):

$$\mathbf{x}_s = \underbrace{\alpha_s \mathbf{x}_{lq} + \gamma_s \hat{\mathbf{x}}_{0|t}}_{\text{Target Mean}} + \underbrace{\frac{\beta_s}{\beta_t} \left( \mathbf{x}_t - \alpha_t \mathbf{x}_{lq} - \gamma_t \hat{\mathbf{x}}_{0|t} \right)}_{\text{Deterministic Direction}}, \quad (9)$$

where $s < t$ represents the previous time step in the discretization schedule and $\hat{\mathbf{x}}_{0|t}$ denotes the clean data predicted at step $t$. To enable pixel-wise modulation of transport dynamics, UDBM extends Eq. (8) by parameterizing the schedules as spatially element-wise operations steered by uncertainty: $\alpha_t \to \alpha_t(\mathbf{u})$, $\gamma_t \to \gamma_t(\mathbf{u})$, and $\beta_t \to \beta_t(\mathbf{u})$.

**Uncertainty-Aware Noise Scheduling.** To instantiate the noise coefficient $\beta_t$ in Eq. (8), we decompose $\beta_t(\mathbf{u})$ into a superposition of a shared bridge component and a terminal relaxation component:

$$\beta_t(\mathbf{u}) = \underbrace{\eta_{\text{bridge}}(\mathbf{u}) t(1-t)}_{\text{Shared Bridge Term}} + \underbrace{\eta_{\text{relax}}(\mathbf{u}) t^2}_{\text{Terminal Relaxation Term}}, \quad (10)$$

where $\eta_{\text{bridge}}(\mathbf{u}) = \lambda_b(\mathbf{I} + \mathbf{u})$ and $\eta_{\text{relax}}(\mathbf{u}) = \mathbf{I} + \mathbf{u}$. The design is grounded in two theoretical motivations:

*1) Shared Manifold Alignment:* The Shared Bridge Term adopts the characteristic convex $t(1-t)$ profile inspired by Brownian Bridge (Li et al., 2023) while amplifies its magnitude via $\eta_{\text{bridge}}(\mathbf{u})$. The high-intensity noise suppresses the statistical discrepancies of heterogeneous degradations by lowering the signal-to-noise ratio (SNR) of degradation-specific features, thereby forcing divergent degradation manifolds to gradually align within a *shared high-entropy latent*

*space* as illustrated in Fig. 2(c). Crucially, $\eta_{\text{bridge}}(\mathbf{u})$ enables spatially adaptive exploration that encourages aggressive stochastic search in high-uncertainty regions to resolve complex degradations, while maintaining lower variance in confident areas to preserve structural fidelity.

*2) Uncertainty Accumulation:* The Terminal Relaxation Term is designed to achieve relaxed terminal constraint in Theorem 3.3, where $t^2$ models the accumulation of uncertainty. It enforces the diffusion coefficient to monotonically increase from the deterministic clean state to the probabilistic degraded state with $\beta_1(\mathbf{u}) \equiv \boldsymbol{\sigma}_u = \mathbf{I} + \mathbf{u}$ in Eq. (3).

**Uncertainty-Adaptive Path Schedule.** To incorporate the viscous dynamics inherent in EOT while bypassing the numerical solution of the intractable coupled system of Partial Differential Equations (PDEs), we propose a parametric approximation. By constructing a path schedule $\alpha_t(\mathbf{u})$ that explicitly satisfies the kinetic constraints derived in Proposition 3.4, we define the mean trajectory $\boldsymbol{\mu}_t$ to follow a geometrically linear path (geodesic) governed by an uncertainty-adaptive time reparameterization:

$$\boldsymbol{\mu}_t = (\mathbf{I} - \alpha_t(\mathbf{u}))\mathbf{x}_{hq} + \alpha_t(\mathbf{u})\mathbf{x}_{lq}, \quad (11a)$$

$$\alpha_t(\mathbf{u}) = \frac{t^{\pi(\mathbf{u})}}{t^{\pi(\mathbf{u})} + (1-t)^{\pi(\mathbf{u})}}, \quad (11b)$$

$$\pi(\mathbf{u}) = (1-\mathbf{u})\pi_{\text{OT}} + \mathbf{u}\pi_{\text{EOT}}, \quad (11c)$$

where $\pi_{\text{OT}} = 1.0$ (Optimal Transport) and $\pi_{\text{EOT}} = 0.5$. This parameterization naturally satisfies the constraints prescribed by Remark 3.5 (details in Appendix E).

**Feasibility of Single-Step Inference.** UDBM enables high-fidelity single-step inference by jointly regularizing terminal dynamics and rectifying transport geometry. As established in Theorem 3.3, the Relaxed Diffusion Bridge ensures bounded mapping coefficients as $t \to 1$, inducing a Lipschitz continuous drift at the terminal. Simultaneously, the linear mean trajectory in Eq. (11a) constrains the flow to a straight geodesic. Theoretically, this guarantees an optimal transport direction and provides a consistent regression target, enabling effective single-step discretization even with variable speed schedules. See Appendix F for details.

## 4. Experiments

We evaluate UDBM under three settings: (a) AiO, training on mixed degradations across five tasks (Zheng et al., 2024); (b) Task-specific, training models for task-specific degradations (Li et al., 2022a); and (c) Generalization, applying the AiO model to real-world and unseen composite benchmarks. Evaluation employs PSNR (dB) (**P**)/SSIM (**S**) for fidelity, and MANIQA (Yang et al., 2022), LIQE (Zhang et al., 2023b), and MUSIQ (Ke et al., 2021) for perceptual

*Table 1.* Quantitative comparison in AiOIR. The best, second-best, and third-best results are highlighted in red, blue, and green respectively. **#P** denotes to parameters. FLOPs (**#F**) are computed on $256 \times 256$ input size, while Runtime (**#R**) is measured on $512 \times 512$ size. The parameter counts, FLOPs, and runtime include the computational overhead of the auxiliary uncertainty-estimation network $\psi(\cdot)$. To ensure a fair comparison, all methods are trained on the same dataset (Zheng et al., 2024).

| Method | Rain (5 sets) | | Low-light | | Snow (2 sets) | | Haze | | Blur | | Average | | Complexity | | |
|---|---|---|---|---|---|---|---|---|---|---|---|---|---|---|---|
| | P↑ | S↑ | P↑ | S↑ | P↑ | S↑ | P↑ | S↑ | P↑ | S↑ | P↑ | S↑ | #P (M)↓ | #F (G)↓ | #R (ms)↓ |
| *Task-Specific Settings* | | | | | | | | | | | | | | | |
| SwinIR[ICCV'21] | 30.78 | .923 | 17.81 | .723 | - | - | 21.50 | .891 | 24.52 | .773 | - | - | **0.90** | 752.13 | 1273.55 |
| MIRNet-v2[TPAMI'22] | 33.89 | .924 | 24.74 | .851 | - | - | 24.03 | .927 | 26.30 | .799 | - | - | 5.90 | **140.92** | **113.47** |
| Restormer[CVPR'22] | 33.96 | .935 | 20.41 | .806 | - | - | 30.87 | .969 | 32.92 | .961 | - | - | 26.10 | 141.00 | 229.13 |
| IR-SDE[ICML'23] | - | - | - | - | 20.45 | .787 | - | - | 30.70 | .901 | - | - | 34.20 | - | 16781.94 |
| RDDM[CVPR'24] | 30.74 | .903 | 23.22 | .899 | 32.55 | .927 | 30.78 | .953 | 29.53 | .876 | 29.36 | .912 | 36.26 | 164.68 | 115.99 |
| *All-in-one Settings* | | | | | | | | | | | | | | | |
| Restormer[CVPR'22] | 27.10 | .843 | 17.63 | .542 | 28.61 | .876 | 22.79 | .706 | 26.36 | .814 | 24.50 | .756 | 26.12 | 141.00 | 229.13 |
| AirNet[CVPR'22] | 24.87 | .773 | 14.83 | .767 | 27.63 | .860 | 25.47 | .923 | 26.92 | .811 | 23.94 | .827 | 5.77 | 301.27 | 330.44 |
| Prompt-IR[NeurIPS'23] | 29.56 | .888 | 22.89 | .847 | 31.98 | .924 | 32.02 | .952 | 27.21 | .817 | 28.73 | .886 | 32.97 | 158.14 | 242.98 |
| DA-CLIP[ICLR'24] | 28.96 | .853 | 24.17 | .882 | 30.80 | .888 | 31.39 | .983 | 25.39 | .805 | 28.14 | .882 | 174.10 | 118.50 | 73.70 |
| DiffUIR[CVPR'24] | 31.03 | .904 | 25.12 | .907 | 32.65 | .927 | 32.94 | .956 | 29.17 | .864 | 30.18 | .912 | 36.26 | 98.81 | 348.88 |
| AdaIR[ICLR'25] | 31.42 | .910 | 23.48 | .900 | 32.89 | .937 | 28.06 | .980 | 29.92 | .882 | 29.15 | .922 | 28.73 | 147.18 | 248.97 |
| BioIR[NeurIPS'25] | 31.20 | .907 | 23.88 | .900 | 32.80 | .937 | 29.34 | .981 | 29.39 | .870 | 29.32 | .919 | 15.8 | 125.61 | 202.18 |
| MOCE-IR[CVPR'25] | 30.46 | .903 | 25.87 | .907 | 31.89 | .926 | 30.98 | .987 | 28.57 | .852 | 29.55 | .915 | 25.35 | 87.55 | 198.78 |
| HOGformer[AAAI'26] | 31.63 | .914 | 25.57 | .917 | 34.08 | .941 | 36.60 | .994 | 29.95 | .884 | 31.57 | .930 | 16.64 | 91.77 | 366.37 |
| **UDBM-S**[ours] | 30.83 | .901 | 25.40 | .900 | 32.48 | .930 | 36.23 | .994 | 29.11 | .861 | 30.81 | .917 | **5.71** | **5.14** | **56.32** |
| **UDBM-M**[ours] | 31.51 | .911 | 25.59 | .906 | 33.17 | .936 | 37.97 | .995 | 29.74 | .876 | 31.60 | .925 | 14.14 | 11.98 | 58.21 |
| **UDBM-L**[ours] | 32.06 | .917 | 26.55 | .915 | 34.00 | .943 | 39.88 | .996 | 30.58 | .895 | 32.61 | .933 | 56.02 | 46.85 | 58.49 |

quality of generalization benchmarks. Due to page limits, more experiment results are shown in Appendix J.

**Implementation Details.** We implement UDBM in PyTorch and train on two NVIDIA RTX 4090 GPUs. Optimization is performed using Adam (lr $= 8 \times 10^{-5}$) minimizing the $L_1$ Loss. Following previous settings (Zheng et al., 2024; Wu et al., 2026), we set the crop size to $256 \times 256$. Training configurations are as follows: Setting (a) runs for 600K iterations with a batch size of 40, while Setting (b) runs for 300K iterations with a batch size of 8. Regarding the architecture, we adopt a modified UNet (Chen et al., 2022; Luo et al., 2023c) as the denoising backbone and a lightweight version for the $\psi(\cdot)$. Details of the uncertainty prediction are provided in Appendix G. We utilized only one step for inference in all tasks. For more details, please refer to Appendix I.2.

### 4.1. Setting (a): All-in-One

**Datasets.** We utilize standard benchmarks across five image restoration tasks (Zheng et al., 2024): the merged datasets (Jiang et al., 2020; Wang et al., 2022b) for deraining, LOL (Wei et al., 2018) for low-light image enhancement, Snow100K (Liu et al., 2018) for desnowing, RESIDE (Li et al., 2018) for dehazing, and GoPro (Nah et al., 2017) for deblurring (detailed in Appendix I.3). For comprehensive evaluation, we provide three versions of UDBM (Smal (S), Medium (M), Large (L)), corresponding to different parameters (details are provided in Appendix I.1).

**Competing methods.** We compare the proposed UDBM against a set of state-of-the-art (SOTA) methods, including task-specific baselines (SwinIR (Liang et al., 2021), MIRNet-v2 (Zamir et al., 2022b), Restormer (Zamir et al., 2022a), IR-SDE (Luo et al., 2023b), RDDM (Liu et al., 2024)) and AiO frameworks (AirNet(Li et al., 2022a), Prompt-IR (Potlapalli et al., 2023), DA-CLIP (Luo et al., 2023a), DiffUIR (Zheng et al., 2024), AdaIR(Cui et al., 2025), BioIR (Cui et al., 2026), MOCE-IR (Zamfir et al., 2025), HOGformer (Wu et al., 2026)).

**Results and Analysis.** As shown in Tab. 1, our UDBM-L achieves the best overall performance. Remarkably, UDBM-L requires approximately $2\times$ fewer FLOPs than HOGformer and achieves a $6\times$ inference speedup over both HOGformer and DiffUIR. The efficiency is attributed to the single-step inference and the lightweight backbone of UDBM. Meanwhile, the flexible transport dynamics and aligning heterogeneous manifolds mitigate multi-task conflicts in UDBM. As shown in Fig. 3, our UDBM-L outperforms others in restoring fine details (Appendix J.4 for more results). Even our medium variant, UDBM-M, outperforms the second-best HOGformer (31.60 dB vs. 31.57 dB) while using fewer parameters and significantly lower computational cost (11.98G Flops vs. 91.77G Flops). Furthermore, the lightweight variant UDBM-S demonstrates a trade-off between performance and efficiency, delivering competitive results with minimal overhead (5.14 GFLOPs, 5.71M parameters).

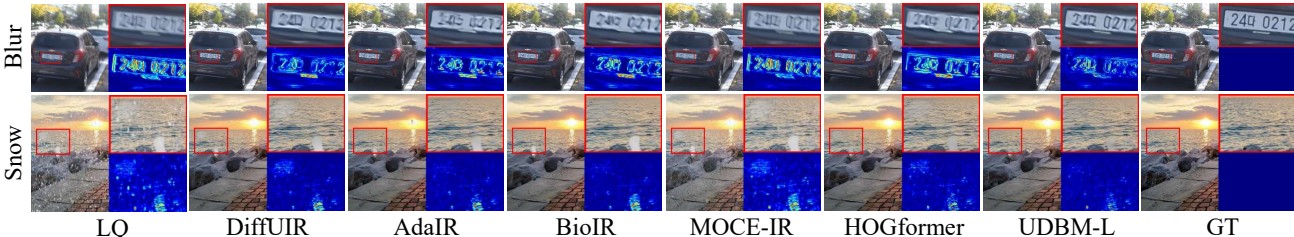

*Figure 3.* Visual comparison of AiOIR. Our UDBM exhibits better artifacts removal and details restoration. Bottom-right: error maps.

*Table 2.* Quantitative comparison on BSD68 dataset.

| Method | $\sigma = 15$ P↑ / S↑ | $\sigma = 25$ P↑ / S↑ | $\sigma = 50$ P↑ / S↑ | Average P↑ / S↑ |
|---|---|---|---|---|
| CBM3D | 33.50 / .922 | 30.69 / .868 | 27.36 / .763 | 30.52 / .851 |
| DnCNN | 33.89 / .930 | 31.23 / .883 | 27.92 / .789 | 31.01 / .867 |
| IRCNN | 33.87 / .929 | 31.18 / .882 | 27.88 / .790 | 30.98 / .867 |
| FFDNet | 33.87 / .929 | 31.21 / .882 | 27.96 / .789 | 31.01 / .867 |
| BRDNet | 34.10 / .929 | 31.43 / .885 | 28.16 / .794 | 31.23 / .869 |
| AirNet | 34.14 / .936 | 31.48 / .893 | 28.23 / .806 | 31.28 / .878 |
| BioIR | 34.15 / .936 | 31.57 / .894 | 28.29 / .808 | 31.34 / .879 |
| **UDBM-L** | **34.23 / .937** | **31.60 / .895** | **28.38 / .810** | **31.40 / .881** |

*Table 3.* Quantitative comparison on GoPro dataset.

| Methods | SAM-Deblur | Restormer | UFormer | BioIR | NAFNet | **UDBM-L** |
|---|---|---|---|---|---|---|
| P↑ | 32.83 | 32.92 | 32.97 | 33.28 | 33.69 | **33.87** |
| S↑ | 0.960 | 0.961 | 0.967 | 0.966 | 0.967 | **0.968** |

### 4.2. Setting (b): Task-Specific

**Denoising.** For image denoising, we evaluate UDBM with task-specific methods (CBM3D (Dabov et al., 2007), DnCNN (Zhang et al., 2017a), IRCNN (Zhang et al., 2017b), FFDNet (Zhang et al., 2018), BRDNet (Tian et al., 2020)), and AiO methods (AirNet and BioIR). All methods are trained on the combined BSD600 (Arbelaez et al., 2010) and WED (Ma et al., 2016) datasets, while evaluated on BSD68 (Martin et al., 2001) under Gaussian noise levels $\sigma \in \{15, 25, 50\}$. As reported in Tab. 2, our methods outperform the recent AiO method BioIR by 0.06dB in PSNR while using only 37.3% of its computational cost.

**Deblurring.** As reported in Table 3, we compare UDBM-L on the GoPro dataset against task-specific methods (Restormer (Zamir et al., 2022a), UFormer (Wang et al., 2022a), NAFNet (Chen et al., 2022), SAM-Deblur (Li et al., 2024)) and the AiO method (BioIR). Our method achieves the best results across all metrics.

### 4.3. Setting (c): Generalization

In this section, we evaluate models with pre-trained weights from Setting (a) to assess generalization.

**Real-world scenarios.** To verify the generalization capa-

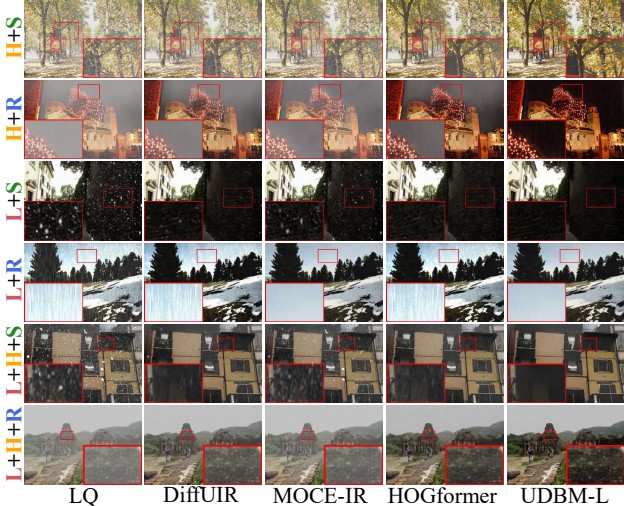

*Figure 4.* Visual comparison of unseen scenarios on the CDD11 dataset. The proposed UDBM shows better degradation removal compared to other methods.

bility on known tasks in real-world scenarios, we conduct evaluations on multiple benchmarks: Practical (Yang et al., 2017) for deraining; MEF (Ma et al., 2015), NPE (Wang et al., 2013), and DICM (Lee et al., 2013) for low-light enhancement; real-world snow images in Snow-100K for desnowing; as well as HIDE (Shen et al., 2019) and Real-Blur (Rim et al., 2020) for deblurring. Quantitative results in Tab. 4 demonstrate that UDBM-L outperforms competing methods across all degradation types.

**Unseen scenarios.** To assess the zero-shot robustness of UDBM against complex unseen scenarios, we conduct experiments on the CDD11 benchmark (Guo et al., 2024), which encompasses low-light (L), haze (H), rain (R), snow (S), and their combinations. As evidenced in Tab. 5, UDBM-L exhibits superior perceptual metrics. As visually demonstrated in Fig. 4, our method demonstrates effective generalization to challenging unseen double and triple degradation scenarios, while other methods suffer from severe degradation residues and noticeable artifacts.

*Table 4.* Quantitative comparison on Real Rain, Real Dark, Real Snow, and Real Blur datasets.

| Method | Real Rain | | | Real Dark | | | Real Snow | | | Real Blur | |
|---|---|---|---|---|---|---|---|---|---|---|---|
| | MANIQA↑ | LIQE↑ | MUSIQ↑ | MANIQA↑ | LIQE↑ | MUSIQ↑ | MANIQA↑ | LIQE↑ | MUSIQ↑ | P↑ | S↑ |
| Prompt-IR | 0.356 | 3.133 | 60.01 | 0.351 | 3.012 | 58.31 | 0.392 | 3.146 | 60.83 | 22.48 | 0.770 |
| DiffUIR | 0.366 | 3.210 | 60.55 | 0.386 | 3.129 | 63.29 | 0.392 | 3.190 | 61.11 | 30.63 | 0.890 |
| AdaIR | 0.372 | 3.230 | 60.10 | 0.348 | 3.136 | 61.74 | 0.393 | 3.163 | 61.27 | 30.17 | 0.843 |
| BioIR | 0.370 | 3.246 | 60.30 | 0.384 | 3.046 | 62.14 | 0.391 | 3.201 | 61.02 | 30.32 | 0.864 |
| MOCE-IR | 0.371 | 3.279 | 60.61 | 0.375 | 3.215 | 62.12 | 0.392 | 3.204 | 61.27 | 29.59 | 0.856 |
| HOGformer | 0.374 | 3.284 | 60.37 | 0.367 | 3.296 | 59.91 | 0.396 | 3.219 | 61.28 | 29.42 | 0.825 |
| **UDBM-L** | **0.380** | **3.355** | **60.86** | **0.405** | **3.347** | **63.67** | **0.397** | **3.254** | **61.43** | **31.14** | **0.903** |

*Table 5.* Quantitative comparison on the CDD11 dataset. We report LIQE metrics (higher is better) for evaluation.

| Method | Single Degradation | | | | Double Degradations | | | | | Triple Degradations | | Average |
|---|---|---|---|---|---|---|---|---|---|---|---|---|
| | L | H | R | S | L+H | L+R | L+S | H+R | H+S | L+H+R | L+H+S | |
| DiffUIR | 1.935 | 4.256 | 4.297 | 4.347 | 1.583 | 1.817 | 1.675 | 3.389 | 3.247 | 1.630 | 1.486 | 2.697 |
| AdaIR | 1.559 | 4.178 | 4.230 | 4.317 | 1.542 | 1.752 | 1.582 | 3.242 | 3.351 | 1.608 | 1.459 | 2.620 |
| BioIR | 1.718 | 4.258 | 4.345 | 4.405 | 1.561 | 1.783 | 1.612 | 3.378 | 3.328 | 1.617 | 1.471 | 2.680 |
| MOCE-IR | 1.745 | 4.314 | 4.247 | 4.358 | 1.579 | 1.760 | 1.678 | 3.397 | 3.283 | 1.602 | 1.479 | 2.677 |
| HOGformer | 1.478 | 4.396 | 4.274 | 4.417 | 1.551 | 1.700 | 1.591 | 3.409 | 3.307 | 1.569 | 1.472 | 2.651 |
| **UDBM-L** | **2.083** | **4.454** | **4.445** | **4.532** | **1.587** | **2.161** | **1.929** | **3.801** | **3.589** | **1.783** | **1.514** | **2.898** |

*Table 6.* Ablation study in AiOIR.

| Component | Method | Rain P↑ / S↑ | Low-light P↑ / S↑ | Snow P↑ / S↑ | Haze P↑ / S↑ | Blur P↑ / S↑ | Average P↑ / S↑ |
|---|---|---|---|---|---|---|---|
| **Path Schedule** (Eq. (11a)) | w/ $\pi(\mathbf{u}) = \mathbf{I}$ | 31.89 / .916 | 26.43 / .914 | 33.78 / .942 | 39.12 / .995 | 30.31 / .893 | 32.31 / .932 |
| | w/ $\pi(\mathbf{u}) = 0.5\mathbf{I}$ | 31.79 / .915 | 25.89 / .913 | 33.82 / .941 | 38.29 / .995 | 30.29 / .890 | 32.02 / .931 |
| **Shared Bridge Term** (Eq. (10)) | w/ $\eta_{\text{bridge}}(\mathbf{u}) = 1.5\lambda_b\mathbf{I}$ | 31.62 / .912 | 25.78 / .910 | 33.29 / .938 | 37.19 / .994 | 30.21 / .892 | 31.62 / .929 |
| **Terminal Relaxation Term** (Eq. (10)) | w/ $\eta_{\text{relax}}(\mathbf{u}) = \mathbf{0}$ | 31.07 / .903 | 24.45 / .901 | 32.87 / .929 | 34.98 / .987 | 30.23 / .892 | 30.72 / .922 |
| | w/ $\eta_{\text{relax}}(\mathbf{u}) = 1.5\mathbf{I}$ | 31.67 / .912 | 25.93 / .913 | 33.62 / .940 | 38.95 / **.996** | 30.45 / .893 | 32.12 / .931 |
| **Uncertainty Estimator** ($\mathcal{U}(\cdot)$) | w/ BayesCap | 31.74 / .912 | 25.86 / .913 | 33.79 / .941 | **40.38** / **.996** | 30.53 / **.895** | 32.46 / .931 |
| | w/ heteroscedastic regression | 31.83 / .913 | 25.42 / .911 | 33.83 / .941 | 38.76 / .994 | 30.28 / .894 | 32.04 / .931 |
| **UDBM-L** | | **32.06 / .917** | **26.55 / .915** | **34.00 / .943** | 39.88 / **.996** | **30.58 / .895** | **32.61 / .933** |

## 4.4. Ablation Study

We validate the proposed components on the AiO benchmark reported in Tab. 6 (due to space limitations, additional ablation results are provided in Appendix J).

**Uncertainty-aware Path Schedule.** The OT schedule ($\pi(\mathbf{u}) = \mathbf{I}$ in Eq. (11a)) ultimately yields suboptimal results due to insufficient refinement steps in high-uncertainty areas. Conversely, $\pi(\mathbf{u}) = 0.5\mathbf{I}$ tends to over-process low-uncertainty regions, where excessive refinement disrupts inherent structural fidelity. Fig. 5(a) illustrates the impact of $\pi_{\text{EOT}}$. As $\pi_{\text{EOT}}$ increases, the trajectory defined by Eq. (11a) converges towards the OT geodesic. Conversely, excessively small values induce abrupt velocity transitions that hinder effective learning.

**Shared Bridge Term.** Replacing the adaptive coefficient in Eq. (10) with a constant ($\eta_{\text{bridge}}(\mathbf{u}) = 1.5\lambda_b\mathbf{I}$) leads to a performance drop, confirming that fixed terminal variance struggles to accommodate heterogeneous degradations, resulting in either insufficient exploration or excessive perturbation. Sensitivity analysis of $\lambda_b$ further illustrated in

Fig. 5(b) reveals that extremely low $\lambda_b$ values fail to align degradation into the shared high-entropy latent space, while the high $\lambda_b$ values may corrupt the structural information from $\mathbf{x}_{lq}$, resulting in a performance drop.

To verify manifold alignment, we visualize the evolution of intermediate features via t-SNE and Silhouette Coefficient (SC) (Rousseeuw, 1987) in Fig. 6. As $t \to 0$, the converging clusters and decreasing SC confirm that our Shared Bridge Term effectively projects heterogeneous degradations into a unified, shared high-entropy latent space.

**Terminal Relaxation Term.** Removing the Terminal Relaxation Term $\eta_{\text{relax}}$ in Eq. (10) significantly degrades performance, validating that drift singularity at the deterministic endpoint destabilizes transport dynamics. A fixed relaxation ($\eta_{\text{relax}}(\mathbf{u}) = 1.5\mathbf{I}$), while avoiding singularity, remains suboptimal as it overlooks the intrinsic uncertainty of the inputs.

**Uncertainty Estimation.** To validate extensibility, we further evaluate alternative uncertainty estimators including BayesCap (Upadhyay et al., 2022) and heteroscedastic regression (Kendall & Gal, 2017). While BayesCap is effective, it incurs a significant 43% runtime overhead. Het-

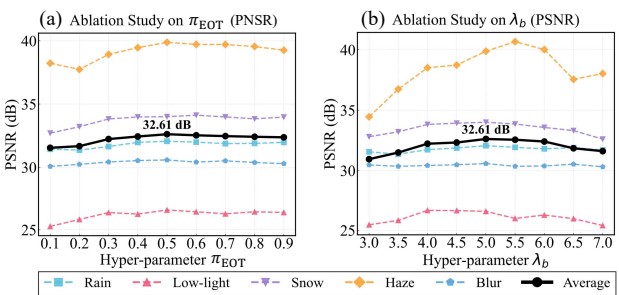

*Figure 5.* Ablation studies on hyper-parameter $\pi_{SB}$ (a) and $\lambda_b$ (b).

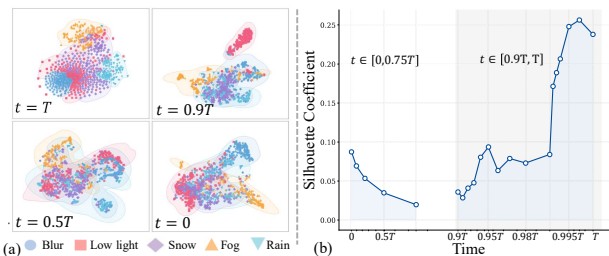

*Figure 6.* Validation of manifold alignment. (a) t-SNE visualization illustrates that degradation-specific features gradually merge into a shared distribution as $t$ decreases. (b) The SC curve of t-SNE quantitatively confirms the alignment.

*Table 7.* Controlled ablations studies on network architectures and transport schedules.

| Setting | Backbone | Schedule | P↑ |
|---|---|---|---|
| End-to-End (S4) | UDBM | None (Direct) | 28.71 |
| Brownian Bridge (S2) | UDBM | Brownian Bridge | 27.32 |
| DiffUIR-Schedule (S1) | UDBM | DiffUIR | 29.63 |
| DiffUIR (Baseline) | DiffUIR | DiffUIR | 30.18 |
| UDBM-Schedule (S3) | DiffUIR | UDBM (Ours) | **32.69** |
| **UDBM-L (Ours)** | UDBM | UDBM (Ours) | 32.61 |

*Table 8.* Quantitative comparisons of PSNR under varying inference steps on low-light image enhancement.

| Steps | 1 | 3 | 10 | 20 | 30 | 50 | 100 |
|---|---|---|---|---|---|---|---|
| DiffUIR | 24.45 | 25.26 | 24.81 | 24.47 | 24.28 | 24.04 | 23.78 |
| **UDBM-L** | 26.55 | 26.58 | 26.55 | 26.50 | 26.47 | 26.43 | 26.36 |

## 5. Conclusion

We presented UDBM, a framework that rethinks AiOIR as a stochastic transport problem. By introducing a relaxed diffusion bridge, we theoretically resolve the drift singularity inherent in deterministic conditioning, providing a robust mechanism for degradation uncertainty. Through dual uncertainty-guided scheduling, UDBM dynamically reconciles diverse distributions into a shared latent space while regulating trajectories based on restoration difficulty. Extensive experiments demonstrate that UDBM achieves state-of-the-art performance with remarkable single-step efficiency, offering a scalable solution for AiOIR. However, UDBM naturally relies on the auxiliary network for estimating uncertainty. Under extreme out-of-distribution scenarios exhibiting domain shifts, the residual estimation may become unreliable, which can subsequently bottleneck the final restoration quality. Future work will explore more robust uncertainty estimators for extreme degradations.

## Acknowledgements

This work was supported by the National Natural Science Foundation of China under Grant U24A20251, 62071500, Shenzhen Science and Technology Program under Grant JCYJ20230807111107015.

## Impact Statement

UDBM achieves high-fidelity image restoration by enabling robust single-step inference in heterogeneous degradations. By significantly reducing computational costs and eliminating the need for task-specific models, UDBM facilitates scalable deployment in resource-constrained environments. Its versatility and efficiency enable broad societal impact in fields relying on clear visual perception, such as autonomous driving safety, disaster response, and mobile imaging.

eroscedastic regression exhibits high numerical sensitivity in AiOIR. In contrast, the residual-based estimator (Zhang et al., 2025) utilized in UDBM achieves the best performance. Details of these methods are in the Appendix G. More experiments on the robustness of UDBM for uncertainty prediction are provided in Appendix J.3.

**Effectiveness of the Proposed Transport Design.** To disentangle the contributions of our UDBM schedules from the network architecture, we conduct controlled ablations in AiOIR, as reported in Tab. 7. Utilizing a traditional Brownian Bridge schedule (S2) is insufficient for AiOIR. Furthermore, applying the DiffUIR schedule to our backbone (S1) yields inferior results compared to the original DiffUIR. In contrast, integrating our UDBM schedule with the DiffUIR backbone (S3) achieves gains. This confirms that the performance improvements stem from our proposed transport design rather than backbone capacity. While S3 slightly outperforms our default UDBM-L (32.61 dB), UDBM-L maintains a trade-off between performance and computational efficiency.

**Stability Under Multi-Step Inference.** While UDBM excels in highly efficient single-step inference, it supports DDIM-style multi-step sampling. As shown in Table 8, when varying inference steps, UDBM maintains stable. It achieves near-saturation at just one step and varies by only 0.22 dB from best to worst, whereas DiffUIR exhibits degradation in a large inference step (varying by 1.48 dB), suffering from imprecise mappings.

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

## Mathematical Notation

Throughout the article and appendices, the uncertainty-aware coefficients $\alpha_t(\mathbf{u})$, $\gamma_t(\mathbf{u})$, and $\beta_t(\mathbf{u})$, as well as the terminal variance $\boldsymbol{\sigma}_u^2$, are spatially varying tensors matching the image dimensions ($H \times W \times C$). Consequently, all arithmetic operations involving these terms (e.g., multiplication, division, addition, and squaring) are performed **element-wise** (Hadamard operations), unless specified otherwise. For simplicity, we omit the explicit dependency ($\mathbf{u}$) in derivations where the context is clear.

## A. Drift Singularity in General Diffusion Bridges

1 In this section, we provide the analysis showing that the drift singularity induced by deterministic terminal conditioning is a fundamental property of diffusion bridges constructed from general Itô diffusions. We further demonstrate that introducing a soft terminal constraint regularizes this singular behavior effectively.

### A.1. General Diffusion Prior and Doob's $h$-Transform

Let $\{\mathbf{x}_t\}_{t \in [0,1]}$ be a $d$-dimensional Itô diffusion in $\mathbb{R}^d$ governed by the stochastic differential equation (SDE):

$$d\mathbf{x}_t = \mathbf{f}(\mathbf{x}_t, t)\, dt + g(t)\, d\mathbf{w}_t, \tag{12}$$

where $\mathbf{f} : \mathbb{R}^d \times [0,1] \to \mathbb{R}^d$ is assumed to be locally Lipschitz in $\mathbf{x}$ and continuous in $t$. This assumption implies that the prior diffusion process is non-singular and well-behaved locally, ensuring that any singularity arising in the bridge process is strictly a consequence of the terminal conditioning. For notational simplicity and without loss of generality, we assume an isotropic diffusion coefficient where $g(t) : [0,1] \to \mathbb{R}_{>0}$ is a bounded, continuous scalar function, and $\mathbf{w}_t$ denotes a standard $d$-dimensional Brownian motion.

Let $p(\mathbf{x}_1 | \mathbf{x}_t)$ denote the transition density of the process, conditioning the diffusion on a target terminal distribution $p_{\text{target}}(\mathbf{x}_1)$ via Doob's $h$-transform yields the bridge SDE:

$$d\mathbf{x}_t = \Big[\mathbf{f}(\mathbf{x}_t, t) + g(t)^2 \nabla_{\mathbf{x}_t} \log h(\mathbf{x}_t, t)\Big] dt + g(t)\, d\mathbf{w}_t, \tag{13}$$

where the harmonic function $h$ represents the likelihood of reaching the target distribution:

$$h(\mathbf{x}_t, t) = \int p(\mathbf{x}_1 | \mathbf{x}_t)\, p_{\text{target}}(\mathbf{x}_1)\, d\mathbf{x}_1. \tag{14}$$

### A.2. Short-Time Asymptotics of the Transition Density

Let $\Delta t = 1 - t$ be the time-to-go. We analyze the relative magnitude of the deterministic and stochastic components in the SDE discretization when $t \to 1$:

$$\Delta \mathbf{x}_t \approx \mathbf{f}(\mathbf{x}_t, t)\Delta t + g(t)\Delta \mathbf{w}_t. \tag{15}$$

The magnitude of the deterministic drift displacement scales linearly with time:

$$\|\Delta \mathbf{x}_{\text{drift}}\| \approx \|\mathbf{f}(\mathbf{x}_t, t)\|\Delta t = \mathcal{O}(\Delta t). \tag{16}$$

In contrast, the stochastic diffusion displacement is governed by the properties of Brownian motion, where $\Delta \mathbf{w}_t \sim \mathcal{N}(\mathbf{0}, \Delta t \mathbf{I})$, $\sim$ denotes to asymptotic behavior. Its typical magnitude (measured by the root-mean-square) scales with the square root of time:

$$\text{RMS}(\Delta \mathbf{x}_{\text{diff}}) = \sqrt{\mathbb{E}[\|g(t)\Delta \mathbf{w}_t\|^2]} = g(t)\sqrt{d \cdot \Delta t} = \mathcal{O}(\sqrt{\Delta t}). \tag{17}$$

Comparing the two terms in the limit $\Delta t \to 0$:

$$\lim_{\Delta t \to 0} \frac{\|\text{Drift}\|}{\|\text{Diffusion}\|} \propto \lim_{\Delta t \to 0} \frac{\Delta t}{\sqrt{\Delta t}} = \lim_{\Delta t \to 0} \sqrt{\Delta t} = 0. \tag{18}$$

This confirms that the diffusion term is asymptotically dominant ($\mathcal{O}(\sqrt{\Delta t}) \gg \mathcal{O}(\Delta t)$). Consequently, the noise effectively masks the local curvature of the drift field, rendering the transition density Gaussian to leading order:

$$p(\mathbf{x}_1 | \mathbf{x}_t) = \mathcal{N}\big(\mathbf{x}_1; \boldsymbol{\mu}(\mathbf{x}_t, t), \bar{\sigma}_t^2 \mathbf{I}\big)\big(1 + \mathcal{O}(1 - t)\big), \tag{19}$$

where $\boldsymbol{\mu}(\mathbf{x}_t, t) = \mathbf{x}_t + \mathbf{f}(\mathbf{x}_t, t)(1-t)$ is the first-order mean expansion. The accumulated diffusion variance is defined as $\bar{\sigma}_t^2 = \int_t^1 g(s)^2 ds \sim g(1)^2(1-t)$. Note that higher-order corrections (arising from the Hessian of $\mathbf{f}$ and variations in $g$) are of order $\mathcal{O}((1-t)^2)$ or smaller, and thus vanish in the limit.

The gradient of the score function relies on the Jacobian of the mean $\boldsymbol{\mu}$ with respect to the state $\mathbf{x}_t$. We expand the gradient operator $\nabla_{\mathbf{x}_t}$ linearly over the definition of $\boldsymbol{\mu}$:

$$
\begin{aligned}
\mathbf{J}_{\boldsymbol{\mu}}(\mathbf{x}_t, t) &\triangleq \nabla_{\mathbf{x}_t} \boldsymbol{\mu}(\mathbf{x}_t, t) \\
&= \nabla_{\mathbf{x}_t} \left[ \mathbf{x}_t + \mathbf{f}(\mathbf{x}_t, t)(1-t) \right] \\
&= \nabla_{\mathbf{x}_t} \mathbf{x}_t + \nabla_{\mathbf{x}_t} \left[ \mathbf{f}(\mathbf{x}_t, t)(1-t) \right].
\end{aligned}
\tag{20}
$$

The first term is the derivative of a vector with respect to itself, which yields the identity matrix $\mathbf{I} \in \mathbb{R}^{d \times d}$. The second term involves the Jacobian of the drift function $\mathbf{f}$. Since $\mathbf{f}$ is locally Lipschitz, its partial derivatives are bounded. Thus:

$$
\mathbf{J}_{\boldsymbol{\mu}}(\mathbf{x}_t, t) = \mathbf{I} + \left( \nabla_{\mathbf{x}_t} \mathbf{f}(\mathbf{x}_t, t) \right)(1-t) = \mathbf{I} + \mathcal{O}(1-t).
\tag{21}
$$

This implies that for small $1-t$, the mapping $\mathbf{x}_t \mapsto \boldsymbol{\mu}(\mathbf{x}_t, t)$ is a near-identity transformation, preserving the geometric structure of the Euclidean distance in the score computation.

### A.3. Deterministic Terminal Constraint and Drift Singularity

Consider the deterministic terminal constraint imposed by a Dirac delta distribution:

$$
p_{\text{target}}(\mathbf{x}_1) = \delta(\mathbf{x}_1 - \mathbf{x}_{lq}).
\tag{22}
$$

In this setting, the harmonic function reduces directly to the transition density evaluated at $\mathbf{x}_{lq}$:

$$
h_{\text{strict}}(\mathbf{x}_t, t) = p(\mathbf{x}_{lq} | \mathbf{x}_t) \propto \exp\left( -\frac{\|\mathbf{x}_{lq} - \boldsymbol{\mu}(\mathbf{x}_t, t)\|^2}{2\bar{\sigma}_t^2} \right).
\tag{23}
$$

To derive the score (the gradient of the log-harmonic function), we apply the chain rule using Eq. (21):

$$
\begin{aligned}
\nabla_{\mathbf{x}_t} \log h_{\text{strict}}(\mathbf{x}_t, t) &= -\frac{1}{2\bar{\sigma}_t^2} \nabla_{\mathbf{x}_t} \|\mathbf{x}_{lq} - \boldsymbol{\mu}(\mathbf{x}_t, t)\|^2 \\
&= -\frac{1}{\bar{\sigma}_t^2} \left( \mathbf{J}_{\boldsymbol{\mu}}(\mathbf{x}_t, t) \right)^\top \left( \boldsymbol{\mu}(\mathbf{x}_t, t) - \mathbf{x}_{lq} \right) \\
&= \frac{\mathbf{x}_{lq} - \mathbf{x}_t}{\bar{\sigma}_t^2} \left( 1 + \mathcal{O}(1-t) \right).
\end{aligned}
\tag{24}
$$

Substituting the asymptotic variance $\bar{\sigma}_t^2 \sim g(1)^2(1-t)$, the correction term in the bridge drift (Eq. (13)) simplifies asymptotically. Assuming the continuity of $g(t)$, we have $g(t)^2 / g(1)^2 \to 1$, yielding the limiting behavior:

$$
g(t)^2 \nabla_{\mathbf{x}_t} \log h_{\text{strict}}(\mathbf{x}_t, t) \sim g(1)^2 \frac{\mathbf{x}_{lq} - \mathbf{x}_t}{g(1)^2(1-t)} = \frac{\mathbf{x}_{lq} - \mathbf{x}_t}{1-t}, \quad \text{as } t \to 1.
\tag{25}
$$

The singularity arises from a fundamental *mismatch in convergence rates* between the numerator and the denominator. Although the state $\mathbf{x}_t$ converges to $\mathbf{x}_{lq}$, the diffusive nature of the underlying Brownian motion ensures that the expected deviation scales as $\|\mathbf{x}_{lq} - \mathbf{x}_t\| \sim \mathcal{O}(\sqrt{1-t})$ (Eq. 17). In contrast, the denominator vanishes linearly as $\mathcal{O}(1-t)$. Since the stochastic deviation (numerator) decays more slowly than the remaining time (denominator), the ratio diverges:

$$
\lim_{t \to 1} \left\| \frac{\mathbf{x}_{lq} - \mathbf{x}_t}{1-t} \right\| \propto \lim_{t \to 1} \frac{\sqrt{1-t}}{1-t} = \lim_{t \to 1} \frac{1}{\sqrt{1-t}} = \infty.
\tag{26}
$$

This confirms that the drift force becomes unbounded, causing numerical instability near the terminal time.

## A.4. Relaxed Terminal Distribution and Drift Regularization

To mitigate this singularity, we introduce a relaxed terminal distribution modeled as a Gaussian centered at $\mathbf{x}_{lq}$ with a pixel-wise variance map $\boldsymbol{\sigma}_u^2 \in \mathbb{R}^{H \times W \times C}$:

$$p_{\text{target}}(\mathbf{x}_1) = \mathcal{N}(\mathbf{x}_1; \mathbf{x}_{lq}, \boldsymbol{\sigma}_u^2), \tag{27}$$

The harmonic function is now the convolution of the transition density and the target Gaussian. Due to the element-wise independence assumption, this yields another Gaussian with summed variances:

$$h_{\text{relaxed}}(\mathbf{x}_t, t) = \mathcal{N}\left(\mathbf{x}_{lq}; \boldsymbol{\mu}(\mathbf{x}_t, t), \bar{\sigma}_t^2 + \boldsymbol{\sigma}_u^2\right). \tag{28}$$

Following the same gradient derivation as before, the score function (computed element-wise) becomes:

$$\nabla_{\mathbf{x}_t} \log h_{\text{relaxed}}(\mathbf{x}_t, t) = \frac{\mathbf{x}_{lq} - \mathbf{x}_t}{\bar{\sigma}_t^2 + \boldsymbol{\sigma}_u^2}\left(1 + \mathcal{O}(1 - t)\right). \tag{29}$$

In the limit $t \to 1$, the transition variance $\bar{\sigma}_t^2 \to 0$, but the regularization term $\boldsymbol{\sigma}_u^2$ prevents the denominator from vanishing. The bridge drift correction converges to a finite value:

$$\lim_{t \to 1} g(t)^2 \nabla_{\mathbf{x}_t} \log h_{\text{relaxed}}(\mathbf{x}_t, t) = \frac{g(1)^2}{\boldsymbol{\sigma}_u^2}(\mathbf{x}_{lq} - \mathbf{x}_1). \tag{30}$$

# B. Theoretical Justification from an Entropy-Regularized Optimal Transport Perspective

In this section, we provide a rigorous theoretical justification for the proposed uncertainty-aware path scheduling strategy by establishing its connection to entropy-regularized optimal transport (EOT) and Schrödinger bridges. We show that the non-linear, uncertainty-dependent interpolation adopted in our diffusion bridge can be interpreted as a spatially adaptive time reparameterization of an entropy-regularized transport geodesic.

## B.1. Background: Dynamics Optimal Transport

Let $\rho_0$ and $\rho_1$ be two probability measures on $\mathcal{X} = \mathbb{R}^d$. The classical quadratic-cost optimal transport problem admits the dynamics Benamou–Brenier formulation (Benamou & Brenier, 2000):

$$\min_{\rho_t, \mathbf{v}_t} \int_0^1 \int_{\mathcal{X}} \frac{1}{2} \rho_t(\mathbf{x}) \|\mathbf{v}_t(\mathbf{x})\|^2 \, d\mathbf{x} \, dt, \quad \text{s.t. } \partial_t \rho_t + \nabla \cdot (\rho_t \mathbf{v}_t) = 0, \tag{31}$$

with boundary conditions $\rho_{t=0} = \rho_0$ and $\rho_{t=1} = \rho_1$. Under mild regularity conditions, the solution corresponds to a constant-speed geodesic in Wasserstein space.

## B.2. Entropy-Regularized Optimal Transport and Schrödinger Bridges

Entropy-regularized optimal transport can be rigorously formulated through the Schrödinger bridge problem (Léonard, 2013), which introduces stochasticity by considering transport relative to a Brownian reference process. In its dynamics formulation, the Schrödinger bridge problem is given by:

$$\min_{\rho_t, \mathbf{v}_t} \int_0^1 \int_{\mathcal{X}} \frac{1}{2} \rho_t(\mathbf{x}) \|\mathbf{v}_t(\mathbf{x})\|^2 \, d\mathbf{x} \, dt, \tag{32}$$

subject to the Fokker–Planck constraint with diffusion coefficient $\varepsilon$:

$$\partial_t \rho_t + \nabla \cdot (\rho_t \mathbf{v}_t) = \varepsilon \, \Delta \rho_t, \tag{33}$$

where $\varepsilon > 0$ denotes the entropy regularization strength (or noise temperature).

As $\varepsilon \to 0$, Eq. (32)–33 recovers the deterministic optimal transport problem, while larger $\varepsilon$ induces increasingly diffusive and stochastic transport paths. This formulation is known to be equivalent to entropy-regularized optimal transport (Léonard, 2013).

## B.3. Detailed Derivation of the HJB–Fokker–Planck System

We derive the optimality conditions using the method of Lagrange multipliers. We introduce a scalar potential field $\phi_t(\mathbf{x})$ as the Lagrange multiplier for the constraint 33. The Lagrangian functional $\mathcal{L}(\rho, \mathbf{v}, \phi)$ is defined as:

$$\mathcal{L} = \int_0^1 \int_{\mathcal{X}} \left[ \frac{1}{2} \rho_t \|\mathbf{v}_t\|^2 + \phi_t \left( \varepsilon \Delta \rho_t - \partial_t \rho_t - \nabla \cdot (\rho_t \mathbf{v}_t) \right) \right] d\mathbf{x}\, dt. \tag{34}$$

### B.3.1. INTEGRATION BY PARTS

To facilitate variational differentiation with respect to $\rho_t$ and $\mathbf{v}_t$, we first transform the constraint terms using integration by parts. Assuming terminal terms vanish at infinity (or on the domain terminal) and at the time endpoints for the variations, we have:

1. **Time derivative term:**

$$- \int_0^1 \int_{\mathcal{X}} \phi_t\, \partial_t \rho_t\, d\mathbf{x}\, dt = \int_0^1 \int_{\mathcal{X}} \rho_t\, \partial_t \phi_t\, d\mathbf{x}\, dt. \tag{35}$$

2. **Advection term:**

$$- \int_0^1 \int_{\mathcal{X}} \phi_t \nabla \cdot (\rho_t \mathbf{v}_t)\, d\mathbf{x}\, dt = \int_0^1 \int_{\mathcal{X}} \rho_t \mathbf{v}_t \cdot \nabla \phi_t\, d\mathbf{x}\, dt. \tag{36}$$

3. **Diffusion term:** Applying Green's second identity (double integration by parts in space), the Laplacian moves to the multiplier:

$$\int_0^1 \int_{\mathcal{X}} \phi_t\, \varepsilon \Delta \rho_t\, d\mathbf{x}\, dt = \int_0^1 \int_{\mathcal{X}} \varepsilon \rho_t \Delta \phi_t\, d\mathbf{x}\, dt. \tag{37}$$

Substituting these back into Eq. (34), the transformed Lagrangian becomes:

$$\mathcal{L} = \int_0^1 \int_{\mathcal{X}} \rho_t \left[ \frac{1}{2} \|\mathbf{v}_t\|^2 + \partial_t \phi_t + \mathbf{v}_t \cdot \nabla \phi_t + \varepsilon \Delta \phi_t \right] d\mathbf{x}\, dt. \tag{38}$$

### B.3.2. OPTIMALITY CONDITIONS

We now take the first variations of the transformed Lagrangian.

**1. Optimal Velocity Field:** Taking the variation with respect to $\mathbf{v}_t$ and setting $\delta \mathcal{L} / \delta \mathbf{v}_t = 0$:

$$\rho_t \mathbf{v}_t + \rho_t \nabla \phi_t = 0 \quad \implies \quad \mathbf{v}_t(\mathbf{x}) = -\nabla \phi_t(\mathbf{x}). \tag{39}$$

**2. Backward HJB Equation:** Taking the variation with respect to the density $\rho_t$ yields the stationarity condition for the potential $\phi_t$. Setting $\delta \mathcal{L} / \delta \rho_t = 0$, we extract the term inside the brackets of Eq. (38):

$$\frac{1}{2} \|\mathbf{v}_t\|^2 + \partial_t \phi_t + \mathbf{v}_t \cdot \nabla \phi_t + \varepsilon \Delta \phi_t = 0. \tag{40}$$

Substituting the optimal velocity $\mathbf{v}_t = -\nabla \phi_t$ into this equation:

$$\frac{1}{2} \| - \nabla \phi_t \|^2 + \partial_t \phi_t + (-\nabla \phi_t) \cdot \nabla \phi_t + \varepsilon \Delta \phi_t = 0. \tag{41}$$

Simplifying the terms:

$$\frac{1}{2} \|\nabla \phi_t\|^2 + \partial_t \phi_t - \|\nabla \phi_t\|^2 + \varepsilon \Delta \phi_t = 0 \quad \implies \quad \partial_t \phi_t + \varepsilon \Delta \phi_t - \frac{1}{2} \|\nabla \phi_t\|^2 = 0. \tag{42}$$

**3. Coupled System:** Combining the optimal control result (Eq. 33) with the primal constraint (Eq. 42), we obtain the fundamental system governing entropy-regularized transport:

$$\begin{cases} \partial_t \rho_t - \nabla \cdot (\rho_t \nabla \phi_t) = \varepsilon \Delta \rho_t, & \text{(Forward Fokker–Planck)} \\ \partial_t \phi_t + \varepsilon \Delta \phi_t = \frac{1}{2} \|\nabla \phi_t\|^2. & \text{(Backward Viscous HJB)} \end{cases} \tag{43}$$

The backward equation is a Hamilton–Jacobi–Bellman (HJB) equation with a viscous regularization term $\varepsilon\Delta\phi_t$. This term serves as a smoothing operator, mathematically justifying why entropy regularization leads to smooth, uncertainty-aware transport trajectories rather than the potentially non-smooth geodesics of classical optimal transport.

### B.4. Interpretation as Adaptive Time Reparameterization

The viscous HJB equation derived above governs the evolution of the transport potential. Crucially, the diffusion term $\varepsilon\Delta\phi_t$ slows down the transport in regions of high curvature in the potential landscape. Our proposed uncertainty-aware path scheduling strategy can be viewed as an approximation of this mechanism: explicitly modulating the transport speed based on uncertainty mirrors the physical effect of the $\varepsilon\Delta\phi_t$ term, which naturally retards transport to allow for diffusive exploration in high-entropy regions.

### B.5. Implication for Mean Trajectories and Time Reparameterization

We now analyze the implications of the derived viscous Hamilton–Jacobi–Bellman equation on the evolution of the transport process. Specifically, we focus on the expected trajectory of a particle transported from a clean image $\mathbf{x}_{hq}$ to a degraded observation $\mathbf{x}_{lq}$.

Under the standard assumption of quadratic transport cost and Gaussian marginals adopted in diffusion bridges, the optimal transport process is a Gaussian bridge. In this setting, the conditional expectation $\mathbb{E}[\mathbf{x}_t]$ is constrained to lie on the straight-line geodesic connecting the endpoints. However, the *rate* of progression along this geodesic depends on the regularization strength. While classical optimal transport ($\varepsilon \to 0$) dictates a constant-speed progression, entropy regularization introduces a competition between deterministic drift and stochastic diffusion.

Formally, the mean trajectory can be expressed as:

$$\mathbb{E}[\mathbf{x}_t] = (1 - s(t))\,\mathbf{x}_{hq} + s(t)\,\mathbf{x}_{lq}, \tag{44}$$

where $s(t) \in [0, 1]$ is a monotonically increasing time reparameterization function satisfying $s(0) = 0$ and $s(1) = 1$.

The effect of the uncertainty-dependent regularization becomes explicit by examining the HJB equation derived in Eq. (43):

$$\mathbf{v}_t = -\nabla\phi_t, \quad \text{where} \quad \partial_t\phi_t + \varepsilon\Delta\phi_t = \frac{1}{2}\|\nabla\phi_t\|^2. \tag{45}$$

In regions with high uncertainty (large $\varepsilon$ and intermediate state), the diffusion term $\varepsilon\Delta\phi_t$ dominates. As a smoothing operator, the Laplacian $\Delta\phi_t$ tends to flatten the potential landscape $\phi_t$, thereby reducing the magnitude of the deterministic drift gradient $\|\nabla\phi_t\|$. Physically, this implies that in high-uncertainty regimes, the transport relies more on stochastic diffusion (exploration) than on deterministic drift (advection). To satisfy the endpoint constraints under reduced drift velocity, the effective progression $s(t)$ along the geodesic is retarded.

## C. Detailed Derivations of Unified Formulation

In this section, we provide rigorous step-by-step derivations to demonstrate that the forward processes of representative diffusion-based image restoration methods (DDBM (Zhou et al., 2023), I$^2$SB (Liu et al., 2023), ResShift (Yue et al., 2024), RDDM (Liu et al., 2024), and DiffUIR (Zheng et al., 2024)) can be unified into our proposed formulation:

$$\mathbf{x}_t = \underbrace{\alpha_t\mathbf{x}_{lq} + \gamma_t\mathbf{x}_{hq}}_{\text{Path Schedule}} + \underbrace{\beta_t\boldsymbol{\epsilon}}_{\text{Noise Schedule}}, \quad \boldsymbol{\epsilon} \sim \mathcal{N}(\mathbf{0}, \mathbf{I}). \tag{46}$$

To ensure clarity and facilitate rigorous verification, we explicitly present the original equations and variable definitions from the respective literature before deriving the mapping to our unified formulation.

### C.1. Denoising Diffusion Bridge Models (DDBM)

**Notation (Zhou et al., 2023):**

- $\mathbf{x}_0$: Data distribution (Clean, $\mathbf{x}_{hq}$).

- $\mathbf{x}_1$: Prior distribution (Degraded, $\mathbf{x}_{lq}$).

**Original Formulation:** DDBM defines the forward diffusion bridge process via the SDE. The marginal distribution $q(\mathbf{x}_t|\mathbf{x}_0, \mathbf{x}_1)$ at time $t$ is derived as:

$$\mathbf{x}_t = \mu_t(\mathbf{x}_0, \mathbf{x}_1) + \sigma_t \boldsymbol{\epsilon}, \tag{47}$$

where the mean $\mu_t$ is defined as:

$$\mu_t(\mathbf{x}_0, \mathbf{x}_1) = \frac{\bar{\sigma}_t^2 \bar{\alpha}_1}{\bar{\sigma}_1^2} \mathbf{x}_1 + \left( \bar{\alpha}_t - \frac{\bar{\sigma}_t^2 \bar{\alpha}_1^2}{\bar{\sigma}_1^2 \bar{\alpha}_t} \right) \mathbf{x}_0. \tag{48}$$

Here, $\bar{\alpha}_t$ and $\bar{\sigma}_t$ correspond to the noise schedule of the underlying VP-SDE or VE-SDE.

**Derivation to Unified Form:** While Eq. (48) appears complex, DDBM typically adopts the **Brownian Bridge** parameterization in practice, where the underlying drift $f = 0$ and diffusion $g = 1$. Under this setting, the coefficients can be simplified as follows:

$$\bar{\alpha}_t = 1, \quad \bar{\sigma}_t^2 = t, \quad \bar{\alpha}_1 = 1, \quad \bar{\sigma}_1^2 = 1. \tag{49}$$

Substituting these into Eq. (48):

$$\text{Coeff of } \mathbf{x}_1 : \frac{t \cdot 1}{1} = t. \tag{50}$$

$$\text{Coeff of } \mathbf{x}_0 : 1 - \frac{t \cdot 1}{1} = 1 - t. \tag{51}$$

Thus, the state $\mathbf{x}_t$ becomes:

$$\begin{aligned} \mathbf{x}_t &= t\mathbf{x}_1 + (1-t)\mathbf{x}_0 + \sigma_{\text{bridge}}(t)\boldsymbol{\epsilon} \\ &= t\mathbf{x}_{lq} + (1-t)\mathbf{x}_{hq} + \beta_t \boldsymbol{\epsilon}. \end{aligned} \tag{52}$$

This strictly matches our unified form with:

$$\alpha_t = t, \quad \gamma_t = 1 - t, \quad \beta_t = \sqrt{t(1-t)}. \tag{53}$$

## C.2. $I^2$SB: Image-to-Image Schrödinger Bridge

**Notation (Liu et al., 2023):**

- $\mathbf{x}_0$: Degraded image ($\mathbf{x}_{lq}$).
- $\mathbf{x}_1$: Clean image ($\mathbf{x}_{hq}$).

**Original Formulation:** $I^2$SB constructs the bridge $p(\mathbf{x}_t|\mathbf{x}_0, \mathbf{x}_1)$ by conditioning a reference diffusion process:

$$q(\mathbf{x}_t|\mathbf{x}_0, \mathbf{x}_1) = \mathcal{N}(\mathbf{x}_t; \boldsymbol{\mu}_t, \Sigma_t \mathbf{I}). \tag{54}$$

The mean $\boldsymbol{\mu}_t$ is derived from the product of the forward and backward marginals of the reference process:

$$\boldsymbol{\mu}_t = \frac{\bar{\sigma}_{1|t}^2 \bar{\alpha}_t}{\bar{\sigma}_{1|t}^2 + \bar{\alpha}_{1|t}^2 \bar{\sigma}_t^2} \mathbf{x}_0 + \frac{\bar{\alpha}_{1|t} \bar{\sigma}_t^2}{\bar{\sigma}_{1|t}^2 + \bar{\alpha}_{1|t}^2 \bar{\sigma}_t^2} \mathbf{x}_1, \tag{55}$$

where $\bar{\alpha}, \bar{\sigma}$ are coefficients from the reference SDE.

**Derivation to Unified Form:** Eq. (55) is a linear combination of $\mathbf{x}_0$ and $\mathbf{x}_1$. We define the coefficients as:

$$c_0(t) = \frac{\bar{\sigma}_{1|t}^2 \bar{\alpha}_t}{\bar{\sigma}_{1|t}^2 + \bar{\alpha}_{1|t}^2 \bar{\sigma}_t^2}, \quad c_1(t) = \frac{\bar{\alpha}_{1|t} \bar{\sigma}_t^2}{\bar{\sigma}_{1|t}^2 + \bar{\alpha}_{1|t}^2 \bar{\sigma}_t^2}. \tag{56}$$

Substituting the notation $\mathbf{x}_0 = \mathbf{x}_{lq}$ and $\mathbf{x}_1 = \mathbf{x}_{hq}$:

$$\mathbf{x}_t = c_0(t)\mathbf{x}_{lq} + c_1(t)\mathbf{x}_{hq} + \Sigma_t^{1/2} \boldsymbol{\epsilon}. \tag{57}$$

The form matches our formulation with:

$$\alpha_t = c_0(t), \quad \gamma_t = c_1(t), \quad \beta_t = \sqrt{\Sigma_t}. \tag{58}$$

## C.3. ResShift: Residual Shifting

**Notation (Yue et al., 2024):**

- $\mathbf{x}_0$: Clean image ($\mathbf{x}_{hq}$).

- $\mathbf{y}$: Degraded image ($\mathbf{x}_{lq}$).

**Original Formulation:** ResShift defines the transition kernel explicitly as shifting the residual ($\mathbf{y} - \mathbf{x}_0$):

$$q(\mathbf{x}_t|\mathbf{x}_0, \mathbf{y}) = \mathcal{N}(\mathbf{x}_t; \mathbf{x}_0 + \eta_t(\mathbf{y} - \mathbf{x}_0), \sigma_t^2\mathbf{I}). \tag{59}$$

**Derivation to Unified Form:** Expanding the mean term:

$$\begin{aligned}\mathbf{x}_t &= \mathbf{x}_0 + \eta_t\mathbf{y} - \eta_t\mathbf{x}_0 + \sigma_t\boldsymbol{\epsilon} \\ &= \eta_t\mathbf{y} + (1 - \eta_t)\mathbf{x}_0 + \sigma_t\boldsymbol{\epsilon}.\end{aligned} \tag{60}$$

Substituting $\mathbf{y} = \mathbf{x}_{lq}$ and $\mathbf{x}_0 = \mathbf{x}_{hq}$:

$$\mathbf{x}_t = \eta_t\mathbf{x}_{lq} + (1 - \eta_t)\mathbf{x}_{hq} + \sigma_t\boldsymbol{\epsilon}. \tag{61}$$

Matching coefficients:

$$\alpha_t = \eta_t, \quad \gamma_t = 1 - \eta_t, \quad \beta_t = \sigma_t. \tag{62}$$

## C.4. RDDM: Residual Denoising Diffusion Models

**Notation (Liu et al., 2024):**

- $\mathbf{I}_{tar}$: Target/Clean image ($\mathbf{x}_{hq}$).

- $\mathbf{I}_{in}$: Input/Degraded image ($\mathbf{x}_{lq}$).

**Original Formulation:** RDDM defines the forward process as:

$$\mathbf{x}_t = \sqrt{\bar{\alpha}_t}\mathbf{I}_{tar} + (1 - \sqrt{\bar{\alpha}_t})\mathbf{I}_{in} + \sqrt{1 - \bar{\alpha}_t}\boldsymbol{\epsilon}, \tag{63}$$

where $\bar{\alpha}_t$ follows the standard cosine or linear schedule.

**Derivation to Unified Form:** Direct substitution yields:

$$\mathbf{x}_t = (1 - \sqrt{\bar{\alpha}_t})\mathbf{x}_{lq} + \sqrt{\bar{\alpha}_t}\mathbf{x}_{hq} + \sqrt{1 - \bar{\alpha}_t}\boldsymbol{\epsilon}. \tag{64}$$

Matching coefficients:

$$\alpha_t = 1 - \sqrt{\bar{\alpha}_t}, \quad \gamma_t = \sqrt{\bar{\alpha}_t}, \quad \beta_t = \sqrt{1 - \bar{\alpha}_t}. \tag{65}$$

## C.5. DiffUIR: Selective Hourglass Mapping

**Notation (Zheng et al., 2024):**

- $\mathbf{x}_0$: Clean image ($\mathbf{x}_{hq}$).

- $\mathbf{I}_{in}$: Degraded image ($\mathbf{x}_{lq}$).

- $\mathbf{I}_{res}$: Residual ($\mathbf{I}_{in} - \mathbf{x}_0$).

**Original Formulation:** DiffUIR defines the process as:

$$\mathbf{x}_t = \mathbf{x}_0 + \bar{\alpha}_t\mathbf{I}_{res} + \bar{\beta}_t\boldsymbol{\epsilon} - \bar{\delta}_t\mathbf{I}_{in}. \tag{66}$$

**Derivation to Unified Form:** Substituting $\mathbf{I}_{res} = \mathbf{x}_{lq} - \mathbf{x}_{hq}$ and $\mathbf{I}_{in} = \mathbf{x}_{lq}$:

$$
\begin{aligned}
\mathbf{x}_t &= \mathbf{x}_{hq} + \bar{\alpha}_t(\mathbf{x}_{lq} - \mathbf{x}_{hq}) - \bar{\delta}_t\mathbf{x}_{lq} + \bar{\beta}_t\boldsymbol{\epsilon} \\
&= \mathbf{x}_{hq} + \bar{\alpha}_t\mathbf{x}_{lq} - \bar{\alpha}_t\mathbf{x}_{hq} - \bar{\delta}_t\mathbf{x}_{lq} + \bar{\beta}_t\boldsymbol{\epsilon} \\
&= (\bar{\alpha}_t - \bar{\delta}_t)\mathbf{x}_{lq} + (1 - \bar{\alpha}_t)\mathbf{x}_{hq} + \bar{\beta}_t\boldsymbol{\epsilon}.
\end{aligned}
\tag{67}
$$

Matching coefficients:

$$
\alpha_t = \bar{\alpha}_t - \bar{\delta}_t, \quad \gamma_t = 1 - \bar{\alpha}_t, \quad \beta_t = \bar{\beta}_t.
\tag{68}
$$

### C.6. Summary of Unified Coefficients

Table 9 summarizes the exact mapping of coefficients for each method.

*Table 9.* Summary of unified coefficients for representative diffusion-based methods.

| Method | Path Schedule ($\alpha_t$) for $\mathbf{x}_{lq}$ | Path Schedule ($\gamma_t$) for $\mathbf{x}_{hq}$ | Noise Schedule ($\beta_t$) |
|---|---|---|---|
| DDBM | $\frac{\bar{\sigma}_t^2\bar{\alpha}_1}{\bar{\sigma}_1^2}$ | $\bar{\alpha}_t - \frac{\bar{\sigma}_t^2\bar{\alpha}_1^2}{\bar{\sigma}_1^2\bar{\alpha}_t}$ | $\sqrt{\bar{\sigma}_t^2 - \frac{\bar{\sigma}_t^4\bar{\alpha}_1^2}{\bar{\sigma}_1^2}}$ |
| I$^2$SB | $\frac{\bar{\sigma}_{1\|t}^2\bar{\alpha}_t}{\bar{\sigma}_{1\|t}^2 + \bar{\alpha}_{1\|t}^2\bar{\sigma}_t^2}$ | $\frac{\bar{\alpha}_{1\|t}\bar{\sigma}_t^2}{\bar{\sigma}_{1\|t}^2 + \bar{\alpha}_{1\|t}^2\bar{\sigma}_t^2}$ | $\sqrt{\Sigma_t}$ |
| ResShift | $\eta_t$ | $1 - \eta_t$ | $\sigma_t$ |
| RDDM | $1 - \sqrt{\bar{\alpha}_t}$ | $\sqrt{\bar{\alpha}_t}$ | $\sqrt{1 - \bar{\alpha}_t}$ |
| DiffUIR | $\bar{\alpha}_t - \bar{\delta}_t$ | $1 - \bar{\alpha}_t$ | $\bar{\beta}_t$ |

# D. Detailed Derivation of the Reverse Process

In this section, we present the step-by-step derivation of the reverse distribution $q(\mathbf{x}_s|\mathbf{x}_t, \mathbf{x}_{lq}, \mathbf{x}_{hq})$, where $s$ and $t$ denote discrete time steps where $0 \le s < t \le 1$ in the reverse process schedule. We utilize Bayes' theorem and the probability density function (PDF) of the normal distribution.

### D.1. Bayesian Formulation

We aim to represent the distribution $q(\mathbf{x}_s|\mathbf{x}_t, \mathbf{x}_{lq}, \mathbf{x}_{hq})$ by its mean and variance. Based on Bayes' theorem, we express the posterior as the product of the forward transition likelihood and the prior at step $s$:

$$
q(\mathbf{x}_s|\mathbf{x}_t, \mathbf{x}_{lq}, \mathbf{x}_{hq}) = \frac{q(\mathbf{x}_t|\mathbf{x}_s, \mathbf{x}_{lq}, \mathbf{x}_{hq}) \cdot q(\mathbf{x}_s|\mathbf{x}_{lq}, \mathbf{x}_{hq})}{q(\mathbf{x}_t|\mathbf{x}_{lq}, \mathbf{x}_{hq})}.
\tag{69}
$$

Since the denominator $q(\mathbf{x}_t|\mathbf{x}_{lq}, \mathbf{x}_{hq})$ is a constant with respect to $\mathbf{x}_s$, we focus on the numerator $q(\mathbf{x}_t|\mathbf{x}_s, \mathbf{x}_{lq}, \mathbf{x}_{hq}) \cdot q(\mathbf{x}_s|\mathbf{x}_{lq}, \mathbf{x}_{hq})$. According to $\mathbf{x}_s = \alpha_s\mathbf{x}_{lq} + \gamma_s\hat{\mathbf{x}}_{0|t} + \beta_s\boldsymbol{\epsilon}_t$, the distributions are defined as:

$$
q(\mathbf{x}_s|\mathbf{x}_{lq}, \mathbf{x}_{hq}) = \mathcal{N}(\mathbf{x}_s; \alpha_s\mathbf{x}_{lq} + \gamma_s\mathbf{x}_{hq}, \beta_s^2\mathbf{I}),
\tag{70}
$$

$$
q(\mathbf{x}_t|\mathbf{x}_s, \mathbf{x}_{lq}, \mathbf{x}_{hq}) = \mathcal{N}(\mathbf{x}_t; \mathbf{x}_s + (\alpha_t - \alpha_s)\mathbf{x}_{lq} + (\gamma_t - \gamma_s)\mathbf{x}_{hq}, (\beta_t^2 - \beta_s^2)\mathbf{I}).
\tag{71}
$$

For clarity in the derivation, we define $\Delta\alpha = \alpha_t - \alpha_s$, $\Delta\gamma = \gamma_t - \gamma_s$, and $\beta_{t|s}^2 = \beta_t^2 - \beta_s^2$.

### D.2. Exponential Expansion

We expand the exponential terms of the PDFs to identify the quadratic form of $\mathbf{x}_s$. The joint probability is proportional to:

$$
\begin{aligned}
&q(\mathbf{x}_t|\mathbf{x}_s, \mathbf{x}_{lq}, \mathbf{x}_{hq}) \cdot q(\mathbf{x}_s|\mathbf{x}_{lq}, \mathbf{x}_{hq}) \\
&\propto \exp\left[-\frac{1}{2}\left(\frac{||\mathbf{x}_t - \mathbf{x}_s - \Delta\alpha\mathbf{x}_{lq} - \Delta\gamma\mathbf{x}_{hq}||^2}{\beta_{t|s}^2} + \frac{||\mathbf{x}_s - \alpha_s\mathbf{x}_{lq} - \gamma_s\mathbf{x}_{hq}||^2}{\beta_s^2}\right)\right].
\end{aligned}
\tag{72}
$$

To isolate $\mathbf{x}_s$, we expand the squared norms. Notice that inside the first norm, we can group terms as $(\mathbf{x}_t - \Delta\alpha\mathbf{x}_{lq} - \Delta\gamma\mathbf{x}_{hq}) - \mathbf{x}_s$.

$$q(\mathbf{x}_t|\mathbf{x}_s, \mathbf{x}_{lq}, \mathbf{x}_{hq}) \cdot q(\mathbf{x}_s|\mathbf{x}_{lq}, \mathbf{x}_{hq})$$
$$\propto exp\left[-\frac{1}{2}\left(\frac{\mathbf{x}_s^2 - 2\mathbf{x}_s(\mathbf{x}_t - \Delta\alpha\mathbf{x}_{lq} - \Delta\gamma\mathbf{x}_{hq}) + C_1}{\beta_{t|s}^2} + \frac{\mathbf{x}_s^2 - 2\mathbf{x}_s(\alpha_s\mathbf{x}_{lq} + \gamma_s\mathbf{x}_{hq}) + C_2}{\beta_s^2}\right)\right], \tag{73}$$

where $C_1, C_2$ are terms independent of $\mathbf{x}_s$. Now, we group the coefficients of $\mathbf{x}_s^2$ and $\mathbf{x}_s$:

$$q(\mathbf{x}_t|\mathbf{x}_s, \mathbf{x}_{lq}, \mathbf{x}_{hq}) \cdot q(\mathbf{x}_s|\mathbf{x}_{lq}, \mathbf{x}_{hq})$$
$$\propto \exp\left[-\frac{1}{2}\left(\underbrace{\left(\frac{1}{\beta_{t|s}^2} + \frac{1}{\beta_s^2}\right)}_{\text{Term A}}\mathbf{x}_s^2 - 2\mathbf{x}_s\underbrace{\left(\frac{\mathbf{x}_t - \Delta\alpha\mathbf{x}_{lq} - \Delta\gamma\mathbf{x}_{hq}}{\beta_{t|s}^2} + \frac{\alpha_s\mathbf{x}_{lq} + \gamma_s\mathbf{x}_{hq}}{\beta_s^2}\right)}_{\text{Term B}}\right)\right]. \tag{74}$$

### D.3. Calculation of Mean and Variance

We match Eq. (74) with the standard form $(\exp[-\frac{1}{2\tilde{\sigma}_t^2}(\mathbf{x}_s^2 - 2\mathbf{x}_s\tilde{\boldsymbol{\mu}}_t)])$.

**1. Variance $\tilde{\sigma}_t^2$:**
$$\frac{1}{\tilde{\sigma}_t^2} = \frac{1}{\beta_t^2 - \beta_s^2} + \frac{1}{\beta_s^2} = \frac{\beta_s^2 + (\beta_t^2 - \beta_s^2)}{(\beta_t^2 - \beta_s^2)\beta_s^2} = \frac{\beta_t^2}{\beta_s^2(\beta_t^2 - \beta_s^2)}. \tag{75}$$

Taking the inverse gives the posterior variance:

$$\tilde{\sigma}_t^2 = \frac{\beta_s^2(\beta_t^2 - \beta_s^2)}{\beta_t^2}. \tag{76}$$

**2. Mean $\tilde{\mu}_t$:** The mean is given by:

$$\tilde{\boldsymbol{\mu}}_t = \frac{\beta_s^2(\beta_t^2 - \beta_s^2)}{\beta_t^2}\left(\frac{\mathbf{x}_t - \Delta\alpha\mathbf{x}_{lq} - \Delta\gamma\mathbf{x}_{hq}}{\beta_t^2 - \beta_s^2} + \frac{\alpha_s\mathbf{x}_{lq} + \gamma_s\mathbf{x}_{hq}}{\beta_s^2}\right)$$
$$= \frac{\beta_s^2}{\beta_t^2}(\mathbf{x}_t - (\alpha_t - \alpha_s)\mathbf{x}_{lq} - (\gamma_t - \gamma_s)\mathbf{x}_{hq}) + \frac{\beta_t^2 - \beta_s^2}{\beta_t^2}(\alpha_s\mathbf{x}_{lq} + \gamma_s\mathbf{x}_{hq}). \tag{77}$$

To simplify, we use the forward process relation $\gamma_t\mathbf{x}_{hq} = \mathbf{x}_t - \alpha_t\mathbf{x}_{lq} - \beta_t\boldsymbol{\epsilon}$. We group terms by $\mathbf{x}_{lq}$ and $\mathbf{x}_{hq}$:

$$\tilde{\boldsymbol{\mu}}_t = \frac{\beta_s^2}{\beta_t^2}\mathbf{x}_t + \left(\frac{\beta_t^2 - \beta_s^2}{\beta_t^2}\alpha_s - \frac{\beta_s^2}{\beta_t^2}(\alpha_t - \alpha_s)\right)\mathbf{x}_{lq} + \left(\frac{\beta_t^2 - \beta_s^2}{\beta_t^2}\gamma_s - \frac{\beta_s^2}{\beta_t^2}(\gamma_t - \gamma_s)\right)\mathbf{x}_{hq}. \tag{78}$$

Substituting $\mathbf{x}_t$ in Eq. (78) yields:

$$\tilde{\boldsymbol{\mu}}_t = \alpha_s\mathbf{x}_{lq} + \gamma_s\mathbf{x}_{hq} + \frac{\beta_s^2}{\beta_t^2} \cdot \beta_t\boldsymbol{\epsilon}. \tag{79}$$

Replacing $\boldsymbol{\epsilon}$ with the predicted noise $\boldsymbol{\epsilon}_{\text{pred}} = \frac{\mathbf{x}_t - \alpha_t\mathbf{x}_{lq} - \gamma_t\hat{\mathbf{x}}_{0|t}}{\beta_t}$ gives the final computable mean:

$$\tilde{\boldsymbol{\mu}}_t = \alpha_s\mathbf{x}_{lq} + \gamma_s\hat{\mathbf{x}}_{0|t} + \frac{\beta_s^2}{\beta_t^2}(\mathbf{x}_t - \alpha_t\mathbf{x}_{lq} - \gamma_t\hat{\mathbf{x}}_{0|t}). \tag{80}$$

### D.4. Sampling Strategies

Having derived the posterior mean $\tilde{\boldsymbol{\mu}}_t$ and variance $\tilde{\sigma}_t^2$, we can now instantiate the generative process. The general update rule at step $t$ samples $\mathbf{x}_s$ from the Gaussian distribution $\mathcal{N}(\tilde{\boldsymbol{\mu}}_t, \sigma_t^2\mathbf{I})$:

$$\mathbf{x}_s = \tilde{\boldsymbol{\mu}}_t + \sigma_t\mathbf{z}, \quad \mathbf{z} \sim \mathcal{N}(\mathbf{0}, \mathbf{I}). \tag{81}$$

Depending on the choice of the schedule $\sigma_t$, we recover different sampling strategies.

**1. DDPM-like Sampling (Stochastic)** To strictly follow the reverse Markov chain derived from the Bayesian posterior, we set the sampling standard deviation $\sigma_t$ equal to the posterior standard deviation $\tilde{\sigma}_t$ derived in Eq. (76). Substituting the mean $\tilde{\boldsymbol{\mu}}_t$ from Eq. (80) and utilizing the predicted noise $\boldsymbol{\epsilon}_{\text{pred}}$, the update rule is:

$$\mathbf{x}_s = \underbrace{\alpha_s \mathbf{x}_{lq} + \gamma_s \hat{\mathbf{x}}_{0|t} + \frac{\beta_s^2}{\beta_t} \boldsymbol{\epsilon}_{\text{pred}}}_{\text{Posterior Mean } \tilde{\boldsymbol{\mu}}_t} + \underbrace{\frac{\beta_s}{\beta_t} \sqrt{\beta_t^2 - \beta_s^2} \cdot \mathbf{z}}_{\text{Posterior Std Dev } \tilde{\sigma}_t \cdot \mathbf{z}} . \tag{82}$$

**2. DDIM-like Sampling (Deterministic)** For the deterministic sampling strategy (DDIM), we set $\sigma_t = 0$. The generalized update rule for arbitrary $\sigma_t$ is:

$$\mathbf{x}_s = \alpha_s \mathbf{x}_{lq} + \gamma_s \hat{\mathbf{x}}_{0|t} + \sqrt{\beta_s^2 - \sigma_t^2} \cdot \boldsymbol{\epsilon}_{\text{pred}} + \sigma_t \mathbf{z}. \tag{83}$$

Setting $\sigma_t = 0$ in Eq. (83):

$$\mathbf{x}_s^{\text{DDIM}} = \alpha_s \mathbf{x}_{lq} + \gamma_s \hat{\mathbf{x}}_{0|t} + \beta_s \boldsymbol{\epsilon}_{\text{pred}}. \tag{84}$$

We can substitute $\boldsymbol{\epsilon}_{\text{pred}} = (\mathbf{x}_t - \alpha_t \mathbf{x}_{lq} - \gamma_t \hat{\mathbf{x}}_{0|t})/\beta_t$ back into Eq. (84):

$$\begin{aligned} \mathbf{x}_s^{\text{DDIM}} &= \left(\alpha_s - \frac{\beta_s}{\beta_t}\alpha_t\right) \mathbf{x}_{lq} + \left(\gamma_s - \frac{\beta_s}{\beta_t}\gamma_t\right) \hat{\mathbf{x}}_{0|t} + \frac{\beta_s}{\beta_t} \mathbf{x}_t. \\ &= \underbrace{\alpha_s \mathbf{x}_{lq} + \gamma_s \hat{\mathbf{x}}_{0|t}}_{\text{Target Mean}} + \underbrace{\frac{\beta_s}{\beta_t}\left(\mathbf{x}_t - \alpha_t \mathbf{x}_{lq} - \gamma_t \hat{\mathbf{x}}_{0|t}\right)}_{\text{Deterministic Direction}} \end{aligned} \tag{85}$$

# E. Kinetic Analysis of Uncertainty-Adaptive Path Schedule

In this section, we provide a derivation to demonstrate that our proposed Path Schedule (Eq. (11)) mathematically satisfies the kinetic constraints identified in Remark 3.5. Specifically, we prove that high uncertainty induces a deceleration of the transport velocity within the intermediate entropic barrier ($t \approx 0.5$) and a compensatory acceleration near the boundaries.

### E.1. Definition of Transport Velocity

Recall the definition of our mean trajectory $\boldsymbol{\mu}_t = \mathbb{E}[\mathbf{x}_t]$:

$$\boldsymbol{\mu}_t = (1 - \alpha_t(\mathbf{u}))\mathbf{x}_{hq} + \alpha_t(\mathbf{u})\mathbf{x}_{lq}. \tag{86}$$

The instantaneous velocity of the mean transport is given by the time derivative of the trajectory:

$$\mathbf{v}_t^{\mu} \triangleq \frac{\partial \boldsymbol{\mu}_t}{\partial t} = \frac{\partial \alpha_t(\mathbf{u})}{\partial t}(\mathbf{x}_{lq} - \mathbf{x}_{hq}). \tag{87}$$

Since $(\mathbf{x}_{lq} - \mathbf{x}_{hq})$ is a constant vector for a given pair, the kinetic magnitude of the transport is governed entirely by the scalar velocity of the schedule coefficient, denoted as $\mathcal{V}_t(\mathbf{u})$:

$$\mathcal{V}_t(\mathbf{u}) \triangleq \frac{\partial \alpha_t(\mathbf{u})}{\partial t}. \tag{88}$$

Our proposed schedule is defined as:

$$\alpha_t(\mathbf{u}) = \frac{t^{\pi}}{t^{\pi} + (1-t)^{\pi}}, \quad \text{where } \pi \equiv \pi(\mathbf{u}) \in [0.5, 1.0]. \tag{89}$$

Here, $\pi(\mathbf{u})$ is a pixel-wise decreasing function of uncertainty $\mathbf{u}$. Specifically, $\pi = 1.0$ corresponds to low uncertainty (OT limit), and $\pi = 0.5$ corresponds to high uncertainty (EOT limit).

## E.2. Derivation of Instantaneous Velocity

We derive the explicit form of $\mathcal{V}_t(\mathbf{u})$ using the quotient rule. Let $N(t) = t^\pi$ and $D(t) = t^\pi + (1-t)^\pi$. The derivative is:

$$\mathcal{V}_t(\mathbf{u}) = \frac{N'(t)D(t) - N(t)D'(t)}{[D(t)]^2}. \tag{90}$$

Calculating the components:

$$N'(t) = \pi t^{\pi-1}, \tag{91}$$

$$D'(t) = \pi t^{\pi-1} - \pi(1-t)^{\pi-1}. \tag{92}$$

Substituting these into the quotient rule:

$$\begin{aligned}
\mathcal{V}_t(\mathbf{u}) &= \frac{\pi t^{\pi-1}\left(t^\pi + (1-t)^\pi\right) - t^\pi\left(\pi t^{\pi-1} - \pi(1-t)^{\pi-1}\right)}{\left(t^\pi + (1-t)^\pi\right)^2} \\
&= \frac{\pi\left[t^{2\pi-1} + t^{\pi-1}(1-t)^\pi - t^{2\pi-1} + t^\pi(1-t)^{\pi-1}\right]}{\left(t^\pi + (1-t)^\pi\right)^2} \\
&= \frac{\pi\left[t^{\pi-1}(1-t)^\pi + t^\pi(1-t)^{\pi-1}\right]}{\left(t^\pi + (1-t)^\pi\right)^2}.
\end{aligned} \tag{93}$$

Factoring out common terms in the numerator:

$$\text{Numerator} = \pi t^{\pi-1}(1-t)^{\pi-1}\underbrace{[(1-t) + t]}_{=1} \tag{94}$$

$$= \pi\left[t(1-t)\right]^{\pi-1}.$$

Thus, we obtain the analytical expression for the element-wise velocity:

$$\mathcal{V}_t(\mathbf{u}) = \pi(\mathbf{u}) \cdot \frac{(t(1-t))^{\pi(\mathbf{u})-1}}{\left(t^{\pi(\mathbf{u})} + (1-t)^{\pi(\mathbf{u})}\right)^2}. \tag{95}$$

## E.3. Kinetic Analysis in Distinct Regimes

We now analyze Eq. (95) to verify the claims in Remark 3.5.

### E.3.1. 1. DECELERATION IN THE ENTROPIC BARRIER ($t = 0.5$)

The maximum entropic barrier is located at the trajectory midpoint $t = 0.5$, where the distributions are maximally mixed. Evaluating $\mathcal{V}_t$ at $t = 0.5$:

$$\begin{aligned}
\mathcal{V}_{0.5}(\mathbf{u}) &= \pi \cdot \frac{(0.25)^{\pi-1}}{((0.5)^\pi + (0.5)^\pi)^2} \\
&= \pi \cdot \frac{(0.5^2)^{\pi-1}}{(2 \cdot 0.5^\pi)^2} \\
&= \pi \cdot \frac{0.5^{2\pi-2}}{4 \cdot 0.5^{2\pi}} \\
&= \frac{\pi}{4} \cdot 0.5^{-2} = \frac{\pi}{4} \cdot 4 = \pi(\mathbf{u}).
\end{aligned} \tag{96}$$

**Conclusion:** The minimum velocity at the midpoint is exactly $\pi(\mathbf{u})$.

- For **Low Uncertainty** ($\mathbf{u} \to 0$), $\pi \to 1$. The velocity $\mathcal{V}_{0.5} \to 1$ (Constant speed).

- For **High Uncertainty** ($\mathbf{u} \to 1$), $\pi \to 0.5$. The velocity $\mathcal{V}_{0.5} \to 0.5$ (Slowed down by 50%).

This proves that high uncertainty explicitly suppresses the drift velocity in the entropic barrier.

This behavior aligns with the viscous HJB dynamics, where the particle moves slowly through the high-entropy region to resolve ambiguity and accelerates near deterministic boundaries to satisfy terminal constraints.

## F. Feasibility of Single-Step Inference

In this section, we derive the Probability Flow ODE (PF-ODE) for UDBM and demonstrate that the Relaxed Diffusion Bridge formulation ensures the boundedness of the velocity field at the terminal $t = 1$. This boundedness, combined with the geometric linearity of the path, guarantees the stability and high fidelity of single-step inference.

### F.1. Derivation of the Probability Flow ODE

The "deterministic assumption" in diffusion models implies that the normalized noise variable $\epsilon$ associated with a trajectory remains invariant over time (i.e., $d\epsilon/dt = 0$). We start with the unified forward marginal distribution defined in UDBM:

$$\mathbf{x}_t = \boldsymbol{\mu}_t + \beta_t \boldsymbol{\epsilon}, \tag{97}$$

where the mean trajectory is $\boldsymbol{\mu}_t = \alpha_t \mathbf{x}_{lq} + \gamma_t \hat{\mathbf{x}}_{0|t}$. Differentiating Eq. (97) with respect to $t$ under the deterministic assumption yields:

$$\frac{d\mathbf{x}_t}{dt} = \dot{\boldsymbol{\mu}}_t + \dot{\beta}_t \boldsymbol{\epsilon}. \tag{98}$$

To obtain a velocity field $\mathbf{v}(\mathbf{x}_t, t)$ that depends only on the state $\mathbf{x}_t$ and the restoration target $\hat{\mathbf{x}}_{0|t}$, we substitute $\boldsymbol{\epsilon} = (\mathbf{x}_t - \boldsymbol{\mu}_t)/\beta_t$ from Eq. (97):

$$\frac{d\mathbf{x}_t}{dt} = \dot{\boldsymbol{\mu}}_t + \dot{\beta}_t \left( \frac{\mathbf{x}_t - \boldsymbol{\mu}_t}{\beta_t} \right). \tag{99}$$

Rearranging the terms, we obtain the general form of the Probability Flow ODE:

$$\frac{d\mathbf{x}_t}{dt} = \underbrace{\frac{\dot{\beta}_t}{\beta_t} \mathbf{x}_t + \left( \dot{\boldsymbol{\mu}}_t - \frac{\dot{\beta}_t}{\beta_t} \boldsymbol{\mu}_t \right)}_{\text{Effective Velocity } \mathbf{v}(\mathbf{x}_t, t)}. \tag{100}$$

Substituting $\boldsymbol{\mu}_t = \alpha_t \mathbf{x}_{lq} + \gamma_t \hat{\mathbf{x}}_{0|t}$, the velocity explicitly involves the restoration term:

$$\mathbf{v}(\mathbf{x}_t, t) = \frac{\dot{\beta}_t}{\beta_t} (\mathbf{x}_t - \alpha_t \mathbf{x}_{lq} - \gamma_t \hat{\mathbf{x}}_{0|t}) + (\dot{\alpha}_t \mathbf{x}_{lq} + \dot{\gamma}_t \hat{\mathbf{x}}_{0|t}). \tag{101}$$

The feasibility of single-step inference relies on two properties: (1) **Boundedness** of the velocity $\mathbf{v}$ at $t = 1$, and (2) **Geometric Linearity** of the trajectory.

### F.2. Boundedness at the Terminal $t = 1$

Single-step inference $\mathbf{x}_0 \approx \mathbf{x}_1 - \mathbf{v}(\mathbf{x}_1, 1)$ requires the velocity $\mathbf{v}(\mathbf{x}_1, 1)$ to be finite. We analyze the behavior of the coefficients in Eq. (100) as $t \to 1$.

In **standard diffusion bridges** (with strict constraints), the noise level often behaves as $\beta_t \propto \sqrt{1-t}$ near $t = 1$. Consequently, the term $\frac{\dot{\beta}_t}{\beta_t} \sim \frac{1}{2(1-t)}$ diverges to infinity, causing numerical instability and large discretization errors.

In contrast, **UDBM** employs a Relaxed Diffusion Bridge where the terminal noise variance is regularized by the uncertainty $\sigma_u^2$. According to our noise schedule (Eq. (10)), at $t = 1$, we have:

$$\beta_1 = 1 + \mathbf{u} > 0. \tag{102}$$

Since $\beta_1$ is a strictly positive constant, the logarithmic derivative $\frac{\dot{\beta}_t}{\beta_t}$ remains finite at $t = 1$. Furthermore, the mean terms $\alpha_t, \gamma_t$ and their derivatives are bounded by design (Eq. (11)). Therefore, the total velocity $\|\mathbf{v}(\mathbf{x}_1, 1)\|$ is bounded, ensuring that the single-step Euler jump is numerically stable.

### F.3. Condition 2: Error Minimization via Geometric Linearity

Our path schedule (Eq. (11a)) strictly enforces the convexity constraint $\gamma_t = 1 - \alpha_t$. This implies that the expected trajectory lies entirely on the linear geodesic connecting the degraded observation $\mathbf{x}_{lq}$ and the clean prior $\mathbf{x}_{hq}$:

$$\mathbb{E}[\mathbf{x}_t] = (1 - \alpha_t(\mathbf{u}))\mathbf{x}_{hq} + \alpha_t(\mathbf{u})\mathbf{x}_{lq}. \tag{103}$$

Geometrically, this represents a straight line segment in the high-dimensional pixel space. For a rectilinear trajectory, the direction of the velocity vector $\mathbf{v}_t$ remains constant throughout the process (pointing directly from $\mathbf{x}_{lq}$ to $\mathbf{x}_{hq}$). While the non-linear schedule $\pi(\mathbf{u})$ introduces a variable scalar speed (temporal curvature), the geometric curvature is zero. Consequently, the single-step inference projects $\mathbf{x}_1$ along the theoretically correct direction. By eliminating the error component associated with geometric curvature, UDBM significantly reduces the discretization error compared to curved stochastic paths, enabling high-quality restoration in a single step.

We analyze the differential geometry of the transport trajectory.

**Theorem F.1** (Zero Geometric Curvature). *Under the UDBM formulation with the convexity constraint $\gamma_t(\mathbf{u}) = \mathbf{I} - \alpha_t(\mathbf{u})$, the trajectory of the deterministic PF-ODE lies strictly on a one-dimensional linear manifold connecting the degraded observation $\mathbf{x}_{lq}$ and the predicted clean prior $\hat{\mathbf{x}}_{0|t}$. Consequently, under the assumption of perfect score matching, the geometric curvature of the transport path is identically zero for all $t \in [0, 1]$.*

*Proof.* Consider the mean trajectory $\boldsymbol{\mu}_t$ defined in Eq. (11a):

$$\boldsymbol{\mu}_t = (\mathbf{I} - \alpha_t(\mathbf{u}))\mathbf{x}_{hq} + \alpha_t(\mathbf{u})\mathbf{x}_{lq}. \tag{104}$$

The velocity vector field of the mean transport is given by the time derivative:

$$\mathbf{v}_t^\mu \triangleq \frac{d\boldsymbol{\mu}_t}{dt} = \frac{\partial \alpha_t}{\partial t}(\mathbf{x}_{lq} - \mathbf{x}_{hq}), \tag{105}$$

Let $\mathbf{d} = \mathbf{x}_{lq} - \mathbf{x}_{hq}$ be the constant displacement vector. The velocity can be factorized as:

$$\mathbf{v}_t^\mu = \mathcal{V}_t(\mathbf{u})\mathbf{d}, \tag{106}$$

where $\mathcal{V}_t(\mathbf{u})$ is the scalar speed schedule derived in Appendix E.

Since the direction of the velocity vector $\mathbf{v}_t^\mu$ is time-invariant (always parallel to the constant vector $\mathbf{d}$), the unit tangent vector $\mathbf{T}_t = \mathbf{v}_t^\mu / \|\mathbf{v}_t^\mu\|$ remains constant throughout the integration time $t \in [0, 1]$. The geometric curvature $\kappa$, defined as the magnitude of the rate of change of the unit tangent vector with respect to arc length $s$, vanishes:

$$\kappa = \left\| \frac{d\mathbf{T}_t}{ds} \right\| = \frac{1}{\|\mathbf{v}_t^\mu\|} \left\| \frac{d\mathbf{T}_t}{dt} \right\| \equiv 0. \tag{107}$$

**Conclusion:** For a trajectory with $\kappa \equiv 0$, the local truncation error of the first-order Euler method (typically $\mathcal{O}(h^2)$) becomes exact regarding the geometric path. Thus, a single integration step $\mathbf{x}_0 \approx \mathbf{x}_1 - \mathbf{v}_1 \cdot 1$ moves the state strictly along the geodesic, theoretically justifying the high fidelity of our single-step inference. $\square$

### F.4. Geometric Validation of Transport Trajectory

we conduct a geometric validation to verify the linearity of the learned transport trajectory.

#### F.4.1. METHODOLOGY: END-POINT ALIGNMENT TEST

Our validation methodology is inspired by the framework of Rectified Flow (Liu et al., 2022), which aims to rectify the stochastic transport path into a deterministic straight-line trajectory. While standard Rectified Flow defines a straight path as having constant velocity (i.e., $Z_t = tZ_1 + (1 - t)Z_0$), the fundamental prerequisite for accurate single-step inference is the geometric linearity of the path. This requires that the velocity field direction remains aligned with the global displacement vector throughout the process, regardless of the scalar speed magnitude.

To empirically verify this geometric linearity (direction consistency), we propose an End-Point Alignment Test. Specifically, we define two vectors:

1. Ideal Transport Vector ($\vec{V}_{\text{ideal}}$): The ground-truth displacement from the degraded input $\mathbf{x}_{lq}$ to the clean target $\mathbf{x}_{hq}$:

$$\vec{V}_{\text{ideal}} = \mathbf{x}_{hq} - \mathbf{x}_{lq}. \tag{108}$$

2. Predicted Restoration Vector ($\vec{V}_{\text{pred}}$): The actual update direction predicted by the model with $N$ inference steps:

$$\vec{V}_{\text{pred}}^{(N)} = \hat{\mathbf{x}}_0^{(N)} - \mathbf{x}_{lq}, \tag{109}$$

where $\hat{\mathbf{x}}_0^{(N)}$ denotes the restored image obtained after $N$ steps.

We calculate the Cosine Similarity between these two vectors:

$$\mathcal{S}_{\cos}(N) = \frac{\vec{V}_{\text{ideal}} \cdot \vec{V}_{\text{pred}}^{(N)}}{\|\vec{V}_{\text{ideal}}\| \|\vec{V}_{\text{pred}}^{(N)}\|}. \tag{110}$$

Since the path is geometrically linear, the direction measured at step $N = 1$ (initial velocity) should be identical to the global direction, validating the feasibility of single-step inference.

F.4.2. EXPERIMENTAL RESULTS AND COMPARATIVE ANALYSIS ON LOL DATASET

To empirically validate the theoretical zero-curvature property, we evaluate the geometric properties on the LOL dataset in the AiOIR task. Furthermore, we compare the convergence behavior of UDBM against DiffUIR (Zheng et al., 2024) to demonstrate the advantage of our rectified transport dynamics.

**1. Geometric Linearity Validation.** We varied the total number of inference steps $N \in \{5, 10, 15, 20\}$ and computed the End-Point Alignment (Cosine Similarity between predicted update vector and ground-truth vector). As shown in Figure 7 (Top), the similarity curve for UDBM reaches approximately **0.987** at $N = 1$ and remains almost constant as $N$ increases. This empirical evidence corroborates Theorem F.1, confirming that the learned transport trajectory is already geometrically straight and requires no further rectification steps.

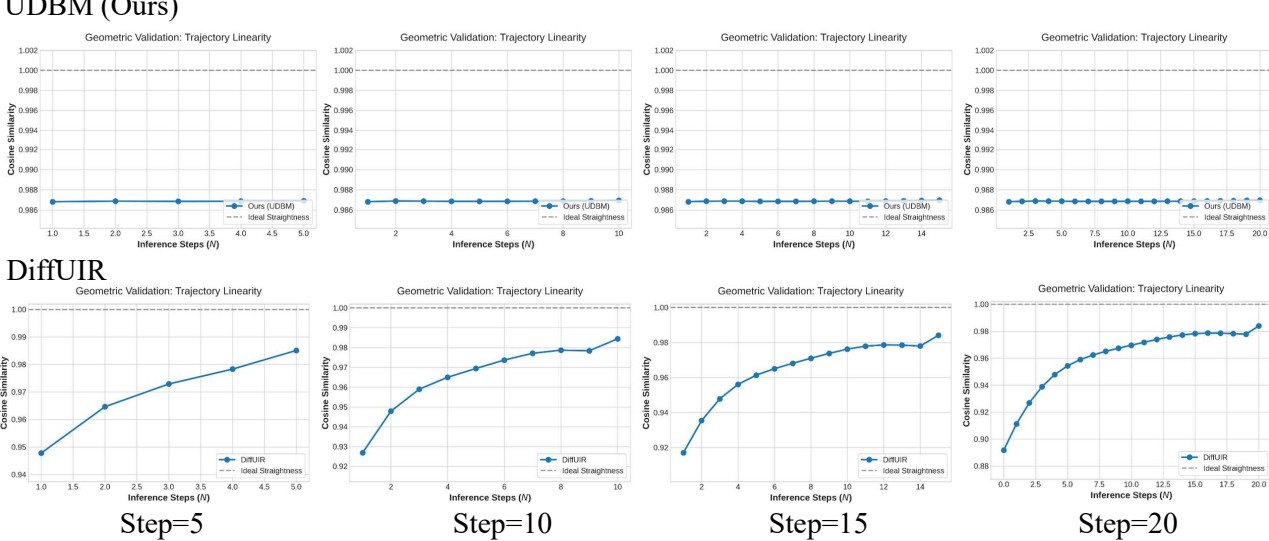

*Figure 7.* Geometric Comparison on LOL Dataset. (Top) End-Point Alignment Test: UDBM achieves optimal direction alignment at Step 1, indicating a straight trajectory. (Down) Convergence Dynamics: UDBM achieves peak performance immediately, whereas DiffUIR exhibits a curved trajectory that necessitates multi-step integration.

**2. Comparison with DiffUIR: Curved vs. Linear Flows.** We further analyze the Cosine Similarity as a function of inference steps for both UDBM and DiffUIR. As illustrated in Figure 7 (Right), distinct transport behaviors are observed:

- **DiffUIR (Curved Trajectory):** DiffUIR exhibits a logarithmic growth in performance, where Cosine Similarity gradually increases with more steps. This pattern indicates that DiffUIR learns a *curved* stochastic differential equation (SDE) path. The Euler solver requires dense discretization to approximate the curve's tangent, resulting in high latency. The pre-similarity (feature consistency) also slowly accumulates, confirming the misalignment of manifolds in the intermediate states.

- **UDBM (Rectified Linear Trajectory):** In contrast, UDBM achieves its peak performance at a single step ($N = 1$). Increasing $N$ yields negligible gains. This confirms that our Shared Bridge Term effectively aligns the heterogeneous degradation manifolds into a unified latent space, and the Relaxed Diffusion Bridge rectifies the flow into a straight geodesic. For in-distribution data, the learned transport dynamics are nearly geometrically exact, rendering multi-step integration redundant. However, for out-of-distribution samples (as shown in Table 15), slight curvature may emerge due to manifold mismatch, where multi-step integration can further refine the restoration.

## G. Uncertainty Estimation Strategies

In this section, we provide detailed formulations for the three uncertainty estimation strategies in our paper: Residual-based Estimation (Ours), BayesCap, and Heteroscedastic Regression. Consistent with the main text, we denote the low-quality input as $\mathbf{x}_{lq}$, the high-quality target as $\mathbf{x}_{hq}$, and the network as $\psi(\cdot)$. The architecture of $\psi(\cdot)$ is shown in Appendix I.1. The pixel-wise uncertainty map is generally represented as $\mathbf{u} = \mathcal{U}(\psi(\mathbf{x}_{lq}))$. All the methods are trained on the specific datasets.

### G.1. BayesCap

The method is based on BayesCap(Upadhyay et al., 2022), designed to estimate uncertainty for frozen pre-trained models.

$\psi(\cdot)$ represents a frozen deterministic restoration backbone. We introduce an additional Bayesian identity mapping network $\Omega(\cdot)$, which takes the output of $\psi(\cdot)$ as input. $\psi(\cdot)$ is performed using Adam (lr $= 8 \times 10^{-5}$) minimizing the $L_1$ Loss with 600k iterations, while $\Omega(\cdot)$ is trained with 100k iterations. The implementation details of $\Omega(\cdot)$ are the same as BayesCap. The composite uncertainty estimation process is:

$$\{\tilde{\mathbf{x}}, \tilde{\alpha}, \tilde{\beta}\} = \Omega(\psi(\mathbf{x}_{lq})). \tag{111}$$

Instead of a point estimate, $\Omega$ predicts the parameters of a Heteroscedastic Generalized Gaussian distribution to model the target $\mathbf{x}_{hq}$. $\tilde{\alpha}$ represents the scale, and $\tilde{\beta}$ represents the shape. The uncertainty $\mathbf{u}$ is derived analytically:

$$\mathbf{u} = \mathcal{U}_{\text{bayes}}(\psi(\mathbf{x}_{lq})) = Sigmoid(\frac{\tilde{\alpha}^2 \Gamma(3/\tilde{\beta})}{\Gamma(1/\tilde{\beta})}), \tag{112}$$

where $\Gamma(\cdot)$ is the Gamma function.

### G.2. Heteroscedastic Regression

This method follows the framework proposed by previous work(Kendall & Gal, 2017) for learning uncertainty.

In this approach, the network $\psi(\cdot)$ is modified to have a dual-head architecture. The network is performed using Adam (lr $= 8 \times 10^{-5}$) with 600k iterations. For a given input $\mathbf{x}_{lq}$, it simultaneously predicts the restored image mean $\hat{\mathbf{x}}$ and a log-variance map $\mathbf{s}$:

$$[\hat{\mathbf{x}}, \mathbf{s}] = \psi(\mathbf{x}_{lq}). \tag{113}$$

The model is trained by minimizing the Negative Log-Likelihood (NLL) of a Gaussian distribution:

$$\mathcal{L}_{\text{NLL}} = \frac{1}{N} \sum_i \left( \frac{1}{2} \exp(-\mathbf{s}_i) \|\mathbf{x}_{hq,i} - \hat{\mathbf{x}}_i\|^2 + \frac{1}{2} \mathbf{s}_i \right). \tag{114}$$

The uncertainty map is derived as the exponential of the learned log-variance:

$$\mathbf{u} = \mathcal{U}_{\text{nll}}(\psi(\mathbf{x}_{lq})) = \exp(\mathbf{s}). \tag{115}$$

### G.3. Residual-based Estimation

This strategy is adopted from (Zhang et al., 2025) and serves as the uncertainty estimator in our UDBM framework.

In this setting, $\psi(\cdot)$ serves as a lightweight auxiliary restoration network. It produces a preliminary restoration $\hat{\mathbf{x}}_{hq} = \psi(\mathbf{x}_{lq})$. $\psi(\cdot)$ is performed using Adam (lr $= 8 \times 10^{-5}$) minimizing the $L_1$ Loss with 600k iterations. The uncertainty map $\mathbf{u}$ is explicitly defined as the absolute residual between this preliminary estimate and the degraded input:

$$\mathbf{u} = \mathcal{U}_{\text{res}}(\psi(\mathbf{x}_{lq})) = \frac{1}{2} |\psi(\mathbf{x}_{lq}) - \mathbf{x}_{hq}|, \tag{116}$$

---

**Algorithm 1** Training of UDBM ($\mathbf{x}_0$-prediction)

---

**Input:** $\mathcal{D}_{hq}, \mathcal{D}_{lq}$, Estimator $\psi$, Network $\mathcal{N}_\theta$. **Params:** $\lambda_b, \pi_{\text{OT}}, \pi_{\text{EOT}}$.
**Freeze** $\psi$.
**repeat**
    $\mathbf{x}_{hq}, \mathbf{x}_{lq} \sim \mathcal{D}; \; t \sim \mathcal{U}(0,1); \; \boldsymbol{\epsilon} \sim \mathcal{N}(\mathbf{0}, \mathbf{I})$ {Sample data and noise}
    $\mathbf{u} \leftarrow \mathcal{U}_{\text{res}}(\psi(\mathbf{x}_{lq}))$ {Estimate pixel-wise uncertainty}
    *// Compute schedule: path $(\alpha, \gamma)$ and noise $(\beta)$ coeff.*
    $\pi \leftarrow (1-\mathbf{u})\pi_{\text{OT}} + \mathbf{u}\pi_{\text{EOT}}; \quad \beta_t \leftarrow \lambda_b(\mathbf{I}+\mathbf{u})t(1-t) + (\mathbf{I}+\mathbf{u})t^2$
    $\alpha_t \leftarrow t^\pi/(t^\pi + (1-t)^\pi); \quad \gamma_t \leftarrow 1 - \alpha_t$ {Convexity constraint}
    $\mathbf{x}_t \leftarrow \alpha_t \mathbf{x}_{lq} + \gamma_t \mathbf{x}_{hq} + \beta_t \boldsymbol{\epsilon}$ {Construct relaxed bridge state}
    $\hat{\mathbf{x}}_0 \leftarrow \mathcal{N}_\theta(\mathbf{x}_t, t, \mathbf{u})$ {Network prediction}
    $\theta \leftarrow \theta - \eta\nabla_\theta\|\hat{\mathbf{x}}_0 - \mathbf{x}_{hq}\|_1$ {Optimize reconstruction loss}
**until** converged

---

where $|\cdot|$ denotes the element-wise absolute difference.

The rationale for using the residual as an uncertainty proxy is straightforward: the magnitude of the modification reflects the difficulty of restoration.

- **Easy Regions (Flat):** In smooth areas (e.g., sky or walls), the low-quality input retains most structural information and is already close to the ground truth. The restoration network makes minimal changes, resulting in a small residual, which correctly signals low uncertainty.

- **Hard Regions (Texture/Edge):** Conversely, complex details (e.g., hair or edges) suffer severe information loss during degradation. The network must significantly "hallucinate" or reconstruct these missing high-frequency details, causing its output to deviate substantially from the blurry input. This large residual effectively captures the high uncertainty inherent in these difficult regions.

While this approach is heuristic, its validity is supported by the findings in BayesCap (Upadhyay et al., 2022).

- **Correlation with Error:** BayesCap explicitly demonstrates that aleatoric uncertainty is often estimated by approximating the per-pixel residuals. Furthermore, their experiments show that a well-calibrated uncertainty map must be highly correlated with the reconstruction error.

- **Justification for Proxy:** BayesCap validates that regions with high error are synonymous with regions of high uncertainty. Since the auxiliary network $\psi(\cdot)$ is trained to restore the image, the residual $|\psi(\mathbf{x}_{lq}) - \mathbf{x}_{lq}|$ is a direct approximation of the restoration error (and thus the information loss). By leveraging the correlation proven in BayesCap, we can effectively use this residual as a computationally efficient proxy for pixel-wise uncertainty.

## H. Algorithm Details

We provide the compact pseudo-code for UDBM. Algorithm 1 details the training, while Algorithm 2 presents the unified sampling (supporting DDPM, DDIM, and single-step mapping).

## I. Experimental Details

### I.1. Network Architecture

In this section, we provide a detailed specification of the network architectures employed in our proposed method. To achieve a balance between computational efficiency and restoration performance, we provide various versions of the networks. Our USBM consists of two networks: $\psi(\cdot)$ for uncertainty prediction, and $\mathcal{N}_{\text{den}}(\cdot)$ for denoising.

---

**Algorithm 2** Unified Inference (DDPM/DDIM Compatible)

---

**Input:** $\mathbf{x}_{lq}, \mathcal{N}_\theta, \psi$. **Params:** Steps $N$ (default 1), $\eta \in [0, 1]$.
$\mathbf{u} \leftarrow \mathcal{U}_{\text{res}}(\psi(\mathbf{x}_{lq})); \quad \tau \leftarrow [1, \ldots, 0]$ {Get uncertainty & time steps}
*// 1. Relaxed Terminal Initialization ($t = 1$)*
$\beta_1 \leftarrow \mathbf{I} + \mathbf{u}; \quad \mathbf{x}_{\tau_N} \leftarrow \mathbf{x}_{lq} + \beta_1 \mathcal{N}(\mathbf{0}, \mathbf{I})$ {Stochastic sampling}
*// 2. Reverse Process*
**for** $i = N$ **to** $1$ **do**
    $t \leftarrow \tau_i; \; s \leftarrow \tau_{i-1}$
    $\hat{\mathbf{x}}_0 \leftarrow \mathcal{N}_\theta(\mathbf{x}_t, t, \mathbf{u})$ {Predict $\mathbf{x}_0$ from current state}
    $\boldsymbol{\epsilon}_{\text{pred}} \leftarrow (\mathbf{x}_t - \alpha_t \mathbf{x}_{lq} - \gamma_t \hat{\mathbf{x}}_0) \oslash \beta_t$ {Derive noise from prediction}
    *// Compute posterior $\tilde{\sigma}_t$ and update direction*
    $\tilde{\sigma}_t \leftarrow \eta \sqrt{\beta_s^2(\beta_t^2 - \beta_s^2)/\beta_t^2}$ {$\eta = 0$ for DDIM, $\eta = 1$ for DDPM}
    $\mathbf{x}_{\text{mean}} \leftarrow \alpha_s \mathbf{x}_{lq} + \gamma_s \hat{\mathbf{x}}_0 + \sqrt{\beta_s^2 - \tilde{\sigma}_t^2} \boldsymbol{\epsilon}_{\text{pred}}$
    **if** $s > 0$ **then**
        $\mathbf{x}_s \leftarrow \mathbf{x}_{\text{mean}} + \tilde{\sigma}_t \mathcal{N}(\mathbf{0}, \mathbf{I})$ {Stochastic update}
    **else**
        $\mathbf{x}_0 \leftarrow \hat{\mathbf{x}}_0$ {At $s = 0$, $\beta_0 = 0$, directly output prediction}
    **end if**
**end for**
**Return** $\text{clip}(\mathbf{x}_0, 0, 1)$

---

## Backbone Architecture

Following the design of Refusion (Luo et al., 2023c), both $\psi(\cdot)$ and $\mathcal{N}_{\text{den}}(\cdot)$ utilize a U-shaped encoder-decoder structure built upon Modified Nonlinear Activation Free Blocks (NAFBlocks).

**The NAFBlock.** As detailed in Refusion, the standard NAFBlock replaces computationally heavy nonlinear activation functions with a **SimpleGate** mechanism. Given feature map $X$ split into $X_1, X_2$, it computes:

$$\text{SimpleGate}(X_1, X_2) = X_1 X_2 \tag{117}$$

**Time-Conditioning.** Since our method operates within a probabilistic diffusion framework, we inject time-step information $t$ into the network. The time embedding is processed by a Multi-Layer Perceptron (MLP) to generate channel-wise scale ($\gamma$) and shift ($\beta$) parameters, which modulate the feature maps within the NAFBlock, ensuring adaptivity to the noise level at each diffusion step.

## Network Configurations

To thoroughly evaluate our method across different computational budgets, we designed a continuous family of network variants, denoted as $v1$ through $v4$. These variants are derived by scaling two primary hyperparameters:

1. **Base Channel Width ($C$):** Denoted as dimensions of the first layer in the network.

2. **Bottleneck Depth:** The number of NAFBlocks in the different layers in the network.

Table 10 details the specific hyperparameters, parameter counts, and computational costs (FLOPs) for these variants.

## Model Ensembles (S, M, L)

We instantiate the $\psi(\cdot)$ and $\mathcal{N}_{\text{den}}(\cdot)$ using combinations of the variants defined above.

We propose three model versions: Small (S), Medium (M), and Large (L). The detailed composition is presented in Table 11.

*Table 10.* **Specifications of Network Variants.** The Depth Configuration refers to the number of blocks in the bottleneck of the $[1, 1, 1, N]$ structure. FLOPs are calculated on a $256 \times 256$ input patch.

| Variant ID | Base Width ($C$) | Bottleneck Depth ($N$) | Params (M) | FLOPs (G) |
|---|---|---|---|---|
| **v1** | 48 | $[1, 1, 1, 28]$ | 43.14 | 35.73 |
| **v2** | 32 | $[1, 1, 1, 16]$ | 12.88 | 11.14 |
| **v3** | 24 | $[1, 1, 1, 28]$ | 10.88 | 9.10 |
| **v4** | 16 | $[1, 1, 1, 28]$ | 3.26 | 2.88 |
| **v5** | 16 | $[1, 1, 1, 10]$ | 2.45 | 2.26 |

*Table 11.* **Configuration of Proposed Model Versions (S/M/L).** The composition column denotes $\psi(\cdot) + \mathcal{N}_{\text{den}}(\cdot)$. Total Params and FLOPs are the sum of both networks.

| Model Version | Denoising $\mathcal{N}_{\text{den}}(\cdot)$ | Uncertainty $\psi(\cdot)$ | Composition | Total Params (M) | Total FLOPs (G) |
|---|---|---|---|---|---|
| **UDBM-S (Small)** | **v4** (16, 28) | **v5** (16, 10) | $v3 + v4$ | 5.71 | 5.14 |
| **UDBM-M (Medium)** | **v3** (24, 28) | **v4** (16, 28) | $v2 + v3$ | 14.14 | 11.98 |
| **UDBM-L (Large)** | **v1** (48, 28) | **v2** (32,16) | $v1 + v2$ | 56.02 | 46.85 |

## I.2. Training Details

To ensure a rigorous and fair comparison, we strictly standardized the training methods for all competing methods across both Task-Specific and All-in-One settings. Specifically, with the exception noted below, all comparison methods were trained from scratch using the identical training and testing splits. For each method, we utilized the officially released open-source codes. We adhered strictly to the original training strategies, including optimization schedules, augmentations, and hyperparameters.

**Note:** Following previous work (Zheng et al., 2024), the results of Task-Specific settings in Tab. 1 are obtained from separate models trained independently on each corresponding dataset.

## I.3. Dataset Details

We evaluate our model on five image restoration tasks. The specific configurations for training and testing are detailed below:

- **Image Deraining:** We utilize the merged datasets (Jiang et al., 2020; Wang et al., 2022b) which cover diverse rain streaks and densities. The dataset consists of 13,712 pairs for training and 4,298 pairs for testing. For zero-shot generalization, we use the real-world *Practical* (Yang et al., 2017) dataset.

- **Low-light Enhancement:** The LOL (Wei et al., 2018) dataset is used as the primary benchmark, comprising 485 training pairs and 15 testing pairs. We further evaluate generalization on the real-world MEF (Ma et al., 2015), NPE (Wang et al., 2013), and DICM (Lee et al., 2013) datasets.

- **Image Desnowing:** We employ the Snow100K (Liu et al., 2018) dataset, containing 50,000 training pairs and 50,000 testing pairs. Evaluations are performed on the Snow100K-S and Snow100K-L subsets, as well as real-world snow images.

- **Image Dehazing:** We adopt the Outdoor Training Set (OTS) from the RESIDE (Li et al., 2018) dataset, which contains 313,950 training images and 500 testing images collected in real-world outdoor environments.

- **Image Deblurring:** The GoPro (Nah et al., 2017) dataset serves as the benchmark with 2,103 training pairs and 1,111 testing pairs. Zero-shot generalization is conducted on the HIDE (Shen et al., 2019), RealBlur-J (Rim et al., 2020), and RealBlur-R (Rim et al., 2020) datasets.

# J. Additional Experiments Results

## J.1. More Quantitative Comparison Results

*Table 12.* Quantitative comparison of NIQE on Real Rain, Real Dark, and Real Snow datasets.

| Method | Real Rain ↓ | Real Dark ↓ | Real Snow ↓ |
|---|---|---|---|
| Prompt-IR | 3.52 | 3.31 | 2.79 |
| DiffUIR | 3.38 | 3.14 | 2.74 |
| AdaIR | 3.40 | 3.24 | 2.78 |
| BioIR | 3.42 | 3.25 | 2.76 |
| MOCE-IR | 3.45 | 3.18 | 2.75 |
| HOGformer | 3.31 | 3.08 | 2.69 |
| **UDBM-L** | **3.26** | **3.07** | **2.63** |

*Table 13.* Quantitative comparison on the CDD11 dataset. We report NIQE metrics (higher is better) for evaluation.

| Method | Single Degradation | | | | Double Degradations | | | | | Triple Degradations | | Average |
|---|---|---|---|---|---|---|---|---|---|---|---|---|
| | L | H | R | S | L+H | L+R | L+S | H+R | H+S | L+H+R | L+H+S | |
| DiffUIR | 4.18 | 4.17 | 4.02 | 4.32 | 6.70 | 6.19 | 6.72 | **4.42** | 4.10 | 6.07 | 5.87 | 5.16 |
| AdaIR | 4.23 | 4.21 | 4.11 | 4.39 | 6.82 | 6.15 | 6.78 | 4.63 | 4.18 | 6.11 | 5.96 | 5.23 |
| BioIR | 4.15 | 4.16 | 4.09 | 4.38 | 6.87 | 6.24 | 6.80 | 4.52 | 4.14 | 6.13 | 5.90 | 5.22 |
| MOCE-IR | 4.21 | 4.22 | 4.06 | 4.41 | 6.76 | 6.23 | 6.85 | 4.56 | 4.19 | 6.15 | 5.95 | 5.24 |
| HOGformer | 4.16 | 4.13 | 4.01 | 4.32 | 6.70 | 6.28 | 6.80 | 4.51 | 4.16 | 6.03 | 5.79 | 5.17 |
| **UDBM-L** | **4.12** | **4.11** | **3.98** | **4.28** | **6.63** | **5.85** | **6.57** | 4.43 | **4.11** | **5.98** | **5.85** | **5.08** |

## J.2. Impact of Inference Steps: In-Distribution vs. Out-of-Distribution

In the main paper, we demonstrated that UDBM achieves state-of-the-art performance using a single inference step ($N = 1$) for the All-in-One benchmark. Here, we provide a deeper analysis of the impact of sampling steps, distinguishing between **In-Distribution (ID)** and **Out-of-Distribution (OOD)** scenarios. This distinction reveals the geometric properties of our learned transport trajectory under different domains.

**In-Distribution (ID) Saturation.** Table 14 summarizes the results on the five tasks seen during training (ID). As discussed in Appendix F.4, our proposed path schedule (Eq. (11a)) explicitly rectifies the transport mean into a linear geodesic ($\gamma_t = \mathbf{I} - \alpha_t$) during training. Theoretically, for a perfectly straight trajectory, the truncation error of the first-order Euler solver is zero. Consequently, increasing $N$ from 1 to 5 does not yield performance gains and may even cause marginal degradation due to the accumulation of neural network approximation errors at each step.

**Out-of-Distribution (OOD) Refinement.** However, the behavior changes when applying the model to unseen complex degradations. We conducted an ablation on the **CDD11** dataset (unseen composite degradations) with inference steps $N \in \{1, 2, 4, 8, 10\}$. As shown in Table 15, unlike the ID scenario, increasing the inference steps on OOD data may lead to a improvement in perceptual quality.

This phenomenon can be attributed to Manifold Shift. While the transport trajectory is perfectly rectified for the training distribution, unseen degradations may lie slightly off the learned linear manifold, introducing non-zero geometric curvature to the optimal transport path. In this context, a single Euler step ($N = 1$) might result in a slight overshoot or undershoot. Increasing the sampling steps ($N > 1$) allows the ODE solver to perform fine-grained corrections along the curved trajectory, thereby recovering better structural details and reducing artifacts in challenging unseen scenarios.

## J.3. Sensitivity and Robustness Analysis of Uncertainty Guidance

In the proposed UDBM framework, the pixel-wise uncertainty map $\mathbf{u}$ acts as a control signal steering the stochastic transport dynamics. To evaluate the behavior of UDBM with respect to $\mathbf{u}$, we conduct a comprehensive sensitivity and robustness analysis. Specifically, we investigate how the restoration quality responds to the variation of uncertainty.

We visualize the comparative results in Figure 10. The experiments are categorized into four groups:

*Table 14.* Impact of inference steps on ID tasks. Increasing steps leads to saturation, confirming the geometric linearity for known domains.

| Steps | Rain | | Low-light | | Snow | | Haze | | Blur | | Average | |
|---|---|---|---|---|---|---|---|---|---|---|---|---|
| | P↑ | S↑ | P↑ | S↑ | P↑ | S↑ | P↑ | S↑ | P↑ | S↑ | P↑ | S↑ |
| 1 | **32.06** | 0.917 | 26.55 | **0.915** | 34.00 | **0.943** | **39.88** | **0.996** | 30.58 | **0.895** | **32.61** | **0.933** |
| 2 | 32.17 | **0.918** | 26.58 | 0.914 | 34.01 | **0.943** | 39.17 | **0.996** | **30.61** | **0.895** | 32.51 | **0.933** |
| 5 | 32.16 | **0.918** | **26.59** | 0.915 | 33.98 | **0.943** | 39.03 | **0.996** | 30.60 | **0.895** | 32.47 | **0.933** |

*Table 15.* Impact of inference steps on OOD data (CDD11 dataset). Unlike ID tasks, multi-step inference may provides perceptual gains for unseen degradations.

| Metric | Step=1 | Step=2 | Step=4 | Step=8 | Step=10 |
|---|---|---|---|---|---|
| **LIQE** ↑ | 2.898 | 2.912 | 2.925 | **2.931** | 2.930 |
| **HyperIQA** ↑ | 0.539 | 0.544 | 0.551 | 0.553 | **0.554** |

**1. Robustness to Prediction Noise (Additive Noise on u).** We inject Gaussian noise into the predicted uncertainty map: $\mathbf{u} = \text{clip}(\mathcal{U}(\psi(\mathbf{x}_{lq})) + k\boldsymbol{\epsilon})$, where $\boldsymbol{\epsilon} \sim \mathcal{N}(0, \mathbf{I})$ and $k \in \{1, 5, 10, 50\}$. As illustrated in the first row of Figure 10, UDBM exhibits stability across varying noise levels.

**2. Robustness to Input Corruption.** We perturb the input to the uncertainty estimator $\psi(\cdot)$ to test its stability: $\mathbf{u} = \mathcal{U}(\psi(\mathbf{x}_{lq} + k\boldsymbol{\epsilon}))$, with $k \in \{0.5, 1.0, 2.0, 5.0\}$. As displayed in the second row, the restoration remains stable under mild input corruption ($k \leq 2.0$). However, as the noise intensity increases ($k = 5.0$), structural deterioration and mode collapse are observed. This performance drop is attributed to the fact that the lightweight estimator $\psi(\cdot)$ was not explicitly optimized for noise robustness during training, leading to erroneous guidance signals under severe corruption.

we further provide quantitative comparison results as shown in Tab. 16. We injected Gaussian noise into the predicted uncertainty map and evaluated the result in low-light conditions without retraining. UDBM is robust to mild-to-moderate inaccuracy, while severe corruption of the uncertainty map degrades performance progressively rather than causing catastrophic failure.

*Table 16.* Quantitative evaluation of robustness to prediction noise on the uncertainty map.

| $\sigma$ | PSNR ↑ | SSIM ↑ | NIQE ↓ |
|---|---|---|---|
| 0.1 | 26.58 | 0.915 | 5.185 |
| 0.3 | 25.43 | 0.912 | 5.177 |
| 0.5 | 25.41 | 0.914 | 5.168 |
| 0.7 | 24.32 | 0.909 | 5.169 |
| 1.0 | 23.57 | 0.907 | 5.138 |

**3. Controllability via Uncertainty Scaling.** To verify the correlation between uncertainty magnitude and restoration strength, we globally scale the predicted map: $\mathbf{u} = \text{clip}(k \cdot \mathcal{U}(\psi(\mathbf{x}_{lq})))$, with $k \in \{0.5, 2.0, 5.0, 10.0\}$. The third row of Figure 10 reveals a monotonic relationship between $k$ and the extent of degradation removal. When $k$ is attenuated ($k = 0.5$), the process yields incomplete restoration with residual degradation artifacts. Conversely, amplifying $k$ enhances the removal of degradation. However, excessive scaling introduces over-processing artifacts, suggesting that $\mathbf{u}$ must fall within a reasonable range to ensure fidelity.

**4. Necessity of Spatial Adaptivity (Fixed Uncertainty).** We replace the adaptive, pixel-wise uncertainty map with a spatially invariant constant: $\mathbf{u} = k\mathbf{I}$, where $k \in \{0.0, 0.3, 0.5, 1.0\}$. As shown in the fourth row, a zero-uncertainty condition ($k = 0$) results in a near-identity mapping, where the degradation remains virtually untouched. As $k$ increases, restoration effects emerge; however, excessive fixed values again lead to artifacts.

**5. Performance under extreme degradation.**

We further evaluate UDBM on the Dense-Haze dataset without retraining to analyze its robustness under extreme degradation and domain shifts. Since this dataset differs substantially from our training distribution and contains exceptionally heavy haze, the auxiliary uncertainty estimation becomes less reliable, leading to sub-optimal restoration quality as shown in Fig. 8.

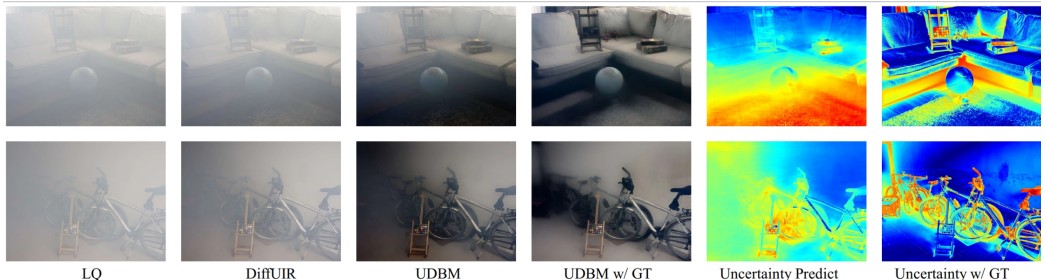

*Figure 8.* Visualization of UDBM and DiffUIR in Dense-Haze dataset.

Notably, we observe a similar performance degradation for DiffUIR under identical conditions. While our UDBM with predicted uncertainty performs better. We further substitute the predicted uncertainty map with the ground-truth absolute residual. This substitution yields noticeable improvements in both visual quality and quantitative performance.

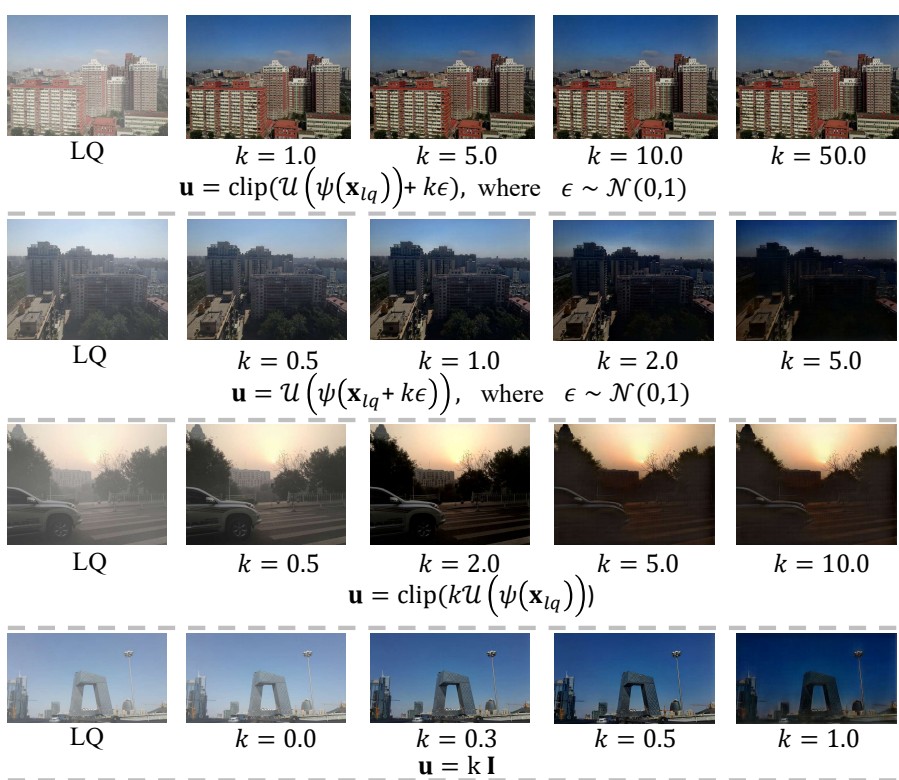

*Figure 9.* Sensitivity and Robustness Analysis of UDBM. Row 1: Additive noise on $\mathbf{u}$ ($k \in \{1, 5, 10, 50\}$). Row 2: Input noise to estimator ($k \in \{0.5, 1.0, 2.0, 5.0\}$). Row 3: Scaling $\mathbf{u}$ ($k \in \{0.5, 2.0, 5.0, 10.0\}$). Row 4: Fixed uncertainty $\mathbf{u} = k\mathbf{I}$ ($k \in \{0.0, 0.3, 0.5, 1.0\}$).

The results demonstrate that the uncertainty map $\mathbf{u}$ effectively regulates the transport dynamics of UDBM. Crucially, the framework is not strictly reliant on perfectly precise uncertainty estimation, demonstrating robustness against mild-to-moderate perturbations in the guidance signal. However, its overall effectiveness is bounded by the reliability of the estimator, which explains the performance degradation observed under extreme out-of-distribution shifts.

## J.4. More Visualization Comparison

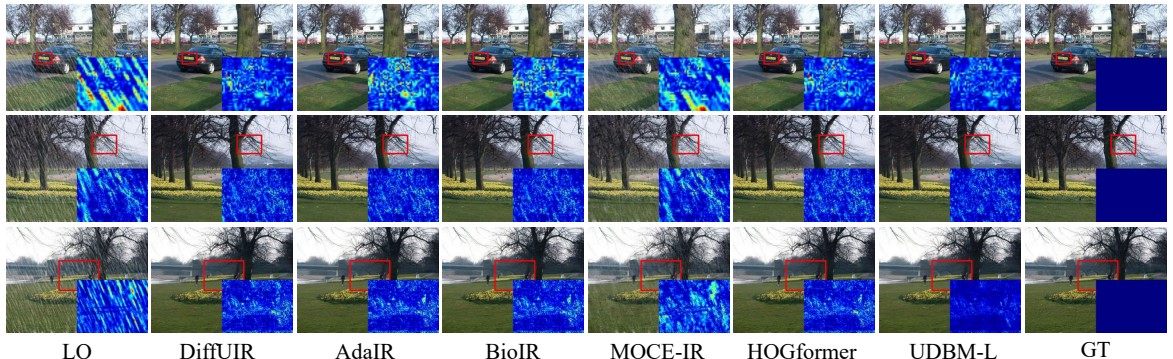

LQ     DiffUIR     AdaIR     BioIR     MOCE-IR     HOGformer     UDBM-L     GT

*Figure 10.* Visual comparison of rain degradation.

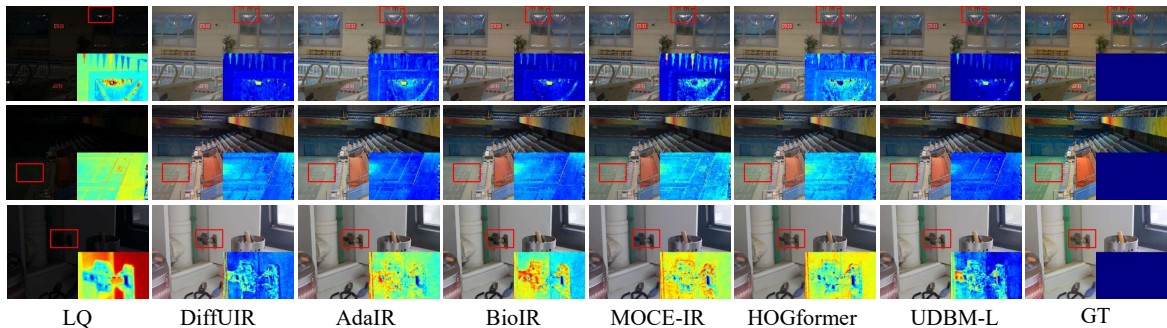

LQ     DiffUIR     AdaIR     BioIR     MOCE-IR     HOGformer     UDBM-L     GT

*Figure 11.* Visual comparison of low-light degradation.

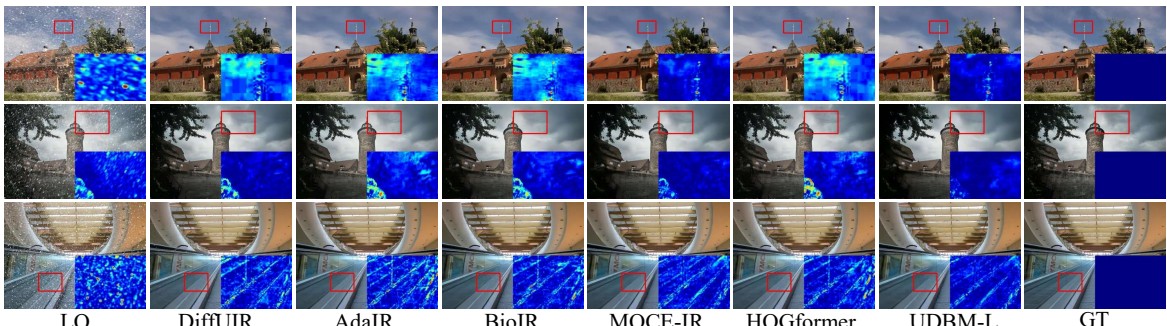

LQ     DiffUIR     AdaIR     BioIR     MOCE-IR     HOGformer     UDBM-L     GT

*Figure 12.* Visual comparison of snow degradation.

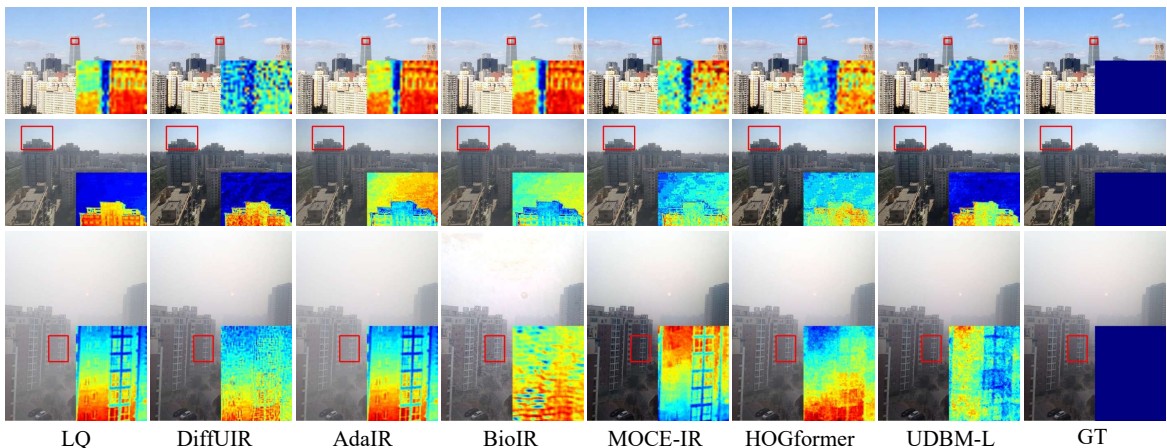

LQ     DiffUIR     AdaIR     BioIR     MOCE-IR     HOGformer     UDBM-L     GT

*Figure 13.* Visual comparison of haze degradation.

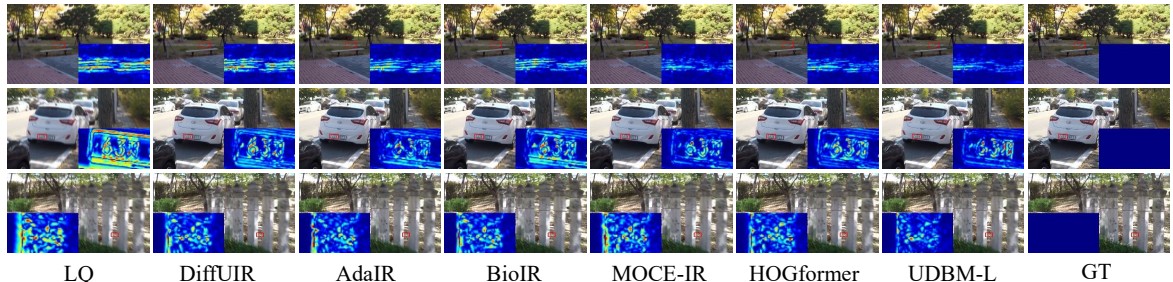

LQ      DiffUIR      AdaIR      BioIR      MOCE-IR      HOGformer      UDBM-L      GT

*Figure 14.* Visual comparison of blur degradation.

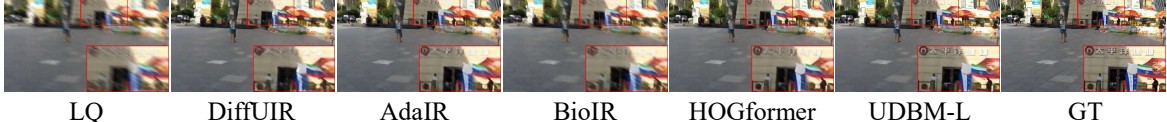

LQ      DiffUIR      AdaIR      BioIR      HOGformer      UDBM-L      GT

*Figure 15.* Visual comparison in the real blur scenario.

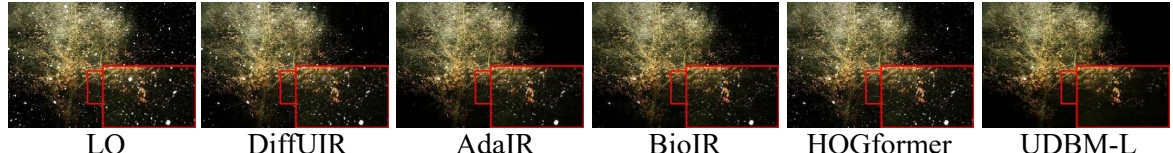

LQ      DiffUIR      AdaIR      BioIR      HOGformer      UDBM-L

*Figure 16.* Visual comparison in the real snow scenario.

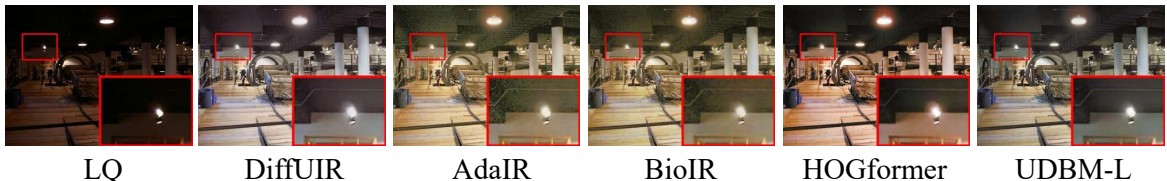

LQ      DiffUIR      AdaIR      BioIR      HOGformer      UDBM-L

*Figure 17.* Visual comparison in the real low-light scenario.

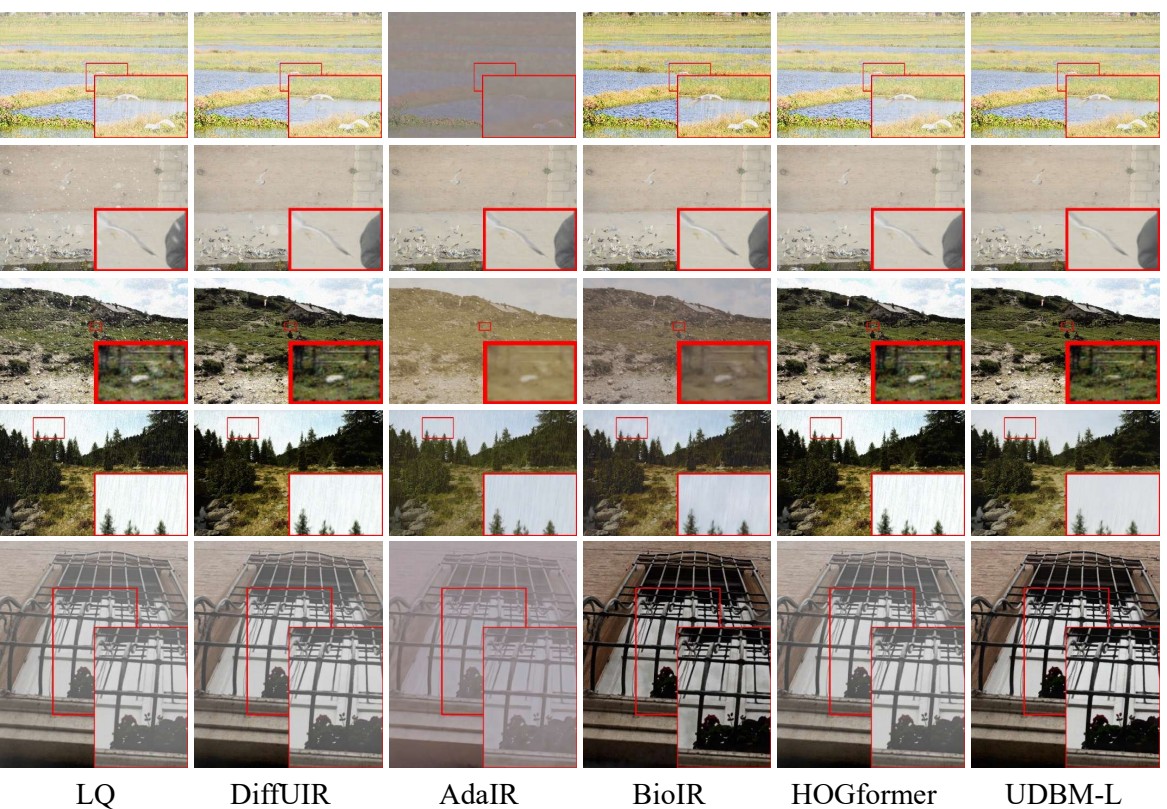

LQ          DiffUIR          AdaIR          BioIR          HOGformer          UDBM-L

*Figure 18.* Visual comparison of the complex degradations in the CDD11 dataset.

