# OpenReview forum: "Unifying Heterogeneous Degradations: Uncertainty-Aware Diffusion Bridge Model for All-in-One Image Restoration"
_ICML.cc/2026/Conference — ICML 2026 regular_

### Official Review · Reviewer_tNqF · 2026-02-19

**Soundness:** 3
**Presentation:** 3
**Significance:** 3
**Originality:** 2
**Overall Recommendation:** 5
**Confidence:** 4

**Summary:**

This paper proposes an Uncertainty-Aware Diffusion Bridge Model (UDBM) for All-in-One image restoration. The authors formulate the restoration process as a stochastic transport problem and introduce a relaxed terminal constraint to mitigate drift singularity. To handle varying degradation severities across different regions and tasks, the model utilizes a dual modulation strategy (path and noise schedules) guided by a pixel-wise uncertainty map. The method demonstrates strong performance across various image restoration benchmarks with efficient single-step inference.

**Compliance With Llm Reviewing Policy:**

Affirmed.

**Final Justification:**

The rebuttal has addressed my concerns.

**Key Questions For Authors:**

1. The extraction of a residual map via an auxiliary network to guide the main restoration process shares conceptual similarities with DA-RCOT. Could you please cite DA-RCOT/BaryIR and briefly discuss the differences (e.g., diffusion bridge vs. optimal transport) in the rebuttal and the final version?

2. Could you clarify if the inference time and computational cost (FLOPs) reported in Table 1 explicitly include the forward pass of the auxiliary network $\psi(\cdot)$ used to compute the uncertainty map?

3. How sensitive is the main diffusion model to the quality of the initial residual map? If the auxiliary network $\psi(\cdot)$ performs poorly on an out-of-distribution or extreme degradation, how gracefully does the UDBM degrade?

**Limitations:**

No. The authors provide a general impact statement discussing broader societal implications, but it would be beneficial to discuss the specific technical limitations of their method. For instance, discussing failure cases where the initial residual estimation from the auxiliary network is inaccurate, and how that impacts the final image generation, would provide a more comprehensive view of the model's robustness.

**Strengths And Weaknesses:**

### Strengths:

* The mathematical motivation is sound and interesting. Softening the terminal constraint to address drift singularity is well-reasoned, and the theoretical connection to entropy-regularized optimal transport (EOT) provides a solid foundation for their design choices.

* Modulating the diffusion schedules spatially based on local degradation severity is a practical and effective strategy for all-in-one restoration, leading to impressive empirical results.

### Weaknesses:

* In Appendix G.3, the paper details that the "uncertainty map" is calculated as the absolute residual between the degraded input and a preliminary restoration from an auxiliary network $\psi(\cdot)$ ($u = \frac{1}{2}|\psi(x_{lq}) - x_{lq}|$). Using a residual map as a degradation-specific cue to condition the transport process is conceptually very similar to recent residual-based methods, e.g., DA-RCOT (Degradation-Aware Residual-Conditioned Optimal Transport) [1] and BaryIR [2]. The paper would greatly benefit from discussing DA-RCOT and BaryIR and briefly discussing how UDBM's residual-guided diffusion bridge differs from or builds upon the residual-conditioned approach.

* While using the residual as a proxy for uncertainty is empirically effective and computationally efficient, it would be helpful to briefly clarify in the main text that this guidance signal is essentially a deterministic residual error map rather than a probabilistic variance. This would make the implementation details more transparent to the readers.

* The framework requires a two-pass process during inference: first running the auxiliary network $\psi(\cdot)$ to obtain the residual, and then running the main diffusion model. It is slightly unclear if the computational overhead of this preliminary step is fully accounted for in the main comparisons.


[1]  “Degradation-Aware Residual-Conditioned Optimal Transport for Unified Image Restoration”, IEEE TPAMI 2025.

[2]  “Learning Continuous Wasserstein Barycenter Space for Generalized All-in-one Image Restoration”, IEEE TPAMI 2026.

---

> ### Author Rebuttal · Authors · 2026-03-30
>
> We sincerely thank the reviewer tNqF for the thoughtful comments. We clarify the relevant points below.
>
>
>
>
>
>
>
>
> **A4.1 Relation to DA-RCOT and BaryIR. (Weakness 1, Question 1)**
> Thank you for pointing out the relevance of these insightful works. DA-RCOT and BaryIR provide valuable perspectives for thinking about degradation-aware guidance in AiOIR. We agree that our uncertainty map is conceptually related in the broad sense that all three methods exploit residual-derived signals as degradation-aware cues. In particular, DA-RCOT demonstrates that residual information serves as an important cue for capturing continuous and spatially non-uniform degradations. This provides strong justification for UDBM's choice to employ the residual as an uncertainty proxy.
>
> However, the role of the residual signal is fundamentally different. DA-RCOT uses the transport residual to condition both the OT cost and the OT map through a two-pass residual-guided design, while BaryIR uses residual embeddings to model degradation-specific residual subspaces with a degradation-agnostic barycenter space. In contrast, in UDBM, the residual-like signal is interpreted as a pixel-wise uncertainty proxy that modulates the diffusion-bridge transport process through the relaxed terminal variance and the adaptive path/noise schedules.
>
> We will add a discussion in the revised manuscript to better clarify the relationships and distinctions.
>
>
> **A4.2 Residual proxy and probabilistic variance. (Weakness 2)**
>
> We agree that this point should be stated more clearly.
>
> In our implementation, the residual-based uncertainty is a proxy signal, rather than a probabilistic variance estimate. We adopt it because it is simple, efficient, and effective in practice.
>
> At the same time, UDBM is not restricted to this choice. Our ablations already show that the framework also works with more explicitly probabilistic estimators such as BayesCap and heteroscedastic regression, although the residual-based proxy offers the best trade-off in our current setting.
>
> We will clarify this distinction in the main text and appendix.
>
>
> **A4.3 Computational overhead of the auxiliary network (Weakness 3, Question 2)**
>
>  Yes, the parameter count, FLOPs, and runtime reported in Table 1 all include the auxiliary uncertainty-estimation network. In Appendix I.1, we already report the configurations of both the denoising backbone and the auxiliary network, together with the cost separately. Since the auxiliary network is lightweight, the additional overhead is small compared with the full UDBM cost. We will make this accounting more explicit in the main experimental section to avoid ambiguity.
>
>
> **A4.4 Sensitivity analysis of the main network to the residual map (Question 3, Limitation)**
>
> As already shown in Appendix J.2, adding noise to the uncertainty map, perturbing the estimator input, globally scaling the uncertainty, or replacing it with a fixed constant map all lead to progressive degradation rather than catastrophic failure.
>
> Regarding out-of-distribution (OOD) scenarios, we have reported the performance on the unseen CDD11 dataset in Table 5 and Fig. 4, where UDBM outperforms competing methods.
>
> We provide two additional experiments to analyze the robustness.
>
> (1) OOD scenario. To assess the OOD performance of the uncertainty estimator, we removed rain data from the training dataset of the uncertainty estimator while keeping the denoiser unchanged, and then conducted evaluations on the rain dataset. UDBM achieves 30.35 dB / 0.892 SSIM, compared to 32.06 dB / 0.917 SSIM when using the normal estimator. This shows that when the auxiliary network becomes unreliable due to an unseen degradation type, the UDBM still remains effective rather than collapsing.
>
> (2) Extreme degradation. We further evaluated UDBM on the highly challenging Dense-Haze dataset without retraining to analyze its behavior under extreme degradation and large domain shift. Since this dataset differs substantially from the training distribution and contains much heavier haze, the auxiliary residual/uncertainty estimation becomes less reliable, and the restoration quality of UDBM is indeed unsatisfactory in this setting. Moreover, we observed a similar issue for DiffUIR under the same conditions. To analyze the bottleneck, we replaced the predicted uncertainty (residual map) with the ground-truth absolute residual, which led to an improvement in visualization (https://anonymous.4open.science/r/ICML-12294/dense_haze.png) and the NIQE metric (shown in the table). This suggests that, in such extreme OOD scenarios, the main limitation may stem from the failure of residual/uncertainty estimation. We will explicitly discuss this limitation and possible future directions in the revised manuscript.
>
> Additional results of uncertainty map with noise are provided in A2.3.
>
> | Method | NIQE ↓ |
> |--------|-------:|
> | DiffUIR | 7.4227 |
> | UDBM (predicted uncertainty) | 6.6526 |
> | UDBM (with GT residual) | 5.9766 |

---

> > ### Author Rebuttal · Reviewer_tNqF · 2026-04-01
> >
> > The rebuttal has effectively addressed my concerns. Overall, this work is well-motivated for which I lean towards Acceptance.

---

> > > ### Author Response · Authors · 2026-04-03
> > >
> > > We sincerely thank the reviewer for the positive feedback and support.
> > >
> > > We are glad that our rebuttal has effectively addressed your concerns, and we greatly appreciate your recognition that the work is well motivated.
> > >
> > > In the revision, we will further improve the manuscript by incorporating the suggestions.

---

### Official Review · Reviewer_xsLz · 2026-03-01

**Soundness:** 2
**Presentation:** 2
**Significance:** 2
**Originality:** 3
**Overall Recommendation:** 4
**Confidence:** 4

**Summary:**

This paper points out the "imprecise mapping" problem encountered when directly using a diffusion model to solve the AiOIR task and develops a new scheduling algorithm to address this issue. The core of the proposed method lies in reformulating the AiOIR as a stochastic transport problem steered by pixel-wise uncertainty. The paper provides a detailed description of the method and experimental results. However, there may be some misinterpretations in the introduction of motivation, and the experimental results have not yet reached state-of-the-art (SOT) performance in the field.

**Compliance With Llm Reviewing Policy:**

Affirmed.

**Final Justification:**

The rebuttal addressed my concerns. I expect the authors to update the main text with results under multi-timesteps and more accurate diagrams. However, I also agree with reviewer 5LHp's concerns about experimental fairness, so I only raised the score to weakly accept.

Overall, I believe this paper meets the ICML conference's acceptance criteria due to its novelty and its solution to the "imprecise mapping" issue. While the existing experiments have minor flaws, they have already demonstrated the effectiveness of the proposed method.

**Key Questions For Authors:**

1. Why does the UDBM require the single inference step?
2. According to the appendix, the architecture of UDBM differs significantly from RDDM/DiffUIR. When the denoising model uses the same architecture and parameter count as DiffUIR, how much performance improvement does UDBM offer compared to DiffUIR?
3. The uncertainty model has a larger parameter count than the denoising model. What is the motivation behind this design? Will this lead to a significant increase in deployment/inference costs?

**Limitations:**

1. The motivation and method seem contradictory, or the description is unclear.
2. The performance of UDBM under larger inference steps is unclear.
3. The experimental results are not convincing enough (don't achieve the SoTA).

**Strengths And Weaknesses:**

### Strengths

1. This paper provides a detailed description of the experimental results and methods.

2. This paper's motivation is clear and meaningful, addressing a pressing issue in the AiOIR field.

3. The idea of using uncertainty to address the "imprecise mapping" problem in flow matching is natural and novel.

### Weaknesses

1. The motivation and method seem contradictory, or the description is unclear. In Figure 1, the authors describe the "imprecise mapping" phenomenon, noting that erroneous flows can lead to incorrect scheduling during denoising and artifacts in the final restoration. However, this error assumes multi-step inference. If UDBM uses a single inference step, the flow will be a straight line, i.e., directly from x1 to x0, and the "imprecise mapping" phenomenon does not seem to occur.

2. UDBM does not seem to require a single inference step. Using Table 10 in the appendix as an example, I suggest that the authors compare experimental results with more steps (e.g., 20 steps, 50 steps, etc.). Only multi-step inference can prove that UDBM avoids the "imprecise mapping" phenomenon seen in previous methods. To my knowledge, DiffUIR performs worse with 50 steps than with 3/5 steps in relatively easy restoration tasks. I want to know whether UDBM can maintain stable performance under more steps. Furthermore, I think the multi-step experimental results should be included in the main text, as they could directly demonstrate UDBM's advantage in avoiding "imprecise mapping."

3. The experimental performance of UDBM seems to be challenged. Data from a recent paper BDG [1] show that, even compared to RAM [2], DCPT [3], which are methods from a year ago, UDBM's performance lags significantly. It is suggested that the source of the performance gap between UDBM and the existing state-of-the-art AiOIR methods be explained.

4. LIQE is not a commonly used IQA metric. Perhaps it could provide numerical metrics for NIQE/PIQE in Tables 4 and 5.

5. This paper omitted some related works [1,2,3,4,5] in AiOIR.

---

[1] Bridging degradation discrimination and generation for universal image restoration. ICLR 2026.

[2] Restore Anything with Masks: Leveraging Mask Image Modeling for Blind All-in-One Image Restoration. ECCV 2024.

[3] Universal Image Restoration Pre-training via Degradation Classification. ICLR 2025.

[4] UniRestore: Unified Perceptual and Task-Oriented Image Restoration Model Using Diffusion Prior. CVPR 2025.

[5] FoundIR: Unleashing Million-scale Training Data to Advance Foundation Models for Image Restoration. ICCV 2025.

---

> ### Author Rebuttal · Authors · 2026-03-30
>
> We thank reviewer xsLz for the insightful feedback and time in reading our paper. We respond point by point below.
>
>
>
> **A3.1 Motivation vs. method (Weakness 1, Limitation 1)**
>
> We are sorry for any misunderstanding in our original description. The scenario is not restricted to multi-step sampling. We agree that Fig. 1(a) may have unintentionally suggested a multi-step phenomenon. Our intent was broader: “imprecise mapping” refers to erroneous conditional guidance that distorts the degraded-to-clean direction, which can occur in both one-step and multi-step restoration. We also provide a 1-step version of AutoDIR (https://anonymous.4open.science/r/ICML-12294/autodir_1step.png) to support this point.
>
> By contrast, UDBM uses pixel-wise uncertainty guidance. Because the signal reflects degradation severity rather than discrete labels, estimation errors affect restoration strength more smoothly, instead of causing hard routing or prompt-selection errors as in control-modulation methods. We will revise Fig. 1 (see https://anonymous.4open.science/r/ICML-12294/revised_fig1.png) and the explanation to make this distinction clearer.
>
> See A2.4 for theory-implementation connections.
>
> **A3.2 Stability under more inference steps (Weakness 2, Limitation 2)**
>
>
>
> Following the suggestion, we extended the comparison on a relatively easy task (low-light enhancement) over a wider range of steps (1/3/10/20/30/50/100).
>
> As shown below, UDBM remains much more stable than DiffUIR: UDBM varies by only **0.22 dB** from best to worst, while DiffUIR varies by **1.48 dB**. This suggests that UDBM maintains a much more stable transport process as the number of inference steps increases.
>
> We will move the analysis to the main text as suggested.
>
> | steps | 1 | 3 | 10 | 20 | 30 | 50 | 100 |
> |---|---:|---:|---:|---:|---:|---:|---:|
> | DiffUIR | 24.45 | 25.26 | 24.81 | 24.47 | 24.28 | 24.04 | 23.78 |
> | UDBM | 26.55 | 26.58 | 26.55 | 26.50 | 26.47 | 26.43 | 26.36 |
>
> **A3.3 Comparison with BDG and other recent methods (Weakness 3, Limitation 3)**
>
> We should clarify that the results are not directly comparable due to different training/evaluation settings. Specifically, we follow the same protocol as DiffUIR [1] (e.g., merged deraining datasets with 13,712 training images, and GoPro for deblurring), whereas BDG uses a different AiO setting with more training data, e.g., Rain13K + SynRain13K (27,422 images) and GoPro + RealBlur.  Under matched settings (e.g., low-light/snow), UDBM (26.55/34.00) clearly outperforms RAM (24.88/32.75) and DCPT (25.39/32.79).
>
> Furthermore, BDG did not retrain DiffUIR on their new data. Since our research adheres to DiffUIR’s original setting, directly comparing our metrics to BDG is unfair.
>
>
>
>
> **A3.4 Adding IQA (Weakness 4)**
>
> Due to space limitations, we supplement the most commonly used NIQE for Tables 4 and 5, which will be included in the revision.
>
> | Method | Real Rain ↓ | Real Dark ↓ | Real Snow ↓ |
> |---|---:|---:|---:|
> | Prompt-IR | 3.52 | 3.31 | 2.79 |
> | DiffUIR | 3.38 | 3.14 | 2.74 |
> | AdaIR | 3.40 | 3.24 | 2.78 |
> | BioIR | 3.42 | 3.25 | 2.76 |
> | MOCE-IR | 3.45 | 3.18 | 2.75 |
> | HOGformer | 3.31 | 3.08 | 2.69 |
> | UDBM-L | **3.26** |**3.07** | **2.63** |
>
>
> | Method | L | H | R | S | L+H | L+R | L+S | H+R | H+S | L+H+R | L+H+S | Avg |
> |---|---:|---:|---:|---:|---:|---:|---:|---:|---:|---:|---:|---:|
> | DiffUIR | 4.18 | 4.17 | 4.02 | 4.32 | 6.70 | 6.19 | 6.72 | **4.42** | **4.10** | 6.07 | 5.87 | 5.16 |
> | AdaIR | 4.23 | 4.21 | 4.11 | 4.39 | 6.82 | 6.15 | 6.78 | 4.63 | 4.18 | 6.11 | 5.96 | 5.23 |
> | BioIR | 4.15 | 4.16 | 4.09 | 4.38 | 6.87 | 6.24 | 6.80 | 4.52 | 4.14 | 6.13 | 5.90 | 5.22 |
> | MOCE-IR | 4.21 | 4.22 | 4.06 | 4.41 | 6.76 | 6.23 | 6.85 | 4.56 | 4.19 | 6.15 | 5.95 | 5.24 |
> | HOGformer | 4.16 | 4.13 | 4.01 | 4.32 | 6.70 | 6.28 | 6.80 | 4.51 | 4.16 | 6.03 | 5.79 | 5.17 |
> | UDBM-L | **4.12** | **4.11** | **3.98** | **4.28** | **6.63** | **5.85** | **6.57** | 4.43| 4.11 | **5.98** |**5.85** |**5.08** |
>
> **A3.5 Omitted related work (Weakness 5)**
>
> Thank you for the pointer. These are important works in AiOIR. FoundIR has already been discussed in our current manuscript. We will add discussions for the remaining suggested works.
>
>
>
>
> **A3.6 Requirement for single inference step (Question 1)**
>
> UDBM supports both single- and multi-step inference (Algorithm 2 provides DDIM-style inference).  We emphasize one-step because it is highly effective. On the AiO benchmark, PSNR at 1/2/4/8/16 steps is 32.61/32.51/32.49/32.10/32.36 on average, showing near-saturation at step 1 (see A3.2). We will clarify this in the revision.
>
>
>
> **A3.7 Performance with the same architecture as DiffUIR (Question 2)**
>
> Please refer to A2.1.
>
> **A3.8 More parameters of the uncertainty model (Question 3)**
>
> We apologize for the typo. The two titles were mistakenly swapped in Table 9. The uncertainty model is actually smaller than the denoiser. Moreover, the parameters, FLOPs, and runtime in Table 1 already include the uncertainty model.

---

> > ### Author Rebuttal · Reviewer_xsLz · 2026-04-02
> >
> > The rebuttal addressed my concerns. I expect the authors to update the main text with results under multi-timesteps and more accurate diagrams. However, I also agree with reviewer 5LHp's concerns about experimental fairness, so I only raised the score to weak accept.

---

> > > ### Author Response · Authors · 2026-04-03
> > >
> > > We sincerely thank the reviewer for the positive follow-up.
> > >
> > > As suggested, we will revise the main text to (1) include the multi-timestep results more explicitly and (2) improve the diagrams to make the method description more precise and easier to follow.  (3) include the related works.
> > >
> > > We would also like to further clarify the concern regarding **experimental fairness**. There may still be some misunderstanding of our experiments by reviewer 5LHp. **All comparisons in our experiments are conducted under a fair setting.** In particular, the situation is **not** that only two compared methods used more powerful settings, while the others were evaluated under weaker ones. Instead, **all compared methods are evaluated under the same experimental protocol.**
> > >
> > > Specifically, for methods whose results had already been reported in prior works under the same unified setting (e.g., **Restormer, AirNet, PromptIR, and DA-CLIP**), we directly adopted the results reported in those papers [1,2]. For more recent methods whose results were not available under the same dataset (e.g., **AdaIR, BioIR, and MoCE-IR**), we **retrained** them while keeping their original model architectures unchanged, but using **the same training dataset, the same batch size, and the same patch size** as in our paper.
> > >
> > > Therefore, **all compared methods (including our UDBM) are evaluated under the same 256×256 patch size, the same batch size, and the same training dataset**, and our method is **not** compared under a more favorable or more powerful training condition.
> > >
> > > To avoid any ambiguity, we will make this protocol **more explicit in the revised manuscript**.
> > >
> > >
> > > [1] Selective Hourglass Mapping for Universal Image Restoration Based on Diffusion Model. CVPR 2024.
> > > [2] Beyond Degradation Conditions: All-in-One Image Restoration via HOG Transformers. AAAI 2026.

---

### Official Review · Reviewer_gkmu · 2026-03-12

**Soundness:** 3
**Presentation:** 3
**Significance:** 3
**Originality:** 3
**Overall Recommendation:** 3
**Confidence:** 4

**Summary:**

This paper proposes UDBM, an uncertainty-aware diffusion bridge framework for AiOIR under heterogeneous degradations. The method formulates image restoration as learning a diffusion bridge from degraded observations to clean images, and introduces a relaxed bridge design to better regularize the terminal dynamics while accounting for uncertainty in the degradation process. To handle diverse and composite corruptions with a single model, UDBM incorporates an uncertainty estimation mechanism to condition the restoration trajectory and adapt to varying degradation characteristics. The approach is evaluated across AiOIR and task-specific restoration settings, with additional tests on real-world and unseen composite degradations to assess generalization.

**Compliance With Llm Reviewing Policy:**

Affirmed.

**Final Justification:**

The authors substantially improves the empirical claims by rebuttal, but the theory-to-objective connection remains somewhat indirect. The contributions of backbone and formulation are not obvious enough. Hence, I prefer to keep my scores.

**Key Questions For Authors:**

1. The paper’s key novelty is the relaxed diffusion bridge / entropy-regularized transport perspective, but it is hard to disentangle its benefit from (i) the noise/schedule parameterization and (ii) the backbone/architecture choices. Could the authors provide controlled ablations?
2. You emphasize effective single-step inference, but the evidence seems limited. Could the authors compare against multi-step inference variants of the same trained model (or a version trained for multi-step), e.g., 1/2/4/8/16 steps?
3. The framework appears to rely heavily on an uncertainty estimator to model degradation uncertainty. Could the authors analyze robustness under imperfect uncertainty estimation?
4. The theoretical motivation via relaxed diffusion bridges and entropy-regularized transport is compelling, but the connection to the implemented training objective is not fully clear.

**Limitations:**

Yes

**Strengths And Weaknesses:**

Strengths:
1. The proposed relaxed diffusion bridge offers a theoretically motivated way to regularize the terminal dynamics and explicitly incorporate degradation uncertainty.
2. The experiments are relatively comprehensive, covering all-in-one restoration, task-specific settings, and generalization to real-world and unseen composite degradations.

Weaknesses:
1. The theoretical motivation (relaxed diffusion bridge and entropy-regularized transport) is interesting, but the paper provides limited empirical evidence isolating how much performance gain comes specifically from the proposed formulation versus the schedule parameterization or backbone architecture.
2. The claim of effective single-step inference is important but insufficiently validated; comparisons with multi-step variants of the same model would better justify this design choice.
3. The framework heavily depends on the uncertainty estimator, yet the robustness of the method under inaccurate uncertainty estimation or domain shift is not thoroughly analyzed.

---

> ### Author Rebuttal · Authors · 2026-03-30
>
> We sincerely thank Reviewer gkmu for the thoughtful comments. We address the questions and concerns below.
>
>
> **A2.1 Controlled ablations (Weakness 1 / Question 1)**
>
> We should clarify that the proposed schedules implement our theory (Sec. 3.4). Specifically, the terminal relaxation term realizes the relaxed diffusion bridge, while the path schedule instantiates entropy-regularized transport. We will explicitly strengthen this mapping in the revision.
>
> Our current ablation studies have already isolated these key components in Table 6:
>
> - Terminal relaxation term: Removing it degrades the constraint to a standard bridge.
>
> - Shared bridge term: Comparing against a fixed schedule proves its necessity.
>
> - Adaptive path schedule: Substituting it with a conventional schedule validates our superiority.
>
> We further conducted the following controlled studies:
>
> - S1: UDBM backbone + DiffUIR [1] schedule
> - S2: UDBM backbone + Brownian Bridge [2] schedule
> - S3: UDBM schedule + DiffUIR backbone
> - S4: end-to-end training with the UDBM backbone
>
>
>
> | Method | Average |
> |---|---:|
> | S1 | 29.63 |
> | S2 | 27.32 |
> | S3 | 32.69 |
> | S4 | 28.71 |
> | DiffUIR | 30.18 |
> | Ours | 32.61 |
>
> These results show that a traditional diffusion bridge (S2) is insufficient for AiOIR, even underperforming direct end-to-end training (S4). Similarly, the DiffUIR schedule with our backbone (S1) is inferior to the original DiffUIR. Our method with a DiffUIR backbone (S3) yields slight improvements, suggesting that much of the gain comes from the schedule. Our default model avoids the higher inference costs of DiffUIR's (Table 1) while keeping the efficiency.
>
>
> We will include these controlled ablations in the revision.
>
>
> **A2.2 Effective single-step inference (Weakness 2 / Question 2)**
>
> UDBM is not limited to one-step inference; Algorithm 2 supports DDIM-style multi-step sampling. As detailed in Appendix J.1, with the same trained model, the average PSNR on the in-distribution AiO benchmark at 1/2/5 steps is 32.61/32.51/32.47, demonstrating near-saturation at just one step. For the out-of-distribution CDD11 benchmark, additional steps provide slight gains, serving as a flow refinement against unseen degradations. Consequently, one-step inference serves as a balance between high-fidelity restoration and fast inference speed.
>
> We further report the average result (PSNR) on the AiO benchmark with 1/2/4/8/16 steps as suggested:
>
> | Steps | Average |
> |---|---:|
> | 1 | 32.61 |
> | 2 | 32.51 |
> | 4 | 32.49 |
> | 8 | 32.10 |
> | 16 | 32.36 |
>
>  We will move this analysis forward in the revision.
>
>
>
>
>
>
> **A2.3 Robustness to imperfect uncertainty estimation (Weakness 3 / Question 3)**
>
> Our current paper already contains two lines of evidence of the robustness to imperfect uncertainty estimation:
>
> - Table 6 compares three uncertainty estimators, showing that UDBM is not tied to one specific uncertainty formulation;
>
> - Appendix J.2 studies four settings for imperfect uncertainty estimation: uncertainty maps with added noise, noisy estimator inputs, global scaling, and fixed constant maps. The results indicate that UDBM does not rely on perfectly accurate uncertainty estimation; it degrades gradually rather than collapsing.
>
> To further strengthen this point, we provide quantitative comparison results. We injected Gaussian noise into the predicted uncertainty map and evaluated the result in low-light conditions without retraining. UDBM is robust to mild-to-moderate inaccuracy, while severe corruption of the uncertainty map degrades performance progressively rather than causing catastrophic failure. Importantly, NIQE remains relatively stable, suggesting no perceptual collapse.
>
> | σ | PSNR ↑ | SSIM ↑ | NIQE ↓ |
> |---:|---:|---:|---:|
> | 0.0 | 26.55 | .915 | 5.20 |
> | 0.1 | 26.58 | .915 | 5.19 |
> | 0.3 | 25.43 | .912 | 5.18 |
> | 0.5 | 25.41 | .914 | 5.17 |
> | 0.7 | 24.32 | .909 | 5.17 |
> | 1.0 | 23.57 | .907 | 5.14 |
>
>
> Please also see Q4.3 for more evidence (OOD scenario and extreme degradations).
>
> We will add clearer pointers in the main text to direct readers to these analyses.
>
>
> **A2.4 Theory vs. training objective (Question 4)**
>
> The theory of relaxed diffusion bridge and entropy-regularized transport is instantiated through the **diffusion process design** (diffusion schedule), rather than through an extra loss. Concretely, the terminal relaxation term realizes the relaxed terminal constraint, while the path schedule is motivated by the entropy-regularized transport analysis.
>
> Algorithm 1 then makes this explicit: at time step t, UDBM constructs $(\alpha_t,\gamma_t,\beta_t)$, forms ($x_t$), predicts ($x_0$), and optimizes the ($L_1$) loss.
>
> We agree that this theory-to-implementation mapping should be stated more explicitly, and we will revise Sec. 3.4 accordingly.
>
>
> [1] Selective Hourglass Mapping for Universal Image Restoration Based on Diffusion Model. CVPR 2024.
> [2] BBDM: Image-to-Image Translation with Brownian Bridge Diffusion Models. CVPR 2023.

---

> > ### Author Rebuttal · Reviewer_gkmu · 2026-04-06
> >
> > The authors substantially improves the empirical claims by rebuttal, but the theory-to-objective connection remains somewhat indirect. The contributions of backbone and formulation are not obvious enough. Hence, I prefer to keep my scores.

---

> > > ### Author Response · Authors · 2026-04-07
> > >
> > > Thank you for the follow-up and for acknowledging that the rebuttal substantially strengthens the empirical evidence. We appreciate your further comment and would like to clarify the two remaining points more precisely.
> > >
> > > **1. Connection between theory and formulation**
> > >
> > > Our intent is **not** to claim that every design choice is obtained as a closed-form derivation from the full theory. Rather, the proposed formulation (i.e., the diffusion schedule) should be understood as a **theory-grounded, tractable instantiation** of the underlying bridge/transport principles, which is also the common practice in related bridge-based methods [1,2,3].
> > >
> > > The theory plays two roles in UDBM:
> > >
> > > **(1) Relaxed diffusion bridge: direct instantiation.** Proposition 1 / Theorem 1 shows that replacing the strict terminal Dirac constraint with an uncertainty-aware Gaussian terminal distribution regularizes the singular drift.
> > >
> > > This is explicitly instantiated in the forward process through the **nonzero terminal variance** and the **terminal relaxation term** in the noise schedule.
> > >
> > > **(2) Entropy-regularized transport: principled approximation.** For the EOT / viscous HJB part, we agree that our path schedule is not an exact closed-form solver of the full coupled PDE system. Instead, the theory specifies the desired transport kinetics: slower traversal in high-uncertainty / high-entropy regions and faster motion in confident regions near the boundaries. Because directly solving the full coupled system is intractable in our setting, we instantiate this principle through a parametric time reparameterization.
> > >
> > > Therefore, this part should be understood as a **theory-grounded approximation** rather than as an exact derivation.
> > >
> > > The remaining concern seems to come from expecting a direct training objective or an exact closed-form schedule derivation from the full Schrödinger Bridge (SB) / transport theory. However, prior work [1,2,3] has explicitly noted that exact training can involve intractable forward score functions or costly simulation, and therefore adopts variational, simulation-free, or customized schedule-based approximations instead of exact solvers for the full theoretical object. UDBM follows the same general paradigm: the **relaxed bridge** is a direct structural instantiation, while the **path schedule** is a principled approximation consistent with the transport kinetics.
> > >
> > > We will revise the paper to make this distinction fully explicit.
> > >
> > > **2. Connection between backbone and formulation**
> > >
> > > We would also like to clarify that the **backbone itself is not the focus or novelty of our paper**. Our reason for adopting this backbone is mainly **efficiency**: compared with the backbone used in DiffUIR, it provides faster inference, and together with our **single-step inference** design, this leads to a substantially more efficient overall system, as shown in Table 1. Therefore, our claim is **not** that the backbone itself is novel or should independently bring a performance gain.
> > >
> > > Rather, the purpose of our controlled studies across different backbones is to show that the restoration gain does **not** come from the architecture itself, but from the proposed formulation/schedule.
> > >
> > > The results in rebuttal support two key conclusions.
> > >
> > > **First**, the backbone itself is **not** the source of the gain. If the backbone were the main reason, then using our backbone alone or combining it with existing schedules should already yield clear improvements. However, this is not observed: end-to-end training with our backbone gives only **28.71 dB**, and even **UDBM backbone + DiffUIR schedule** reaches only **29.63 dB**, both below the DiffUIR baseline.
> > >
> > > **Second**, the **schedule / formulation** is the main source of improvement. This is evidenced in two ways:
> > >
> > > 1. **Compared to the existing AiO formulation:** applying our proposed formulation to the DiffUIR backbone yields **32.69 dB**, which is a **+2.51 dB** gain over the original DiffUIR (**30.18 dB**).
> > >
> > > 2. **Compared to a standard diffusion bridge:** on the UDBM backbone, our AiO-tailored schedule yields **32.61 dB**, whereas replacing it with the standard Brownian Bridge schedule gives only **27.32 dB**. This **+5.29 dB** gap shows that a generic diffusion bridge is insufficient for AiOIR, and that the task-specific schedule design is critical.
> > >
> > >
> > > Therefore, the main improvement comes from the proposed **relaxed bridge + dual uncertainty-aware scheduling**, rather than from architecture.
> > >
> > > We will revise the paper to make it more explicit.

---

### Official Review · Reviewer_5LHp · 2026-03-12

**Soundness:** 3
**Presentation:** 3
**Significance:** 3
**Originality:** 3
**Overall Recommendation:** 3
**Confidence:** 5

**Summary:**

This paper introduces the Uncertainty-Aware Diffusion Bridge Model (UDBM), a novel framework that overcomes the conflicting optimization objectives in All-in-One Image Restoration (AiOIR) by reframing the restoration process as a pixel-wise, uncertainty-guided stochastic transport problem. To address the limitations of existing coarse and rigid methods, UDBM utilizes a relaxed diffusion bridge formulation that resolves drift singularity, paired with a dual modulation strategy: a noise schedule that aligns diverse degradations into a shared latent space, and an adaptive path schedule that dynamically guides the restoration trajectory. By effectively rectifying these transport dynamics, UDBM achieves state-of-the-art performance across multiple complex degradation types in just a single inference step.

**Compliance With Llm Reviewing Policy:**

Affirmed.

**Key Questions For Authors:**

Please see weakness

**Strengths And Weaknesses:**

Pros:
1. Overall, the paper is well-written and well-organized.
2. The author validated the approach across various image restoration tasks.
3. The ablation experiment is extensive.

Cons:
1. Unfair training settings. This work employed a larger patch size, rendering the experimental results unconvincing. Comparisons should be conducted under fair training settings.
2. Using diffusion bridge model for image restoration is not very novel, and this work appears to be a somewhat incremental contribution.
3. From the ablation experiments, it is evident that optimal parameters vary across different tasks, posing significant challenges for the practical application of the model.

---

> ### Author Rebuttal · Authors · 2026-03-30
>
> We sincerely thank the reviewer 5LHp for the constructive feedback. We address the issues point by point below.
>
> **A1.1 Fairness of the training setting (Cons 1)**
>
> We clarify that our method does not benefit from using a larger training crop. As stated in the Experiments, our research follows the setting of previous works [1,8]. Under this setting, the training patch size is set to **256×256 for all compared methods**. The concern may stem from other public AiOIR implementations [2,3] that adopt a 128×128 patch size for training.
>
> We will explicitly clarify this protocol in the revision.
>
>
>
>
> **A1.2 Novelty (Cons 2)**
>
>
> We respectfully disagree that the method is merely incremental. We agree that diffusion bridges are an important foundation. However, it is difficult for these traditional diffusion bridges (e.g., I$^2$SB [9], DDBM [10]) to handle the heterogeneity of degradations and their shared information in AiOIR. DDBM underperforms in AiOIR settings, as shown in S2 of A2.1. Our claim is not that we are the first to utilize a diffusion bridge in image restoration, but that we reformulate the diffusion bridge specifically for AiOIR, where a unified model must handle both the heterogeneity of degradations and their shared restoration structure.
>
> Our UDBM models the transition from heterogeneous degradation distributions to clean image distributions by considering the many-to-one nature of AiOIR tasks. Specifically, we construct this distribution transition process through the following three designs:
>
> (1) Relaxed bridge. We introduce an uncertainty-aware terminal distribution, which models the stochastic nature of heterogeneous degradations. More importantly, as shown in Proposition 1 and Theorem 1, while the standard bridge suffers from a drift singularity as $t \to 1$, our formulation regularizes the drift into a bounded, Lipschitz-continuous one.
>
>
> (2) Uncertainty-aware noise schedule. We design the noise schedule to progressively align heterogeneous degradations into a shared high-entropy latent space, while still preserving deterministic structural information from the input. This reduces optimization conflict across degradations and improves shared information learning for UDBM.
>
>
> (3) Uncertainty-adaptive path schedule: Motivated by entropy-regularized transport, the path schedule allocates denser refinement to high-uncertainty regions while enabling efficient transport in confident regions. This gives the transport process the adaptability needed for AiOIR-specific heterogeneity, which conventional fixed diffusion bridge schedules lack.
>
>
> In this sense, our contribution is a novel diffusion bridge formulation tailored to AiOIR, rather than an incremental reuse of existing diffusion bridge models. We will revise the paper to make the distinction from existing task-specific diffusion bridges clearer.
>
>
>
>
>
>
>
> **A1.3 Hyperparameters and practical deployment (Cons 3)**
>
> We do not tune hyperparameters separately for each degradation. We agree that different degradations may prefer slightly different hyperparameter values, but this is a common challenge in AiOIR due to degradation heterogeneity, rather than a method-specific issue. Similar behavior is also observed in prior AiO works [4,5,6]. Many prior works report ablations result only on a single degradation [2,3] or only in terms of average performance [7,8]. In contrast, we report the trends across different degradations and select hyperparameters based on the best average performance across all tasks, which is more comprehensive and consistent with the practical goal of unified deployment. Moreover, according to Fig. 5, within a reasonable hyperparameter range, the average performance varies roughly within 0.2-0.3 dB around the optimum, which indicates good robustness rather than the need for per-task retuning.
>
>
> [1] Selective hourglass mapping for universal image restoration based on diffusion model. CVPR 2024.
>
> [2] Bio-Inspired Image Restoration. NeurIPS2025.
>
> [3] Adair: Adaptive all-in-one image restoration via frequency mining and modulation. ICLR 2025.
>
> [4] Universal image restoration pre-training via degradation classification.ICLR 2025.
>
> [5] All-in-One Image Restoration via Causal-Deconfounding Wavelet-Disentangled Prompt Network. TIP 2026.
>
> [6] Learning Domain-Aware Task Prompt Representations for Multi-Domain All-in-One Image Restoration. ICLR 2026.
>
> [7] Complexity Experts are Task-Discriminative Learners for Any Image Restoration. CVPR 2025.
>
> [8] Beyond degradation conditions: All-in-one image restoration via hog transformers. AAAI 2026.
>
> [9] I$^2$SB: Image-to-Image Schrodinger Bridge. ICML 2023.
>
> [10] BBDM: Image-to-Image Translation with Brownian Bridge Diffusion Models. CVPR 2023.

---

> > ### Author Rebuttal · Reviewer_5LHp · 2026-04-01
> >
> > I cannot agree with the clarification regarding unfair training. The performance of image restoration models is highly dependent on training settings. Among the many methods compared, only two used more powerful training settings, which means the comparison with other methods remains unfair.

---

> > > ### Author Response · Authors · 2026-04-01
> > >
> > > We thank the reviewer for the follow-up.
> > >
> > > We would like to clarify that **all comparisons in Table 1 are conducted under a unified training protocol**. Specifically, for methods whose results were not taken directly from the original papers (e.g., AdaIR, BioIR, and MoCE-IR), we **retrained** them under **the same experimental setting**, i.e., with **the same 256×256 patch size, the same batch size, and the same training dataset**. For several additional baselines (e.g., Restormer, AirNet, PromptIR, and DA-CLIP), we report the results provided by DiffUIR [1] and HOGformer [2], where these methods were also retrained under the **same protocol**. Therefore, the comparison in Table 1 is obtained under a fair protocol.
> > >
> > >
> > > [1] Selective hourglass mapping for universal image restoration based on diffusion model. CVPR 2024.
> > >
> > > [2] Beyond degradation conditions: All-in-one image restoration via hog transformers. AAAI 2026.

---

### Decision · Program_Chairs · 2026-04-30

**Decision:**

Accept (regular)

**Comment:**

This paper addresses all-in-one image restoration by introducing a relaxed diffusion bridge to better model degradation uncertainty and avoid theoretical issues in standard formulations, as well as introducing a dual modulation strategy to enable flexible and effective restoration across different degradation types. It receives one accept, one weak accept and two weak rejects. The idea is novel and well-motivated and the experiments can validate the effectiveness of the proposed method. The decision of this paper is accept. However, the authors need to carefully revise the paper according to the reviews in the final version.